

# *crestr* An R package to perform probabilistic climate reconstructions using fossil proxies

Manuel Chevalier[1,2]

[1]Institute of Geosciences, Sect. Meteorology, Rheinische Friedrich-Wilhelms-Universität Bonn, Auf dem Hügel 20, 53121 Bonn, Germany
[2]Institute of Earth Surface Dynamics, Geopolis, University of Lausanne, Lausanne, Switzerland

**Correspondence:** Manuel Chevalier (chevalier.manuel@gmail.com)

**Abstract.** Statistical climate reconstruction techniques are practical tools to study past climate variability from fossil proxy data. In particular, the methods based on probability density functions (PDFs) are powerful at producing robust results from various environments and proxies. However, accessing and curating the necessary calibration data, as well as the complexity of interpreting probabilistic results, often limit their use in palaeoclimatological studies. To address these problems, I present

a new R package (`crestr`) to apply the CREST method (Climate REconstruction SofTware) on diverse palaeoecological datasets. `crestr` includes a globally curated calibration dataset for six common climate proxies (*i.e.* plants, beetles, chironomids, rodents, foraminifera, and dinoflagellate cysts) that enables its use in most terrestrial and marine regions. The package can also be used with private data collections instead of, or in combination with, the provided dataset. It also includes a suite of graphical diagnostic tools to represent the data at each step of the reconstruction process and provide insights into the effect

of the different modelling assumptions and external factors that underlie a reconstruction. With this R package, the CREST method can now be used in a scriptable environment, thus simplifying its use and integration in existing workflows. It is hoped that `crestr` will contribute to producing the much-needed quantified records from the many regions where climate reconstructions are currently lacking, despite the existence of suitable fossil records.

## 1   Introduction

Fossil-based climate reconstruction techniques are commonly used to quantify past climates and shed light on the nature of climate drivers across various spatial and/or temporal scales. Over the years, a growing number of distinct techniques of increasing complexity has been proposed, each one being based on a unique set of assumptions regarding the modelling of ecological datasets and their translation into climate parameters (*e.g.* Birks et al. (2010), Chevalier et al. (2020b)). Of this array

of techniques, Weighted-Averaging (WA, ter Braak and van Dame (1989)), Weighted-Averaging Partial Least Square (WA-PLS, ter Braak et al. (1993)) and the Modern Analogue Technique (MAT, Overpeck et al. (1985)) have been the most widely employed. Despite their conceptual simplicity and demonstrated capacity to reliably reconstruct climate from palaeoecological





datasets, the limited availability of robust calibration datasets (*i.e.* regional collections of modern proxy samples) beyond the Northern Hemisphere extratropics has, however, hindered their application in these regions, despite the existence of suitable
records from all environments worldwide (Chevalier et al., 2020b).

The abundance of such un-quantified fossil pollen records from southern Africa triggered the development of the CREST (Climate REconstruction SofTware) method (Chevalier et al., 2014). Built upon from the original work of Kühl et al. (2002) —— who first proposed to replace the commonly-used modern proxy samples with modern proxy geolocalised occurrence data to estimate probabilistic proxy-climate relationships — CREST estimates and combines probability density functions
(PDFs) to reconstruct climate parameters. Using a private collection of modern plant occurrences held by the South African National Botanical Institute (SANBI), CREST has been successfully employed to reconstruct diverse temperature, precipitation and moisture-related variables for different time periods across the southern African tropical and subtropical regions (see for instance Chase et al. (2015b), Chase et al. (2015a), Chevalier and Chase (2015), Chevalier and Chase (2016), Lim et al. (2016), Cordova et al. (2017)). A global, multi-proxy calibration dataset containing millions of modern occurrence data was
subsequently released (Chevalier, 2019) to support the use of CREST beyond southern Africa and contribute to the creation of quantified climate records worldwide (Yi et al. (2020), Hui et al. (2021)).

In addition to its broad applicability, CREST also bears some fundamental statistical properties that make it well-adapted to the analysis of palaeoecological datasets (Chevalier et al., 2020b). While techniques such as MAT or WA-PLS are primarily designed to associate modern proxy observations with their 'most likely' or 'mean' climate values only, CREST estimates,
weights and propagates all the climate values that are compatible with the observed fossil data. The posterior climate reconstructions obtained from CREST can thus be understood as a weighted ensemble of all plausible climate values and not a simpler, less informative 'most likely' or 'best' climate estimate with statistical errors (see the conceptual Fig. 1). While the latter might be optimal when a sample is analysed in complete isolation, the presence of independent, local or regional information (*i.e.* other samples for the same core or independent records) usually provides additional information that may not
always be consistent with such an isolated, analytical solution. In practice, joint solutions based on all the available information often differ from most likely climate estimates (Fig. 1). Using methods that can model the full range of climate uncertainties associated with a proxy sample is thus critical to bring various reconstructions and even climate simulations together in the most cohesive way.

Despite their definite advantages, uncertainty-based frameworks like CREST can be complex to employ, and their results
are challenging to process and represent. To facilitate access and use of such probabilistic techniques to a broad range of non-experts, I present a new R package `crestr` that enables employing CREST and its built-in calibration dataset in a scriptable environment. In addition to its technical core, the package also contains an array of graphical diagnostic tools to represent the data at different pivotal steps of the reconstruction process and facilitate objective evaluations of the data and results. This paper is structured as follows. First, Section 2 summarises the mathematics and assumptions underpinning the approach and
introduces the embedded calibration dataset. Then, Section 3 explains the philosophy and main elements of the package and describes the format of the different input files required. Finally, Section 4 documents a step-by-step guided tour of the package,

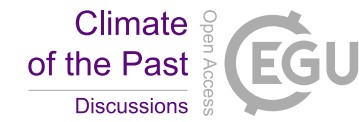

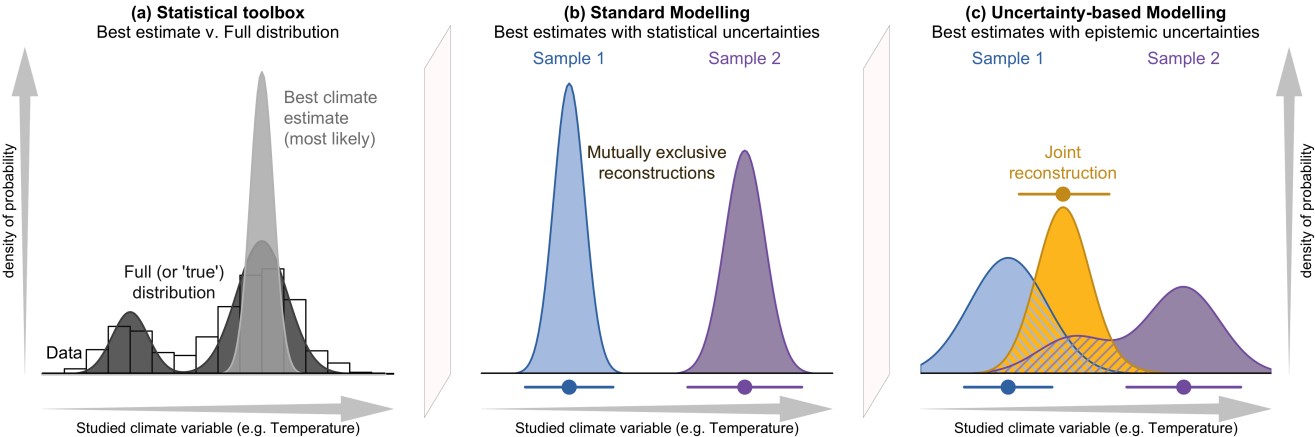

**Figure 1.** (a) Conceptual illustration of the differences between a modelling approach based on the estimation of the most likely climate where all the probabilities are concentrated around the mode (light grey), and a modelling approach focused on the full spread of the data with the probabilities more spread along the climate gradient (dark grey). In both cases, the area under the curve sums to one. The two types of approaches are illustrated in (b) and (c) with two theoretical fossil samples (in blue and purple) representing two independent reconstructions of the same climatic parameter for the same time period. (b) The two reconstructions are derived from a method focused on only estimating the most likely climate value, resulting in 'apparently' incompatible reconstructions. (c) The same samples are reconstructed using an approach that estimates their complete uncertainty distributions. In this case, the response of the blue sample is broader, and the response of the purple sample becomes bimodal. When the full spread of these uncertainties is considered, the two reconstructions are not incompatible anymore, and a joint climate estimate (gold) can be derived from their overlapping sections (hashed polygons).

illustrating the successive stages of a CREST analysis and how to use the diagnostic tools to reproduce a recently published pollen-based temperature reconstruction (Chevalier et al., 2021).

## 2 CREST method & Calibration data

### 2.1 The probability density function (PDF) approach

As is standard with statistical climate reconstruction techniques, the core process of CREST can be decomposed into two major stages: 1) estimating the modern climatic responses of the proxies observed in the fossil sequence (Fig. 2a-c) and 2) combining these responses to reconstruct past climates (Fig. 2d). In the following sections, the main elements of these two stages are presented along with all the parameters and/or modelling assumptions that can be modified in `crestr`. For an in-depth description of the method and its assumptions, the reader is, however, referred to .

As is standard with statistical climate reconstruction techniques, the core process of a CREST reconstruction can be decomposed into two major stages: 1) estimating the modern climatic responses of the proxies observed in the fossil sequence (Fig. 2a-c) and 2) combining these responses to reconstruct past climates (see Fig. 2d). The main elements of these two stages are





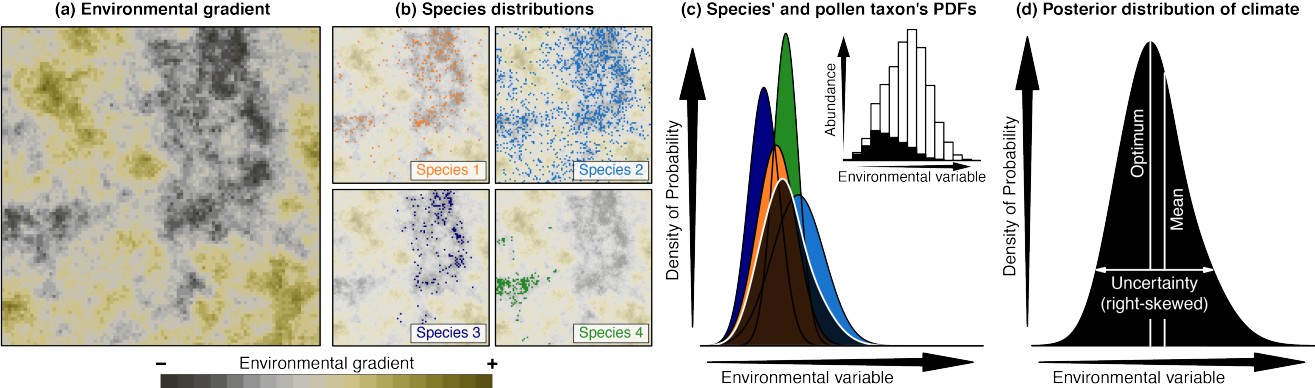

**Figure 2.** Conceptual representation of CREST using artificial pollen data. (a) Spatial distribution of a climate variable to reconstruct (e.g. temperature). (b) Occurrences of four species part of the same pollen group exhibiting marked preferences for the lowest values of that climate (e.g. dark/cold values) cross the study area. (c) Combination of the four species PDFs (colours) into the pollen PDF (black). The histogram represents the proportion of the modern climate space (white) occupied by at least one of the four species (black), highlighting the higher chances of observing the taxon at the lower end of the climate gradient. (d) Example of a posterior climate distribution resulting from the multiplication of PDFs and the type of synthetic statistics (e.g. optimum, mean, uncertainty range) one can derive from it.

presented in the following sections, along with all the parameters and/or modelling assumptions that can be modified in 'crestr,.
For an in-depth description of the method and its assumptions, the reader is, however, referred to Chevalier et al. (2014).

### 2.1.1   Stage 1: Modelling the proxy-climate relationships

In CREST, PDFs are used to transform the information contained in the modern observations of biological climate proxies into probabilistic climate responses. A PDF thus represents a weighted ensemble of all the conditions where the proxy is observed today. PDFs can be fitted in one or two steps depending on the nature and taxonomic resolution of the studied proxy. Climate
responses are first fitted at the species level (hereafter $\mathrm{PDF_{sp}}(c,s)$ with $c$ representing the studied climate variable and $s$ a species), and when necessary, these $\mathrm{PDF_{sp}}(c,s)$ are then combined together to meet the taxonomic resolution of the fossil taxon (hereafter $\mathrm{PDF_{tx}}(t,c)$ with $t$ representing the observed taxon).

For all the species, their empirical mean climate and associated variance are calculated from their modern distributions following Eq. 1 and 2. These two parameters can be intuitively interpreted as the climate preference and tolerance of the
species. For a robust estimation, it is recommended to exclude species with too few observations. Different studies have shown that a threshold of a minimum of $\mathrm{N}_s \geq 20\text{-}25$ distinct occurrences usually leads to robust estimates (*e.g.* Chevalier et al. (2014), Chevalier et al. (2021)). However, this number can vary between regions and climates. Each observation can also be weighted to account for the uneven distribution of modern climate (Kühl et al. (2002), Bray et al. (2006)). Extreme values are usually under-represented (see, for instance, the inset histogram on Fig. 2c), and this bias can push the estimation of the $\mathrm{PDF_{sp}}(c,s)$
towards the mean climate observed across the study area (*i.e.* towards the centre of the "climate space"). It can also artificially





shrink the posterior range of the reconstructions. In CREST, this weighting can be accounted for by first sorting the N climate values that compose the climate space into bins of equal sizes (*e.g.* 2°C or 50 mm). Then, each climate value is given a weight defined as the inverse of the abundance of the bin it belongs to (Eq. 3). With this correction, abundant climate values are down-weighted, while the rarer ones are up-weighted, and the distribution of modern climate is more 'balanced'.

$$\overline{m}_{s,c} \;=\; \frac{1}{\sum_i k(c_i)} \sum_{i=1}^{N_s} k(c_i)c_i \tag{1}$$

$$s_{s,c}^2 \;=\; \frac{1}{\sum_i k(c_i)} \sum_{i=1}^{N_s} k(c_i)\left(c_i - \overline{m}_{s,c}^2\right) \tag{2}$$

$$k(c_j) \;=\; \left(\frac{1}{N}\sum_{i=1}^{N} \mathbf{1}_{c_i \in \mathrm{bin}_{c_j}}\right)^{-1} \tag{3}$$

These two parameters, $\overline{m}_{s,c}$ and $s_{s,c}$, are used to define a regular, unimodal distribution for the $\mathrm{PDF}_{\mathrm{sp}}(s,c)$. In CREST, the shape of these distributions is pre-defined to be either normal (Eq. 4) or log-normal (Eq. 5). The decision should be made based

on the climate variable to reconstruct. In general, log-normal distributions are recommended for variables that are undefined for negative values, such as precipitation variables.

$$\mathrm{PDF}_{\mathrm{sp}}(s,c) \;=\; \frac{1}{\sqrt{2\pi s_{s,c}^2}} \exp\left(-\frac{(c - \overline{m}_{s,c})^2}{2s_{c,s}^2}\right) \tag{4}$$

$$\mathrm{PDF}_{\mathrm{sp}}(s,c) \;=\; \frac{1}{\sqrt{2\pi\sigma^2 c^2}} \exp\left(-\frac{(\ln(c) - \mu)^2}{2\sigma^2}\right) \tag{5}$$

$$\text{with} \begin{cases} \mu = \ln(\overline{m}_{s,c}) - \frac{1}{2}\ln\left(1 + \frac{s_{s,c}^2}{\overline{m}_{s,c}^2}\right) \\ \sigma^2 = \ln\left(1 + \frac{s_{s,c}^2}{\overline{m}_{s,c}^2}\right) \end{cases}$$

Finally, the $\mathrm{PDF}_{\mathrm{sp}}(s,c)$ of the $S(t)$ species composing taxon $t$ are linearly combined to create the climate response of taxon $t$ to climate variable $c$ (Eq. 6). This linear combination ensures that all the climate values that support the presence of at least one species have a non-null probability in the $\mathrm{PDF}_{\mathrm{tx}}(t,c)$. Contrary to the previous step, no additional constraints are added here. The distribution of the $\mathrm{PDF}_{\mathrm{tx}}(t,c)$ can thus be asymmetrical and even multimodal if different (groups of) composing species exhibit distinct climate requirements. An additional option is to weigh the different $\mathrm{PDF}_{\mathrm{sp}}(s,c)$ by the square root of

the number of individual grid cells composing their distribution ($N_s$). Considering that it is harder to estimate robust parameters with few points, this weighting gives more importance to the species with more extensive geographical distributions today, or, to say it differently, it gives more weight to the species whose climate responses can be the most robustly defined.





$$\text{PDF}_{\text{tx}}(t,c) \;=\; \frac{1}{\sum_s \sqrt{\text{N}_s}} \left( \sum_{s=1}^{S(t)} \sqrt{\text{N}_s}\; \text{PDF}_{\text{sp}}(s,c) \right) \tag{6}$$

### 2.1.2 Stage 2: Reconstructing climate

With the $\text{PDF}_{\text{tx}}(t,c)$ calibrated, posterior climate reconstructions can be estimated from their multiplication (Eq. 7, where $z$ represents the age or depth of the sample to reconstruct, and Fig. 2d). Due to their exponential nature, the $\text{PDF}_{\text{tx}}(t,c)$ rapidly converge to zero beyond the range of the modern observations of its composing species. As such, the multiplication of these $\text{PDF}_{\text{tx}}(t,c)$ ensures that only the climate values where all the taxa can coexist have a non-null (or, more precisely, a non-infinitesimal) probability. In addition, the presence or absence of each taxon is considered independent from the others. As

such, it is possible to select a subset of climatically sensitive taxa to reconstruct each climate variable and maximise the reconstruction signal (Chevalier and Chase, 2015), even if it is not always mandatory (Chevalier et al., 2021). These definitions of sensitive taxa are always specific to a specific region and variable and should not be generalised (Chevalier et al., in press).

$$\text{PDF}_{\text{rcnstrctn}}(c,z) \;=\; \left( \prod_{t=1}^{T(z)} \text{PDF}_{\text{tx}}(t,c)^{\omega(t,z)} \right)^{\left( \sum_t \omega(t,z) \right)^{-1}} \tag{7}$$

Finally, the selected $\text{PDF}_{\text{tx}}(t,c)$ can be weighted with the parameter $\omega(t,z)$, which can take any positive values. For pres-

ence/absence observations, the weights of taxon $t$ will be either one (the taxon is observed) or zero (the taxon is not observed). For compositional data, the weights can be the observed percentages (*i.e.* values between 0 and 100) or rescaled percentages when the observed percentages are not directly proportional to the taxa composition across the catchment. In most cases, using raw percentages to weight taxa can give considerable weight to abundant, ubiquitous taxa that do not have a well-defined climate response and strongly limit the influence of rarer taxa with more explicit climate preferences.

An empirical normalisation is proposed in CREST to account for the varying production rates, distribution and preservation processes impacting the relative proportions of fossil taxa observed in the sediments (Chevalier et al., 2014). This scaling entails calculating the average percentage of each taxon when it is present and dividing all its percentages by this taxon-specific scaling factor (Eq. 8, where $O(t,z)$ represents the observed percentage of taxon $t$ at age or depth $z$). With this normalisation, all the taxa vary on a standardised scale. The average presence is given a weight of 1, and values below and above this average

presence threshold are assumed to represent lower and higher abundance in the environment. While the empirical nature of this solution makes it imperfect and sensitive to the quality of the data, it nevertheless enables using percentages to inform the reconstructions. One alternative solution is to convert the percentages to presence and absence, but this approach can be limiting when most of the observed variability concerns changes in the relative proportions of different taxa, rather than a clear succession of taxa with different climate requirements (see for instance the pollen diagram in Chase et al. (2015b)). These

different weighting options are available in crestr, but users can also provide a weighting strategy of their design to better account for the specificity of their data.





$$\omega(t,z) \; = \; \frac{O(t,z)}{\left(\sum_z O(t,z)\right) \, / \, \left(\sum_z \mathbf{1}_{O(t,z)>0}\right)} \tag{8}$$

## 2.2 CREST calibration dataset

A multiproxy calibration dataset to estimate PDFs from a global collection of geolocalised presence-only data (hereafter proxy
distributions) was first presented in Chevalier (2019). These data were obtained from the Global Biodiversity Information
Facility (GBIF) database, an online collection of geolocalised observations of biological entities (GBIF, 2018). The calibration
dataset (hereafter *gbif4crest*, Chevalier, 2020) contains the species distributions of six common palaeoecological fossils: the
five taxa presented in the original version of the dataset — plants (GBIF (2020l), GBIF (2020h), GBIF (2020k), GBIF (2020f),
GBIF (2020g), GBIF (2020m), GBIF (2020i), GBIF (2020n), GBIF (2020j), GBIF (2021a), GBIF (2021b)) for fossil pollen
and macrofossils, chironomids (GBIF, 2020b), beetles (GBIF, 2020a), diatoms (GBIF, 2020c) and foraminifera (GBIF, 2020d)
– to which rodents (GBIF, 2020e) were recently added (Fig. 3). These data were curated and stored in a relational database to
ensure the consistency of the data.

The coordinates of all the presence records of these six common palaeoecological fossil proxies were upscaled at a spatial
resolution of $0.25 \times 0.25°$ (hereafter QDGC for Quarter-Degree Grid Cell). They were subsequently associated with
terrestrial and oceanic environmental variables at the same resolution ((Fick and Hijmans (2017), Zomer et al. (2008), Locarnini
et al. (2019), Zweng et al. (2018), Garcia et al. (2019a), Garcia et al. (2019b), Reynolds et al. (2007), see details in Tables 1 and
2). The QDGC spatial resolution is an empirical trade-off between numerous factors, including the resolution of the presence
data, the quality of the data or the spatial representativity of the studied proxy (see discussions in Chevalier et al. (2014) and
Chevalier (2019)). However, this trade-off may be suboptimal in some situations, and for that reason, `crestr` can also be
used with the raw GBIF data and even alternative calibration datasets.

In its current version (V2), the *gbif4crest* calibration dataset contains about 25.3 million unique presence data for the six
proxies. Unfortunately, the density of data available varies between proxies and regions (Fig. 3). Plant data dominate the
calibration dataset (>22 million unique occurrences) and allow for the use of `crestr` across all landmasses where vegetation
currently grows. For the five other proxies, the calibration data are not as extensive. However, these datasets are regularly
updated by GBIF. For example, the first version of the gbif4crest dataset released in 2018 contained about 17.5 million QDGC
entries, while the new version contains approximately 25.3 million entries (~44% increase). The range of 'reconstructible'
areas is thus rapidly broadening (see, for instance, the coverage of Russia by plant data compared to the first version of the
gbif4crest dataset presented in Chevalier (2019)).

The *gbif4crest* database is composed of three main types of data: taxonomic data (`TAXA` table on Fig. 4), distribution
data (`DISTRIB` and `DISTRIB_QDGC` tables) and diverse geopolitical, climatological, and environmental data (`DATA_QDGC`
table). Its structure is slightly different from the first version presented in Chevalier (2019), with a grouping of all the sep-
arate QDGC tables in a unique `DATA_QDGC` table to enable faster data extraction. In addition, various environmental and
geographical descriptors were also introduced to characterise each grid cell and allow a more refined data selection. These in-
clude elevation and elevation variability (Amante and Eakins, 2009), the country (https://www.naturalearthdata.com) or ocean





**Figure 3.** Data density of the six climate proxies available in the gbif4crest calibration database. The total number of unique species occurrences (N) is indicated for each proxy. The maps are based on the 'Equal Earth' map projection to better account for the relative sizes of the different continents.



**Table 1.** List of terrestrial variables available in the gbif4crest database. Each one can be selected in crestr using its associated code. List of abbreviations: (Temp.) Temperature, (Precip.) Precipitation.

| Terrestrial variables | | |
|---|---|---|
| Code | Full name | Source |
| bio1 | Mean Annual Temp. (°C) | Fick and Hijmans (2017) |
| bio2 | Mean Diurnal Range (°C) | Fick and Hijmans (2017) |
| bio3 | Isothermality (x100) | Fick and Hijmans (2017) |
| bio4 | Temp. Seasonality (standard deviation x100) (°C) | Fick and Hijmans (2017) |
| bio5 | Max Temp. of the Warmest Month (°C) | Fick and Hijmans (2017) |
| bio6 | Min Temp. of the Coldest Month (°C) | Fick and Hijmans (2017) |
| bio7 | Temp. Annual Range (°C) | Fick and Hijmans (2017) |
| bio8 | Mean Temp. of the Wettest Quarter (°C) | Fick and Hijmans (2017) |
| bio9 | Mean Temp. of the Driest Quarter (°C) | Fick and Hijmans (2017) |
| bio10 | Mean Temp. of the Warmest Quarter (°C) | Fick and Hijmans (2017) |
| bio11 | Mean Temp. of the Coldest Quarter (°C) | Fick and Hijmans (2017) |
| bio12 | Annual precip. (mm) | Fick and Hijmans (2017) |
| bio13 | Precip. of the Wettest Month (mm) | Fick and Hijmans (2017) |
| bio14 | Precip. of the Driest Month (mm) | Fick and Hijmans (2017) |
| bio15 | Precip. Seasonality (Coefficient of Variation) (mm) | Fick and Hijmans (2017) |
| bio16 | Precip. of the Wettest Quarter (mm) | Fick and Hijmans (2017) |
| bio17 | Precip. of the Driest Quarter (mm) | Fick and Hijmans (2017) |
| bio18 | Precip. of the Warmest Quarter (mm) | Fick and Hijmans (2017) |
| bio19 | Precip. of the Coldest Quarter (mm) | Fick and Hijmans (2017) |
| ai | Aridity Index (unitless) | Zomer et al. (2008) |

(https://www.marineregions.org) names, as well as different levels of ecological classification for the terrestrial (Olson et al., 2001) and marine (Costello et al., 2017) realms. The first and last observation dates are also now included, along with the type of observation, as reported by GBIF (see `DISTRIB_QDGC` table on Fig. 4). Finally, the `DATA_QDGC` table was entirely recalculated using a new protocol that better accounts for coastal margins, implying that some coastal climate values will differ, however marginally, between the different versions of the gbif4crest dataset.

Due to its large size (about 15 Gb), this database is not downloaded when installing the package, but it can be accessed in different ways. First, the data are stored in an open-access, cloud-based PostgreSQL database that can be accessed via `crestr`. This solution is the recommended option, as users without any SQL knowledge can benefit from the package's interface to automatically query the database by providing study-specific parameters (*e.g.* the name of the taxa or geographical boundaries for the study area) to import all the necessary data in the correct format to the R environment (see Section 4.2). Second,

advanced users can also directly query the database to extract and curate data from the `DISTRIB` or `DISTRIB_QDGC` tables



**Table 2.** List of marine variables available in the gbif4crest database. Each one can be selected in crestr using its associated code. List of abbreviations: (SST) Sea Surface Temperature, (SSS) Sea Surface Salinity.

| Oceanic variables | | |
|---|---|---|
| Code | Full name | Source |
| sst_ann | Mean Annual SST (°C) | Locarnini et al. (2018) |
| sst_jfm | Mean Winter SST (°C) | Locarnini et al. (2018) |
| sst_amj | Mean Spring SST (°C) | Locarnini et al. (2018) |
| sst_jas | Mean Summer SST (°C) | Locarnini et al. (2018) |
| sst_ond | Mean Fall SST (°C) | Locarnini et al. (2018) |
| sss_ann | Mean Annual SSS (PSU) | Zweng et al. (2018) |
| sss_jfm | Mean Winter SSS (PSU) | Zweng et al. (2018) |
| sss_amj | Mean Spring SSS (PSU) | Zweng et al. (2018) |
| sss_jas | Mean Summer SSS (PSU) | Zweng et al. (2018) |
| sss_ond | Mean Fall SSS (PSU) | Zweng et al. (2018) |
| icec_ann | Mean Annual Sea Ice Concentration (%) | Reynolds et al. (2007) |
| icec_jfm | Mean Winter Sea Ice Concentration (%) | Reynolds et al. (2007) |
| icec_amj | Mean Spring Sea Ice Concentration (%) | Reynolds et al. (2007) |
| icec_jas | Mean Summer Ice Concentration (%) | Reynolds et al. (2007) |
| icec_ond | Mean Fall Sea Ice Concentration (%) | Reynolds et al. (2007) |
| diss_oxy | Dissolved Oxygen Concentration (µmol/L) | Garcia et al. (2018a) |
| nitrate | Nitrate Concentration (µmol/L) | Garcia et al. (2018b) |
| phosphate | Phosphate Concentration (µmol/L) | Garcia et al. (2018b) |
| silicate | Silicate Concentration (µmol/L) | Garcia et al. (2018b) |

using the `dbRequest()` function and subsequently associate these data with climate variables. Finally, the full *gbif4crest* calibration dataset can be downloaded as an SQLite3 portable database file from Chevalier (2020).

## 3   The `crestr` R package

### 3.1   Philosophy of the package

The `crestr` package has been designed for two independent but complementary modelling purposes. The probabilistic proxy-climate responses can be used to quantitatively reconstruct climate from their statistical combination, such as in Chevalier et al. (2021) or Chevalier and Chase (2015), or, alternatively, they can be used in a more qualitative way to determine the (relative) climate sensitivities of different taxa in a given area to characterise past ecological changes, such as in Chevalier et al. (in press) or Quick et al. (2021). For a simpler access to the different functionalities, default values are provided for all parameters

to enable a rapid generation of preliminary results that users can then use as a starting point to fine-tune the model to their





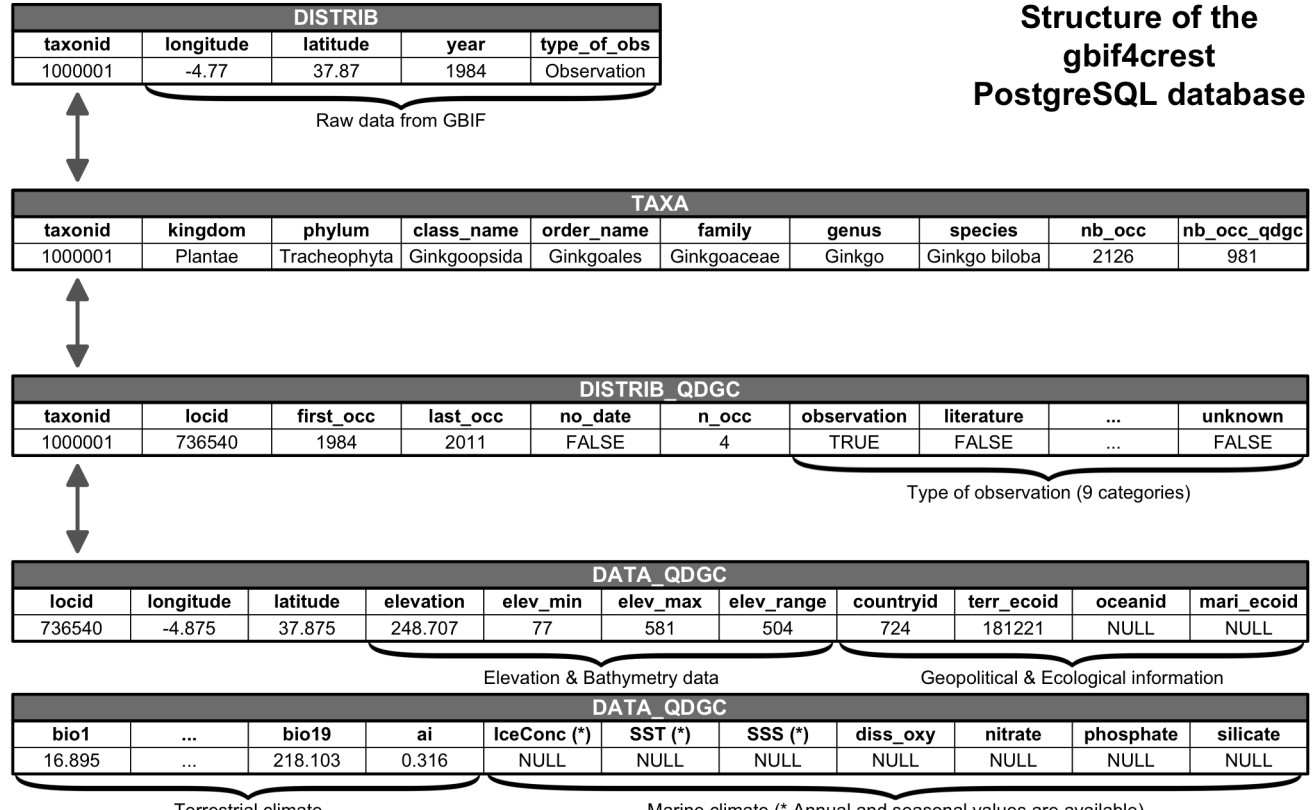

**Figure 4.** Structure of the gbif4crest PostgreSQL database. By default, the package extracts data from the TAXA, DISTRIB-QDGC and DATA-QDGC tables. The DISTRIB table contains the raw occurrence data obtained from GBIF and can be used to process the data at a different spatial resolution.

data. To guide users in this task and avoid the common 'black box' criticism faced by many complex statistical tools, a large array of publication-ready, graphical diagnostic tools was designed to automatically represent CREST data in a standardised way. These tools include plots of the calibration data, the estimated climate responses, the reconstructions and more. They can be used to look at the data from different perspectives to help interpret the results, and possibly to help identify potential

issues or biases in the selected data and/or parameters. Such diagnostic tools are available for every stage of the process and, as exemplified in section 4, they can be generated with a single line of R code.

### 3.2   The central element: the `crestObj` object

In `crestr`, all the CREST-related data are stored within a single S3 object of the class `crestObj`. Most functions of the package will take a `crestObj` as their main input and will return an updated version of that object. In practice, a `crestObj`

is a nested list that contains five sub-lists, each one grouping a specific type of information, such as the calibration data, the





**Figure 5.** Structure of a crestObj, here called 'rcnstrctn', with the five main sub-lists in colour. For simplicity, lists with many elements ('tax' or 'clim') are represented with double framed boxes. The unframed terminal nodes on the right-hand side of each branch are simple R objects, such as numbers, characters, vectors or data frames. The names of the functions that modify the objects in the sub-lists are indicated on the right.



fitted climate responses, and the reconstructions (Fig. 5). Wrapper functions have been implemented to manipulate and/or modify the information contained in a `crestObj` so that, users are never expected to manually modify their `crestObj` — even if it is possible. The five sub-lists contain the following information:

- `inputs`: contains the raw data (*e.g.* the counts/percentages, the ages of the samples or the names of the fossil taxa).

- `parameters`: contains the parameters provided at the different stages of the analysis (*e.g.* the data extraction or the fitting and combination of the PDFs)

- `modelling`: contains all the data related to the estimation of the PDFs (*e.g.* the distribution data for calibration, the climate space or the PDFs themselves).

- `reconstructions`: contains all the results (*e.g.* best estimates, synthetic error measurements as well as the full
posterior distribution of the uncertainties).

- `misc`: contains some additional metadata relative to the reconstruction (*e.g.* the site location or, most importantly, information relative to the proxy-species equivalency described in section 3.3.2).

## 3.3 Input data for `crestr`

Five different input data files are compatible with crestr. However, most applications will only require two file (the `df` and `PSE`
files, see below) to be created. More specific applications may require up to four of these files. All the files can be prepared outside the R environment and imported using standard R functions.

### 3.3.1 The fossil data (`df`)

The `df` data frame is only required if `crestr` is used for reconstructing climate and can be omitted if the objective is limited to modelling the climate response(s) of different taxa. `df` should be a data frame with the different samples entered as rows,
with either the age, depth, or sample ID as the first column and the fossil data in the subsequent columns. `df` can contain raw counts, percentages, presence/absence (1s and 0s) or even relative weights to be used in the reconstruction (see examples in Section 4.5).

### 3.3.2 The proxy-species equivalency (`PSE`) table

The `PSE` data frame is required to use the *gbif4crest* calibration dataset. It is used to associate fossil taxon names to the species
distributions of various species stored in the `TAXA` and `DISTRIB_QDGC` tables (Fig. 4). When all the fossil taxa are identified at the species level, the `PSE` table should be a simple data frame with one row per taxon. In most cases, however, fossil taxa are identified at a higher taxonomic level (sub-genus, genus, sub-family, family). These varying levels of identification should be encoded in the `PSE` file to link one or more (groups of) species to their common fossil taxon name.

In practice, a `PSE` file is composed of five columns (Table 3). The first one (*Level*) contains an integer that indicates the level
of taxonomic resolution of the row (1 for Family, 2 for Genus, 3 for Species and 4 for taxa that should be excluded from the



**Table 3.** Example classification of four pollen taxa from the example case study, each one with a different level of taxonomic resolution. The last column 'Taxonomic resolution' is added here for explanatory purposes only and is not required in a real 'PSE' table.

| Level | Family | Genus | Species | ProxyName | Taxonomic resolution |
|---|---|---|---|---|---|
| 1 | Asteraceae | | | Asteracae undiff. | Family |
| 2 | Asteraceae | *Stoebe* | | *Stoebe*-type | Subfamily |
| 2 | Asteraceae | *Elytropappus* | | *Stoebe*-type | Subfamily |
| 2 | Asteraceae | *Artemisia* | | *Artemisia* | Genus level |
| 3 | Arecaceae | *Elaeis* | *Elaeis guineensis* | *Elaeis guineensis* | Species |
| 4 | | | | Triletes spores | To be excluded |

reconstruction, *e.g.* 'Triletes spores' in the example case study below). The fifth column *ProxyName* contains the name of the taxon and columns two to four contain the taxonomic classification of that taxon as *Family*, *Genus* and *Species*, respectively. For simplicity, a pre-formatted version of the `PSE` table with the names of all the taxa to study can be generated by `crestr` using the `createPSE(list_of_taxa)` function that generates a spreadsheet with the correct structure and with the Level
and ProxyName columns automatically filled in.

When performing the species to proxy classification with the `crest.get_modern_data()` function (see Section 4.2), `crestr` first classifies the taxa with the lowest taxonomical resolution (*i.e.* when *Level* is equal to one), and then increases the resolution *Level* by *Level*. In the example in Table 3, different taxonomic resolution levels are provided for different plant species belonging to the highly diverse Asteraceae family (the daisy family). To distribute all the Asteraceae species
observed across the study area to their appropriate category, all the species are first classified as 'Asteraceae undiff.'. In a second time, the classification of some of these Asteraceae species is refined when reaching the better resolved sub-groups (*Stoebe*-type and *Artemisia* at *Level* 2). At the end of the process, the 'Asteraceae undiff.' group only contains Asteraceae species that grow in the study area but are not part of the genuses *Stoebe*, *Elytropappus* or *Artemisia*. The latter are categorised separately as *Stoebe*-type or *Artemisia*. Additional taxa can also be included to the `PSE` file to exclude species that are known
to not be part of a group. For instance, this could have been used to simplify the climate response of the 'Asteraceae undiff.' group by excluding more species from it, even if the pollen grains corresponding to these species have not been observed. This categorisation process can be time-consuming, as all the taxa must be classified in a unique `PSE` table, and will often require many iterations to be optimised. The results of the different assignments are stored in the `crestObj` returned by the `crest.get_modern_data()` function and can be evaluated by checking `rcnstrctn$misc$taxa_notes`.

### 3.3.3 The alternative modern calibration dataset (`distributions`)

Users that prefer fitting proxy-climate relationships from their personal calibration data instead of the proposed *gbif4crest* dataset should prepare a `distributions` dataset following the specific structure presented in Table 4. The first two columns





should contain the species names (or unique identifiers) and the corresponding proxy name, respectively. If more than one
species correspond to one taxon, the PDFs will be fitted in two steps, as explained in section 2.1. The following two columns
contain the coordinates of the species occurrence data. Finally, the last columns contain the climate values to be reconstructed.
An optional column called `weight` can be added to `distributions` in fifth position (*i.e.* between the coordinates and the
climate variables) if one wants to weight the different observations. For example, the (relative) abundance of taxa observed from
modern proxy samples can be used when fitting the PDFs to give more importance to the observations where that abundance
is highest.

**Table 4.** Template for the *distributions* data frame. The weights column, here indicated with a '*', is optional and can be omitted or its
values all set to 1 to assign the same weight to each observation. The number of rows of the table should correspond to the number of unique
occurrences available.

| Species name | Taxon Name | Longitude | Latitude | Weight* | clim_1 | ... | clim_n |
|:---:|:---:|:---:|:---:|:---:|:---:|:---:|:---:|
| *Stoebe plumosa* | *Stoebe*-type | 18.875 | -34.375 | 20 | 15.8 | ... | 711 |
| *Elytropappus rhinocerotis* | *Stoebe*-type | 18.375 | -33.625 | 32 | 16.9 | ... | 477 |
| ... | ... | ... | ... | ... | ... | ... | ... |
| *Elaeis guineensis* | *Elaeis guineensis* | -4.375 | 10.875 | 4 | 27.4 | ... | 1020 |

### 3.3.4 The `climate_space` data frame

This data frame is only necessary if the users use a personal calibration dataset instead of the *gbif4crest* dataset. It will enable
the use of the climate space weighting (Section 2.1.1). Its structure is straightforward, with the first two columns containing
longitudes and latitudes and the subsequent columns the climate variables to reconstruct. The spatial resolution and the ordering
of the climate variables should be identical to the `distributions` table (Table 4).

### 3.3.5 The `selectedTaxa` data frame

The last data frame that may be used to inform the reconstruction is a data frame of ones and zeros called `selectedTaxa`.
This data frame has as many rows and columns as there are taxa and climate variables, respectively. Each entry should
be either 1 or 0 and is used to indicate if the taxon should be used to reconstruct the climate variable (value = 1) or not
(value = 0). If it is not provided, a default `selectedTaxa` data frame with all entries set to 1 is added to the `crestObj`.
Users can modify this information at any point using the `includeTaxa()` and `excludeTaxa()` built-in functions. The
`crest.get_modern_data()` function also modifies this data frame by setting the value to -1 when the `PSE` classification
failed for a taxon or when the amount of data in the study area is insufficient to fit a reliable PDF (see the parameters description
in section 4.2).



### 3.4 Package dependencies

The `crestr` package is built in R (R Core Team, 2020) using the `devtools` package (Wickham et al., 2020). `crestr` depends on numerous packages, including: `clipr` (Lincoln, 2020), `DBI` (R Special Interest Group on Databases (R-SIG-DB) et al., 2021), `openxlsx` (Schauberger and Walker, 2020), `pals` (Wright, 2021), `plot3D` (Soetaert, 2021), `plyr` (Wickham, 2011), `raster` (Hijmans, 2021), `rgdal` (Bivand et al., 2021), `rgeos` (Bivand and Rundel, 2020), `RPostgres` (Wickham et al., 2021), `scales` (Wickham and Seidel, 2020), `sp` (Pebesma and Bivand, 2005, Bivand et al. (2013)), `stringr` (Wick-
ham, 2019) and `viridis` (Garnier et al., 2021). The dedicated documentation, tutorials and application examples found at https://mchevalier2.github.io/crestr were generated and formatted by the package `pkgdown` (Wickham and Hesselberth, 2020).

### 4 Step-by-step user guide for **`crestr`**

#### 4.1 Example application: Pollen-based Mean Annual Temperature reconstructions from marine core MD96-2048

To illustrate the different ways of using `crestr` and its graphical diagnostic tools, I use pollen data that were recently analysed
with the original CREST software to reconstruct mean annual temperature (MAT) from marine core MD96-2048. The core was retrieved off the coast of South Africa and Mozambique near the mouth of the Limpopo River, and the terrestrial sediments are expected to come from the entire catchment of the Limpopo River, as well as the smaller local river catchments (Dupont et al. (2019), Dupont et al. (2011), Castañeda et al. (2016)). The MAT reconstruction is based on 181 fossil pollen samples and more than 150 terrestrial pollen taxa and covers the last 790,000 years. As the catchment of these marine sediments is large,
an extensive calibration dataset covering all vegetation zones from tropical Africa to the temperate southwestern tip of South Africa was designed to prevent any artificial reduction in the possible range of variability of the reconstruction. The glacial-interglacial trends and amplitude of the MAT reconstructions were validated by comparing them with regional temperature records and other global indicators of glacial-interglacial temperature variability (*e.g.* Antarctic temperature or global sea-level curves, see Chevalier et al. (2021) for more details). Here, I reproduce this reconstruction using the original parameterisation
of the CREST algorithm to showcase how it can easily be replicated in a few lines of code with the `crestr` package. Due to the update of the climate data of the calibration dataset described in section 2.2, marginal differences in the order of tenths of Celsius degrees can, however, be observed between the original publication and this reproduction. All the necessary datasets and R code are available as supplementary material.

#### 4.2 Formatting the calibration data in a **`crestObj`**

As the *gbif4crest* dataset was used to fit the PDFs, using the function called `crest.get_modern_data()` is required to extract the calibration data from the new cloud-based *gbif4crest* calibration database. All the parameters of the function were defined by the characteristics of the proxy (pollen), the climate to reconstruct (MAT / bio1) and the definition of the study area (East and southern Africa) and are reproduced here. In addition, the three input files (*i.e.* `df`, `PSE` and `selectedTaxa`) required to realise this reconstruction (see section 3.3) are reproduced from the published dataset (Chevalier et al., 2020a). The



following points describe the different parameters of the `crest.get_modern_data()` and how they relate to the data extraction and modelling.

– The parameter `taxaType` is used to choose the type of proxy used and takes a value between 1 to 6 for plants (*i.e.* pollen and plant macroremains), beetles, chironomids, foraminifers, diatoms and rodents, respectively.

– The name(s) or code(s) of the `climate` variables to study should be provided here (see Tables 1 and 2 or use the
function `accClimateVariables()` for a list of accepted values). Here, 'bio1' means mean annual temperature.

– Geographical parameters can be provided to create a regional subset of the *gbif4crest* dataset that is adapted to the studied data. These can include minimum and maximum longitude and latitude (`xmn`, `xmx`, `ymn`, `ymx`), continent, ocean or country names (see `accCountryNames()` and `accBasinNames()` for a list of accepted names), and also some ecological classifiers, such as realms, biomes or ecoregions (see `accRealmNames()` for a list of accepted names).
Only the occurrences that respect all the specified constraints will be returned.

– To estimate reliable PDFs, it is recommended to use at least 20 distinct occurrences for each species, but different values can be specified with the `minGridCell` parameter.

– Optional information about the site, such as a name and coordinates, can be provided and, where possible, this information will be represented on the different graphical diagnostic tools created by `crestr` (*e.g.* the location of the record
can be added to the maps if the coordinates are provided).

```
rcnstrctn <- crest.get_modern_data(
  df = MD96_2048,
  pse = PSE,
  selectedTaxa = selectedTaxa,
  taxaType = 1,
  climate = c('bio1'),
  xmn = NA, xmx = NA,
  ymn = -35, ymx = NA,
  continents = 'Africa',
  countries = c('South Africa', 'Kenya',
    'Lesotho', 'eSwatini', 'Botswana',
    'Mozambique', 'Zimbabwe', 'Zambia',
    'Malawi', 'Tanzania', 'Namibia',
    'Uganda', 'Rwanda', 'Burundi'),
  realms = NA,
  biomes = NA,
```





```
      ecoregions = NA,
      minGridCells = 20,
      site_info = c(34.0167, -26.1667),
site_name = 'MD96-2048',
      dbname = "gbif4crest_02",
      verbose = TRUE
   )
```

The `crest.get_modern_data()` function reads all the data and parameters, extracts the data from the cloud-based

*gbif4crest* database, processes the distribution data and returns everything as a structured `crestObj` — here called `rcnstrctn`; Fig. 5 — that will be read and modified by all subsequent functions. Alternatively, the function `crest.set_modern_data()` could be called instead of `crest.get_modern_data()` to use personal calibration data instead of the *gbif4crest* database. If the calibration data needed for this study were available as a `distributions` and `climate_space` data frames (Section 3.3), similar results would be obtained using the following command:

```
rcnstrctn <- crest.set_modern_data(
      df = MD96_2048,
      distributions = distributions,
      climate_space = climate_space,
      selectedTaxa = selectedTaxa,
climate = c('bio1'),
      weight = FALSE,
      minGridCells = 20,
      site_info = c(34.0167, -26.1667),
      site_name = 'MD96-2048',
verbose = TRUE
   )
```

### 4.3 Estimating the climate responses (the PDFs)

The probabilistic proxy-climate relationships, *i.e.* the PDFs, are estimated from the presence data using the `crest.calibrate()` function. As described in section 2.1.1, the different parameters should be carefully considered to produce reliable PDFs. These
include specifying:

- the shape of the species PDFs, which should be either 'normal' or 'lognormal', depending on the variable to reconstruct.

- the width of the climate bins (`bin_width`) expressed in the variables' units (*e.g.* 2°C or 50 mm) if the PDFs have corrected for an heterogenous climate space (`climateSpaceWeighting` set to `TRUE`). Dividing the total climate space in 15-25 bins often leads to good results, but other values are possible.





– the number of intervals required to divide the studied climate range and fit the PDFs `npoints`. This will ultimately
define the climate resolution of the reconstructions. `crestr` runs faster with a lower resolution but this can lead to a
problem of aliasing of the reconstructions.

– set `geoWeighting` to `TRUE` if the species PDFs of the different composing species should be weighted according to
the square-root of the extent of their modern distribution.

```
rcnstrctn <- crest.calibrate(
       rcnstrctn,
       shape = c('normal'),
       climateSpaceWeighting = TRUE,
       bin_width = c(2),
npoints = 500,
       geoWeighting = TRUE,
       verbose = TRUE
     )
```

### 4.4 Assessing the coherency of the climate space and climate responses

In every study involving estimating relationships between biological entities and environmental parameters, the first step is
always to ensure that the defined calibration dataset is as coherent as possible. This includes ensuring that 1) all the important
taxa are present, and their distribution is not truncated, 2) the climate values to reconstruct are likely to be in the study area (the
reconstructions are bounded by the lowest and highest values observed in the modern climate space) and 3) there is no large
sampling or representativity bias (*e.g.* along country borders due to different sampling efforts). Ideally, the climate sampling
should be as homogeneous as possible, even if the extreme climate values will always be under-represented compared to
the median ones. Equally importantly, the occurrence data across that space should be similarly distributed to ensure a proper
sampling of the climate space. However, deviations from a theoretical one-to-one (or at least proportional) equivalence between
climate and occurrence data abundance are not necessarily a bad characteristic. In our case study, the spatial variability represent
true patterns in regional species diversity with the presence of several biodiversity hotspots across eastern and southern Africa
(Myers et al., 2000). All these elements should be checked and accounted for while designing the final calibration dataset.

The 'climateSpace' graphical diagnostic tool was thus designed for a rapid assessment of all these characteristics (Fig.
6). This diagnostic figure is also very important to identify potential local or global correlations between different climate
variables and assess the risks of confounding variables (Juggins (2013), Chevalier et al. (2020b)). Any change to any of the
parameters related to the definition of the climate space (study area, climate variables to reconstruct) will require to re-run
`crest.get_modern_data()` or `crest.set_modern_data()` with updated parameters and/or data.

```
plot_climateSpace(rcnstrctn, save=TRUE,
  filename='Figure 6.png',
```





```
     as.png=TRUE, png.res=600,
     width=6.9, height=4.4,
y0=0.4,
     add_modern=TRUE
)
```

With the study area and climate space defined, the next step is to search for taxa that show specific relationships with climate. While all species eventually respond to all climate variables, within a given region they can be more sensitive to one over another. The low taxonomic resolution of some fossil proxies, such as pollen data, can also mask strong species-climate relationships. Looking at each individual climate response(s) and assessing their significance within the boundary conditions of the study is thus critical. The 'taxaCharacteristics' diagnostic plot was designed for this task (Fig. 7). One summary plot can be generated for each taxon, where the geographical distributions and climate responses can be assessed and inter-compared.

As illustrated in Fig. 7, Ericaceae is preferentially observed in the colder environments of the study area, its higher percentages occur during glacial periods, and a coherent response of all its composing species can be observed despite a high diversity (141 species with at least 20 unique occurrences across the study area). All these elements indicate that Ericaceae can be considered as an indicator of colder environmental conditions in eastern and southern Africa. Sensitivities to other variables can also be expected but are not considered in this study. Making such sensitivity inferences can be key to define a list of taxa that can be used to reconstruct temperature (Chevalier and Chase (2015), Chevalier et al. (2021)), but also to support qualitative interpretations of palaeoecological datasets (Chevalier et al. (in press), Quick et al. (2021)).

```
plot_taxaCharacteristics(rcnstrctn,
    taxanames='Ericaceae', save = TRUE,
    filename = 'Figure 7.png',
    as.png=TRUE, png.res=600,
width=6.9, height=8.13,
    add_modern=TRUE
)
```

Another useful diagnostic plot is the 'violinPDFs' (Fig. 8) that represents the PDFs of a selection of taxa on the same scale, which helps comparing the different responses. All the violins have the same area (all the probabilities sum to 1) and the taxa are ranked by increasing value of their temperature optima (*i.e.* the temperature corresponding to the peak of the PDF). However, due to the possible multimodality of the PDFs and differences in tolerance ranges, this ranking does not mean that a taxon on the left always represents colder conditions than a taxon on the right. This is illustrated in many ways on Fig. 8 with, for example, *Cassia*-type that is estimated to experience warmer conditions than *Coffea*-type ~61% of the time (based on 100,000 random draws from their respective PDFs) despite having a 'colder' climate optimum, or *Diospyros* that can tolerate much warmer conditions than most taxa with warmer optima. This type of representation can be particularly helpful to have objective interpretations of ecological changes from pollen diagrams (Chevalier et al., in press, Quick et al. (2021)).

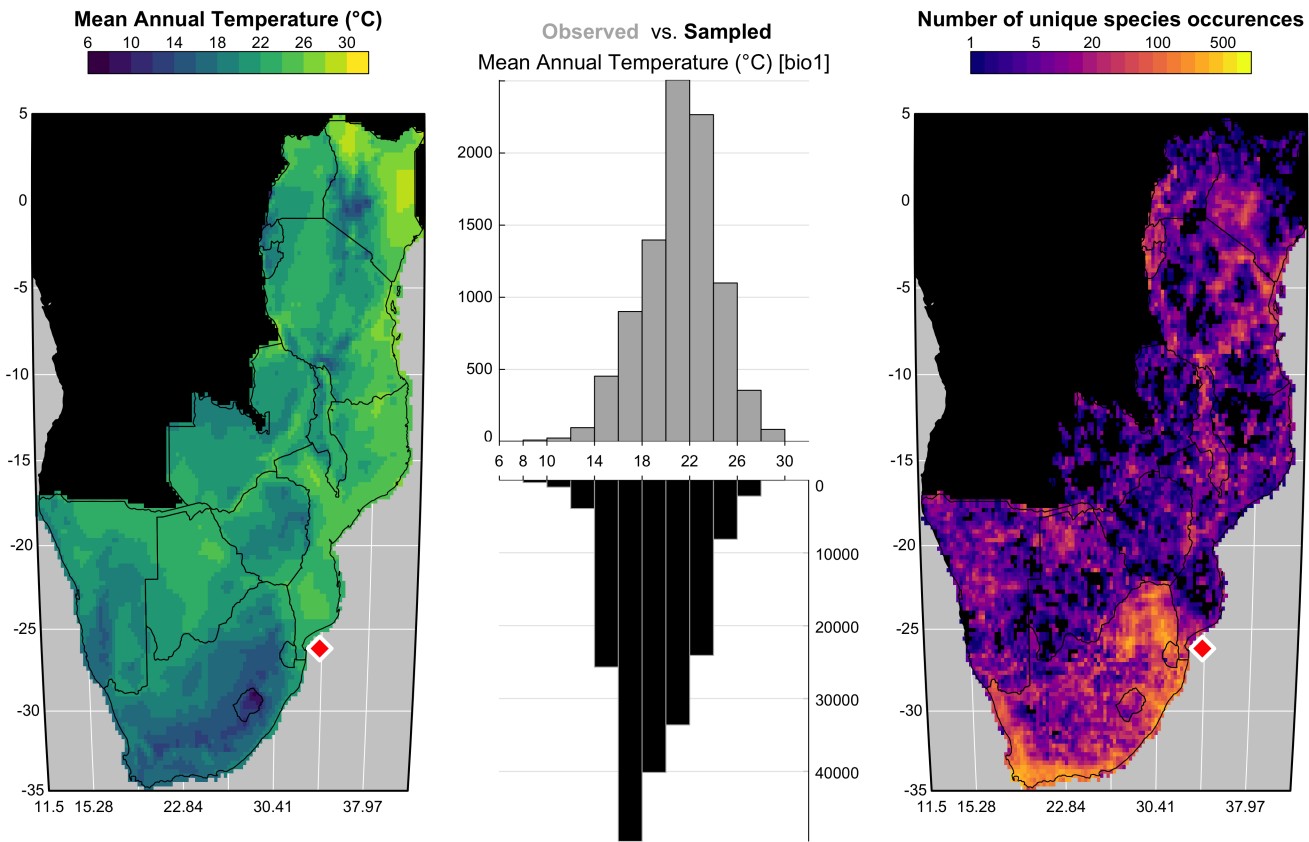

**Figure 6.** 'climateSpace' graphical diagnostic tool to evaluate the calibration dataset. The map on the right represents the density of unique species occurrences in each grid cell, here highlighting a certain bias towards South Africa. The lower abundance of plant data available from Angola and the Democratic Republic of Congo (data not shown) is the reason why these two countries were excluded from the study area. The map on the left represents the studied climate variable (MAT) across the study area. The double histogram in the middle represents the distribution of MAT (in grey, top), while the black histogram (bottom) represents how this distribution is sampled by the calibration data. Differences between the two histograms can be used to identify biases in the calibration dataset. Here, the small shift of the black histograms towards colder values (towards the left) is another way of seeing that more data are available from South Africa than other countries and reflects the regional patterns of biodiversity. If more variables had been selected in this study, additional rows would be added to the figure with a similar climate map and histograms and a scatterplot of the climate variables to highlight potential local or regional modern correlations.



**Figure 7.** 'taxaCharacteristics' graphical diagnostic tool to assess the sensitivity of taxa to climate. In the top row, the map represents the density of unique species occurrences per grid cell and the time series represents the variability of the taxon against time or depth. The bottom row is specific to each selected variable (only one row in this case), where the map represents the geographical distribution of the taxon (in white) against the climate background, the histogram represents the climate space (in colour) and how this climate is sampled by the taxon (in black) and finally the right plot represents the climate response of the taxon (in black) as well as the response of all the composing species (in grey). This diagram shows that Ericaceae is preferentially associated with colder temperatures in the study area. The red diamond indicates the location of the studied record.



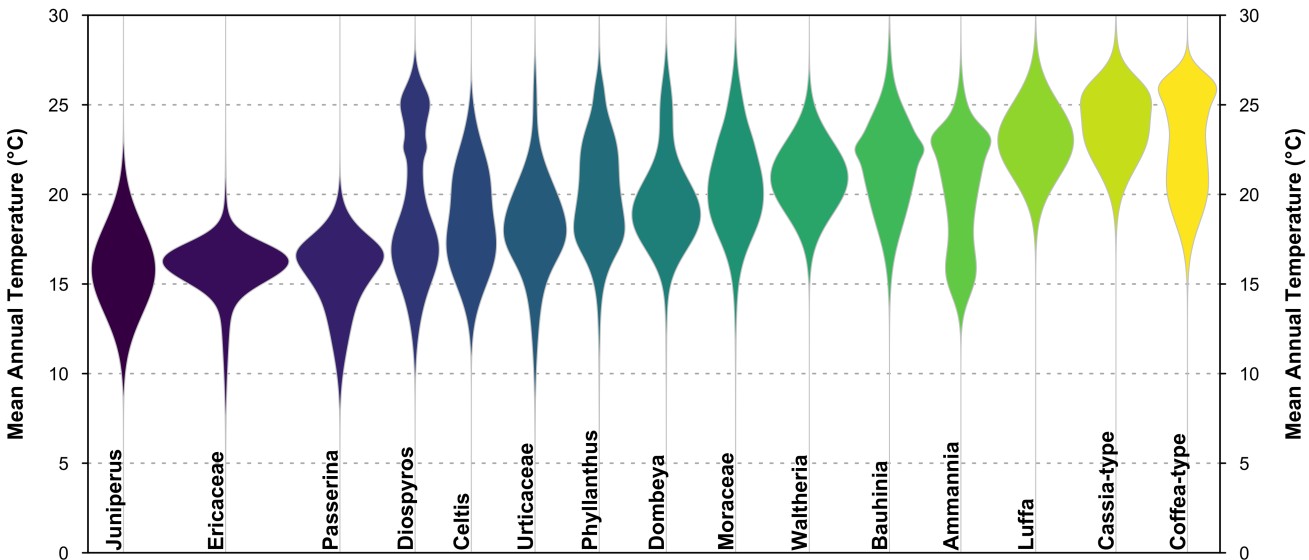

**Figure 8.** 'violinPDFs' graphical diagnostic tool to represent the PDFs of various taxa (here only a subset of 15). The taxa are sorted and colour coded by their temperature optima (i.e. the temperature corresponding to the peak of their PDFs).

```
tax <- sample(rcnstrctn$input$taxa.name,
              15)
plot_violinPDFs(rcnstrctn,
taxanames = tax,
    save = TRUE,
    filename = 'Figure 8.png',
    as.png=TRUE, png.res=600,
    width = 6.9, height = 3,
ylim=c(0,30)
)
```

## 4.5  Reconstructing climate

Along with the `df` data frame provided to the `crest.get_modern_data()` or `crest.set_modern_data()` func-
tions, a set of reconstruction parameters have to be chosen to combine PDFs and estimate climate parameters. The `selectedTaxa`
data frame that is stored in the `crestObj` (see Fig. 5) defines the taxa that will be used to reconstruct each climate variable
(by default, all taxa are included if sufficient data are available to fit a PDF). This selection can be modified by using the
`includeTaxa()` and `excludeTaxa()` functions. Here for instance, both Aizoaceae and Chenopodiaceae/Amaranthaceae
were excluded because they are not primarily sensitive to temperature in southern Africa.





```
rcnstrctn <- excludeTaxa(rcnstrctn,
    taxa=c('Aizoaceae',
        'Chenopodiaceae/Amaranthaceae'),
    climate=c('bio1')
)
```

Climate reconstructions are performed by the `crest.reconstruct()` function. A minimum 'presence threshold' below
which the taxa will be always considered absent can also be provided to reduce the noise of the fossil dataset (*e.g.* pollen per-
centages lower than 1 or 2% are commonly excluded from reconstructions, Chevalier et al. (2020b)). If `presenceThreshold`
is set to zero as is the case here, all the strictly positive pollen percentages are considered as true presences and used accordingly
to reconstruct MAT. To weight the taxa as described in Eq. 7, four options are available in `crestr`:

- The data can be converted to presence/absence with all the values above and below `presenceThreshold` being
  changed to ones and zeros, respectively. This option is recommended for data such as macrofossils for which relative
  abundances cannot be reliably estimated.

- The data can be converted to percentages to weight the taxa according to their relative abundance. This option is recom-
  mended for data where reliable and direct proportions can be estimated.

- The data can be normalised following the method proposed by Chevalier et al. (2014) and described here by Eq. 8. This
  option is recommended for pollen data for instance.

- The data can be directly weighted by the values provided in `df`, which implies that users can define their own specific
  weighting strategy.

```
rcnstrctn <- crest.reconstruct(
    rcnstrctn,
    presenceThreshold = 0,
    taxWeight = "normalisation",
    verbose = TRUE
)
```

## 4.6 Analysing and understanding the reconstruction(s)

Characterising the most important factors underlying a reconstruction is often complex. In this section, I present three different
graphical diagnostic tools that provide different perspectives on the reconstructed data and help opening the statistical black
box in a visual way. First, the full probabilistic breadth of the reconstructions can be represented by using the standard R
`plot()` function, which has been adapted to plot data stored in a `crestObj` (Fig. 9). While the probabilistic representation
of the data should be preferred because it represents all the information available, a simpler version of this plot with the climate





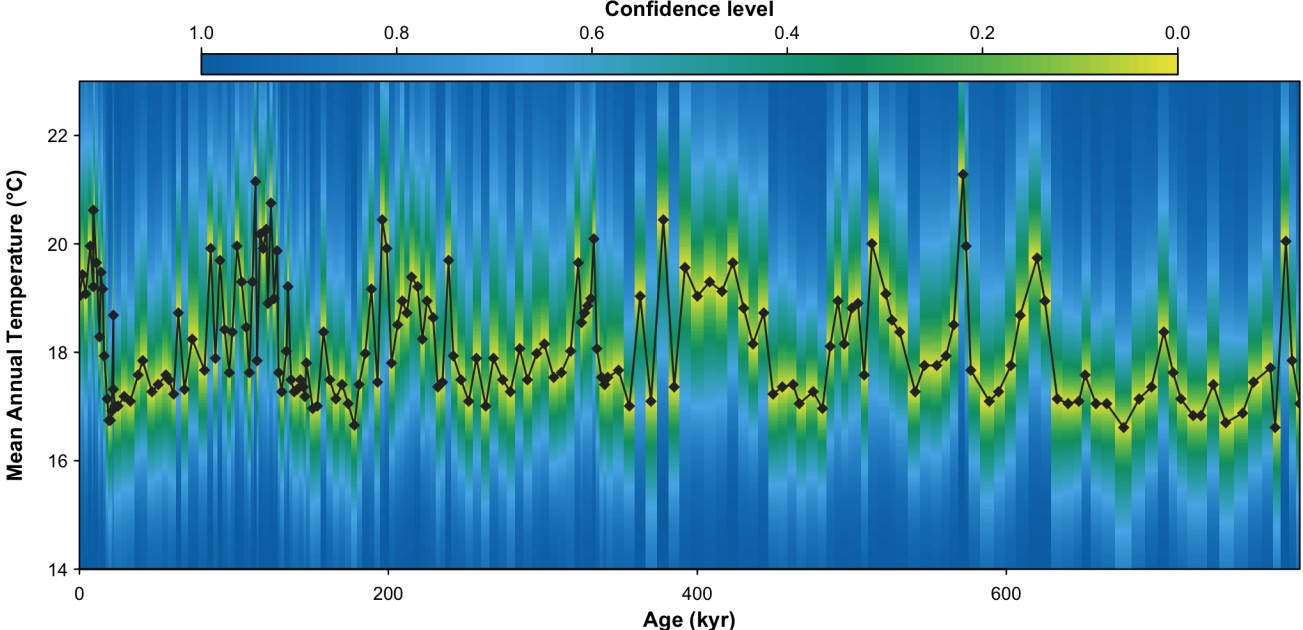

**Figure 9.** MAT probabilistic reconstructions for marine core MD96-2048. The yellow-green-blue colour gradient represents the uncertainties associated with each sample. This reconstruction is identical to the reconstruction presented in Chevalier et al. (2021) and the reader is referred to this publication for an in-depth validation and discussion of these results.

optima and more common uncertainty ranges represented as colour bands can be obtained by specifying `simplify=TRUE` (not shown).

```
plot(rcnstrctn,
  filename='Figure 9.png',
  save=TRUE,
  as.png=TRUE, png.res=600,
  width=6.9, height=3.54,
  ylim=c(14,23), uncertainties=1,
  simplify=FALSE,
  col=plot3D::gg2.col(200)[1:100],
  pt.cex=0.8, pt.lwd=1, pt.col='#2c2c2c'
)
```

The `plot_combinedPDFs()` function can then be used to identify the taxa that are driving the reconstruction by focusing on the sample level (Fig. 10). In this plot, the PDFs of all the taxa present in the sample and selected to reconstruct the climate




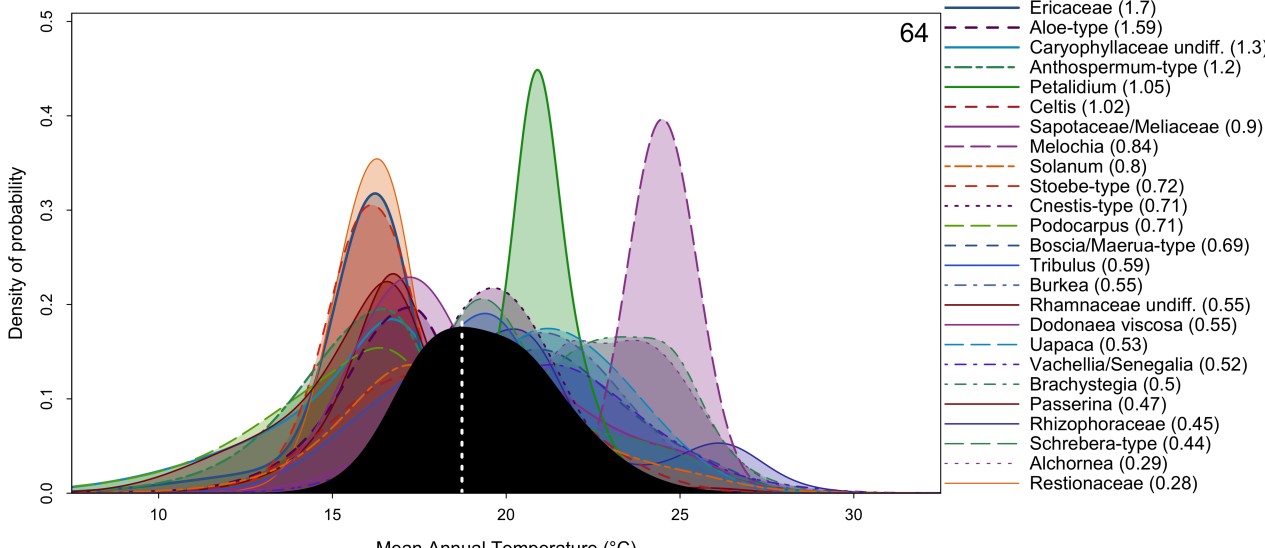

**Figure 10.** 'combinedPDFs graphical diagnostic tool that shows the combination of the PDFs of all the taxa recorded in a sample (in colour). The thickness of the lines is proportional to the weight of the taxa in the sample (absolute value indicated next to each taxon name). The black curve represents the posterior MAT reconstruction, from which a 'best' climate estimate can be estimated from the maximum of the curve and uncertainties derived by calculating the area under the curve.

variable are represented along with the reconstruction. This type of plot can help identify if there is a particular PDF that is at
odds with the general assemblage, which can be indicative of a confounding factor. It is also useful to visualise the full spread of the uncertainties and, by extension, highlight that reconstructions can be multimodal. The presence of multimodality can be the underlying cause of apparent noise in the reconstructions because minor changes in the taxa composition or percentages can force the system to oscillate between two maxima and thus 'appear' noisy, even if the background rate of change is minor. This can also be seen from the reconstruction plot if the full uncertainties are represented (`simplify=FALSE`).

```
plot_combinedPDFs(rcnstrctn,
        samples=3, only.present=TRUE,
        only.selected=TRUE, save = TRUE,
        filename = 'figure 10.png',
        as.png=TRUE, png.res=600,
width=6.9, height=3,
        xlim=c(7.5,32.5)
)
```

Finally, a standard post-processing analysis of CREST reconstructions is a form of leave-one-out (LOO) analysis that is done with the `loo()` function. In the CREST context, a LOO analysis consists in repeatedly 'unselecting' one taxon at a time,





re-running the reconstruction without that taxon and measuring the associated reconstruction anomalies. A detailed analysis of the results can contribute to a deeper insight regarding which taxa are the most important in driving the reconstructed climate signal. Taxa that exhibit large LOO values indicate that they have a large influence on the reconstruction. However, it does not necessarily mean that they are strong climate indicators. Large LOO values can arise when the PDFs are biased by unaccounted factors and are, as a result, at odds with the rest of the PDFs. Such factors can, for instance, include an incomplete estimate of the climate response, which induces a bias or shift of the climate preferences, or a sensitivity of the taxon to other climatic (*e.g.* aridity instead of temperature) or non-climatic (*e.g.* edaphic conditions) factors. In all cases, it is usually preferable to exclude such taxa from the reconstruction.

```
rcnstrctn <- loo(rcnstrctn,
  verbose = TRUE
)
```

The LOO analysis is a powerful tool to understand the taxa that primarily drive the reconstruction and the LOO results can be represented as a common stratigraphic diagram, where each row represents the effect of removing a taxon from the reconstruction (Fig. 11). For example, the expected net cold effect of including Ericaceae in the MAT reconstructions from MD96-2048 pollen record is immediately visible on Fig. 11. Similarly, the effect of removing either Ericaceae or Caryophyllaceae undiff. from the sample dated at 64 kyr (represented on Fig. 10) is quite strong with a net warming of 1.19 and 0.44°C, respectively. This type of plot can thus be used for both global and sample-specific inferences about the drivers of the reconstruction. Depending on the vegetation composition, some taxa can even sometimes be categorised as cold indicators when the assemblage represents warm conditions, or warm indicators when the assemblage represents cold conditions (*e.g. Hypoestes/Dicliptera*-type on Fig. 11).

```
tax <- rcnstrctn$inputs$taxa.name[6:25]
plot_loo(rcnstrctn,
    taxanames=tax,
    xlim=c(0, 340), save = TRUE,
    filename = 'Figure 11.png',
    as.png=TRUE, png.res=600,
    width=3.5, height=8,
    bar_width=3, col_neg='coral3',
    col_pos='darkcyan'
)
```

## 4.7  A wrapper function

To simplify the use of the package, the three stages of the reconstruction process – data acquisition (`crest.get_modern_data()` or `crest.set_modern_data()` if the *gbif4crest* dataset is not used [not shown]), calibration (`crest.calibrate()`)



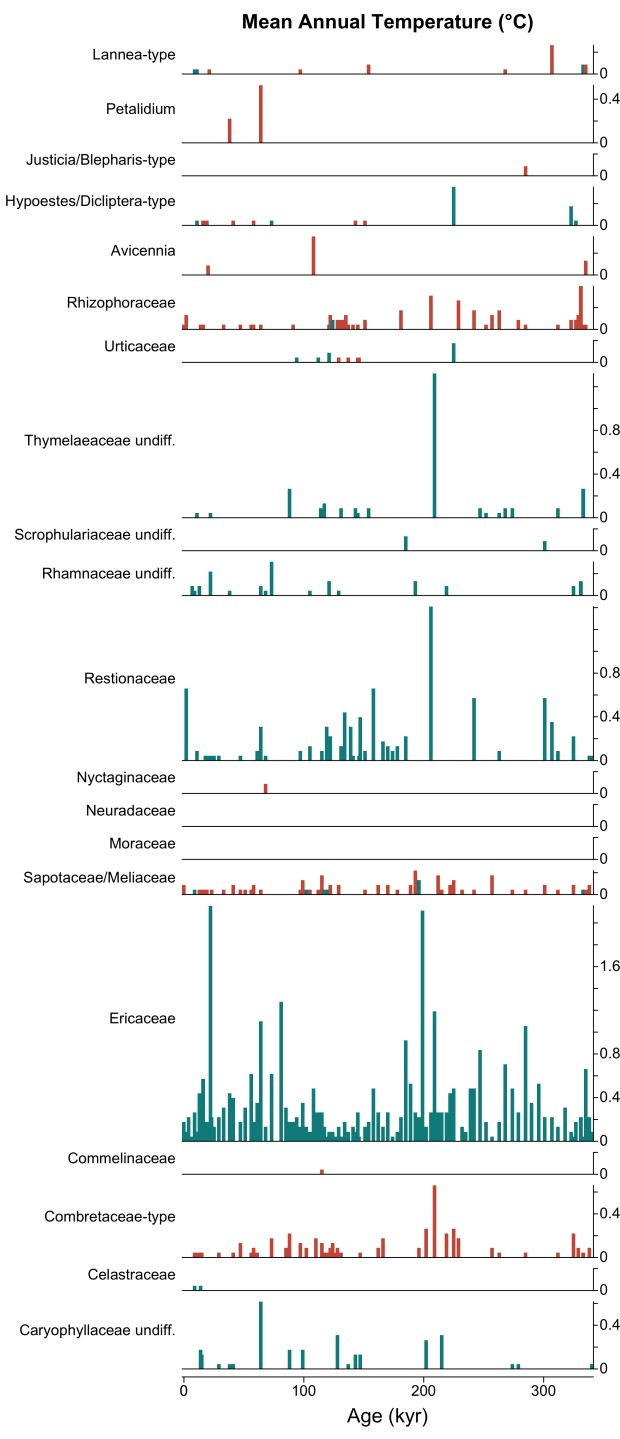

**Figure 11.** Leave-one-out (LOO) graphical diagnostic tool to illustrate the influence of different taxa on the reconstructions. Here, the results are only showed for a subset of the taxa observed in marine core MD96-2048 (only 20 out of the 171 available taxa are represented). The height of each bar represents the net effect (in °C) of removing the taxon from the reconstruction. The colours indicate the sign of the anomaly with blue (cf. Ericaceae) and red (cf. Combretaceae-type) taxa being cold and warm indicators in this setting, respectively.





and reconstruction (`crest.reconstruct()`) – can be called in one line of code using the wrapper function `crest()`. This function takes the same parameters described for the 'step-by-step' functions with the same default values and may be
preferred, for instance, when reconstructing several records at the same time.

## 4.8 Exporting the reconstructions

All the data stored in the `crestObj` can be easily exported from the R environment as spreadsheets and RData files using either `export_pdfs()` to save the climate responses of the studied taxa, or `export()` to save the reconstructions and many associated data in a publishable format. The latter also saves the `crestObj` as an RData file for easy reloading and/or
sharing of the data.

```
export(rcnstrctn,
  loc='path/to/folder',
  fullPosterior = TRUE,
  loo = TRUE,
weights = TRUE,
  pdfs = TRUE
)
```

## 4.9 Citing building elements

Finally, all the reconstructions derived from `crestr` are built on numerous, independent research efforts, including data
compilations, modelling projects, statistical developments and/or software engineering. To support the long-term development and/or support of all these building elements, it is crucial to always acknowledge all of them, even if their processing is mostly invisible to the users. As such, the list of references that must be cited for each use of `crestr` is automatically included in the summary tab of the spreadsheet generated by the `export()` function. In addition, the citation information can also be directly obtained from R using `cite_crest(rcnstrctn)`. The function looks at the type of data that were used (*e.g.* the subset of
the GBIF data were used, which climate variables) and returns a corresponding list of references. For example, a simple way of crediting all the contributors for the MAT reconstruction from marine core MD96-2048 presented here could be the following:
*"To create this MAT reconstruction from the pollen record from marine core MD96-2048 (Dupont et al. (2019)), we employed the CREST method (Chevalier et al. (2014), Chevalier (2019)). The PDFs were estimated by combining the MAT field of Fick and Hijmans (2017) and the plant occurrence data of GBIF (GBIF (2020m), GBIF (2020k), GBIF (2020g)). The numerical*
*analyses were realised with the crestr R package v1.0.0 (this reference)."*

## 5   Perspectives and conclusion

CREST is a statistical method designed 1) to model probabilistic relationships between climate proxies and climate from modern observations and 2) to use these relationships to reconstruct past climate quantitatively. Initially, its use was enabled



by a python point-and-click graphical user interface to favour accessibility to a diverse spectrum of palaeoclimatologists,
palaeoceanographers and palaeoecologists, including those with limited coding experience (Chevalier et al., 2014). However,
such accessibility limited flexibility and how the method could be most effectively employed. Therefore, this article introduces
an open-source R package to use CREST in a programmatic environment. The benefits of this transition are significant and
include scriptability (*i.e.* the possibility of analysing many records automatically and sequentially), reproducibility (*i.e.* the
capacity to reproduce an analysis) and better inter-operability (*i.e.* R packages are compatible with all computer systems).
However, maintaining the highest level of accessibility remained at the core of the development process and is illustrated in
the final product by the small number of functions necessary to run a complete analysis and by the suite of detailed graphical
diagnostic figures.

The package is aimed at all researchers that are interested in using CREST to analyse palaeoecological datasets. In addition,
its broad applicability will allow taking advantage of the recent growth of curated, open-access fossil datasets that created
unprecedented opportunities to reconstruct climate from a wide range of proxies, particularly in regions where they are urgently
needed. This package will also contribute to the current transition from single-site to multi-site studies that is necessary to
better understand past climate dynamics. However, it is essential to remember that running such techniques on several datasets
should be done carefully, as many factors can impact the reconstruction process. Calibration datasets and reconstructions
should always be assessed, possibly against independent evidence when they are available, even if there is, unfortunately, no
single way to validate a reconstruction (but see Chevalier et al. (2020b) for some discussions on the generic principles).

Over time, the `crestr` package will be enriched with new functionalities to facilitate reconstruction validation. The on-
line documentation will also be updated with diverse examples and tutorials based on real applications and assessments
(https://mchevalier2.github.io/crestr/). Finally, bug reports, feedback, and suggestions for newer functionalities and graph-
ical diagnostic tools are encouraged and can be transmitted to the author directly or through GitHub's bug report portal
(https://github.com/mchevalier2/crestr/issues).

*Code and data availability.* The *crestr* package is currently accessible from GitHub (https://github.com/mchevalier2/crestr), where I also
welcome feedback and strongly encourage contributions and enhancements via the issue tracker (https://github.com/mchevalier2/crestr/issues).
The package has also been submitted to CRAN. The data used for the example application (pollen record from marine core MD96-2048 and
associated temperature reconstructions) can be accessed from https://doi.pangaea.de/10.1594/PANGAEA.915923.

*Competing interests.* The author declare no competing interests.

*Acknowledgements.* This research has been supported by the Swiss National Science Foundation (grant no. CRSK-2_195875), with addi-
tional support from the University of Lausanne. I thank Brian Chase, Yoshi Maezumi, Lynne Quick and, last but not least, Mine Altinli
for their invaluable comments and suggestions on early versions of the manuscript. NOAA High Resolution SST data provided by the





NOAA/OAR/ESRL PSL, Boulder, Colorado, USA, from their Web site at https://psl.noaa.gov/data/gridded/data.noaa.oisst.v2.highres.html.

610    And finally, I would also like to address a special thanks to the rticles contributors for their Climate of the Past Rmarkdown template!





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
