# Peer review of "crestr An R package to perform probabilistic climate reconstructions from palaeoecological datasets"

_Climate of the Past, 2021_

## Referee Comment (RC1)

**General Comments**

I very much appreciate the effort the author has made to share his code or should I say software with the community. Without having tested the R package by myself, I am confident to say that it constitutes a very valuable contribution for the community and that many researchers can benefit from it. The package is well described in the manuscript and it seems that the author achieved his goal of keeping the hurdles for application as low as possible while at the same time providing a reasonable degree of flexibility. From a user's perspective package's automated query of the databases is the optimal solution. Also, the diagnostic tools seem to be very helpful for assessing the quality of past climates reconstructions.

Overall, the manuscript conveys the functioning of the R package in a decent way. However, there are several inaccuracies in the language and sometimes it seems the paper has been written with a lack of thoroughness. For example, there are plots without axis labels. The figure and table captions are generally very short. I would suggest to give all of them a careful review. In principle, the figures as well as the tables should be understandable from their captions and the reader should not have to search in the main text for the relevant pieces of information. Additionally, formulas appear to be detached from the text and the definitions of the entities that appear in formulas are sometimes given way before the formula is presented. In general, any formula should be part of a sentence. The structure of the manuscript is well chosen. The introductory part could, however, do a better job in setting the scene. It took me a long time to understand, that we are dealing with proxy data on the one hand, with the abundance of a certain proxy being given in continuous units with respect to a depth (or time) axis and on the other hand with binary presence data from modern observations spread over the entire globe, which if supplement by the corresponding climate data serves for the calibration of the 'climate response function'. Maybe, this is clear to everybody working with palaeoecological proxy data, to me it was not. Another point that confused me while reading, is that in Section 2. there is a clear two-level structure comprised of taxa and the species of each taxon. Correspondingly, the way how PDF's of the different species are combined to the one of a taxon is introduced. Later in the manuscript, more levels added to this structure like families and genus. I do neither understand how PDFs are combined on these intermediate levels nor what is actually measured in the proxy data? I thought, one could either measure the abundance a taxon or of specific species in a sediment core? Again, maybe this is obvious to people who are used to work with palaeoecological proxy data.

My greatest concern relates to the mathematical presentation of how the package actually computes the past climate's reconstruction and with this to Section 2. of the manuscript. This part of the manuscript is definitely lacking the required accuracy and is in some sense misleading. The author calls the function $\text{PDF}_{\text{reconstrctn}}(c, z)$ a *posterior climate reconstruction* (compare l.110). This suggests, that a full Bayesian approach has been pursued. Instead, after having careful studied Section 2. I understand that the climate reconstruction presented by the author is in fact a maximum likelihood estimate of the past climate and that the uncertainties presented in Fig.9 correspond to the

percentiles of the corresponding likelihood function. I would really like to encourage the author to implement a full Bayesian approach in the R package, if not in this version, then in an updated version of the software.

Despite this major issue, I would like to emphasize that this manuscript is only the description of what has actually been the main part of the work, and that is the R package. I consider this package as a very valuable contribution and this paper is a decent description of the package and thus definitely merits publication.

**Detailed Comments on Section 2.**

Section 2. starts with a derivation of what is called the 'climate response' or 'PDF' of a given species from the combined modern presence observations and climate data. First, I would recommend to give the object an unambiguous name. Since the term 'PDF' simply describes a certain class of functions, I strongly prefer 'climate response function' which could then be associated with some greek letter like $\chi$ or $\rho$ for example.

In fact, the starting point for the derivation of the climate response function is the joint probability distribution of two random variables, namely the climate $C$ and the presence of a species $S$ which belongs to some taxon $t$. While $C$ is a continuous random variable, the random variable $S$ is in fact a binary random variable that indicates the presence (S=1) or the absence of the species (S=0). Let the joint distribution be denoted as

$$\rho_{S,C}(s,c). \tag{1}$$

Given spatially extended observation data with $N_{\text{obs}}$ observed tuples $(s_i, c_i)$, as depicted in Fig. 2(a) and (b), an empirical distribution can be defined as follows:

$$\rho_{S,C}^{\text{obs}}(s,c) = \frac{\pi^{\text{obs}}(s)}{N_{\text{obs}}} \sum_{i \in \mathcal{N}_s} \delta(c - c_i), \tag{2}$$

where let $\mathcal{N}_s$ indicate set of indices with $s_i = s$, such that $\sum_{i \in \mathcal{N}_s}$ indicates the sum over aver all $c_i$ which have been observed in combination with a given value of $s$. $\delta(x)$ is the $\delta$-distribution and $\pi^{\text{obs}}(s) = \frac{N_s}{N_{\text{obs}}}$, with $N_s$ denoting the total number of observations where $s_i = s$ (where the species is either present ($N_1$) or absent ($N_0$)).

It seems that the authors in fact build what they term *climate response of a given species* ($\text{PDF}_{\text{sp}}(c,s)$) on the probability distribution of $S$ conditioned on $C$:

$$\rho_{S|C}(s|c) = \frac{\rho_{S,C}(s,c)}{\rho_C(c)}. \tag{3}$$

$\rho_{S|C}(s|c)$, certainly is a function of $s$ **and** $c$, however, it is important to note that it is not a probability density function with respect to $c$. In words, this function expresses how probable the species will be present, given that the climate has the value $c$. The fact that the marginal distribution of the climate variable is not uniform but instead follows some probability $\rho_C(c)$ is already accounted for in Eq.(3), by the division by $\rho_C(c)$. That's why Eq.(3) is called a conditional probability. Compare line 82:

*Each observation can also be weighted to account for the uneven distribution of modern climate (Kühl et al. (2002), Bray et al. (2006)). Extreme values are usually under-represented (see, for instance, the inset histogram on Fig. 2c), and this bias can push the estimation of the PDF sp (c, s) towards the mean climate observed across the study area (i.e. towards the center of the "climate space").*

The authors then impose $S = 1$ and turn the Eq.(3) into a probability density with respect to $C$ by introducing a convenient normalization. However, this is all done implicitly and can only be understood from spending some time looking at Eq.(1) and (3) from the manuscript, where the authors compute the expected value of this new probability density they call *climate response* with respect to $C$. However, they do not discuss why $\rho_{S=1|C}(s = 1|c)$ can at all be interpreted as a measure for the *climate response* of a given species. As already mentioned, much of the mathematical considerations are omitted in the manuscript, which makes it very hard for the reader to find out what is actually going on 'behind the scenes'.

As a next step, the expectation and variance of the *climate response* computed from the observations (Eqs.(1) and (2) from the manuscript) are used to define a continuous normal or log-normal pdf for a given species. Here, I wonder if this is the most convenient way to estimate $\text{PDF}_{\text{sp}}(s, c)$. Would a least square optimization of the functional form to the observational data yield different and potentially better results?

Finally, the $\text{PDF}_{\text{sp}}(s, c)$ of different species of the same taxon are combined to yield one $\text{PDF}_{\text{t}}(s, c)$. It is, however, not explained how the quantity $\text{PDF}_{\text{t}}(s, c)$ should be interpreted. Does it express some probability, and if yes, what probability? As a reader, one could assume that the taxon's *climate response* should indicate the probability to find at least one species that is part of the taxon under given climatic conditions. However, this interpretation is inconsistent with the Eq.(6) of the manuscript, since this probability reads:

$$p(T = 1|C) = 1 - \prod_{\text{sp}}(1 - \rho_{\text{sp},S=1,C}(1, C)), \tag{4}$$

Where $T$ is the random variable associated with the presence $T = 1$ or absence $T = 0$ of a taxon, that is, at least one of it's comprising species, and $\rho_{\text{sp},S=1,C}$ are the climate response functions of the different species comprised in the taxon. Thus, $1 - \rho_{\text{sp},S=1,C}(1, C)$ denotes the chances for the absence of the species sp and the product over all species gives the chance to find none of the species given that the climate is $C$.

There is a substantial deficit of justification for the way the taxon's pdf is defined. I would highly recommend to elaborate more on this, since combining different probability densities is often not as trivial as it seems and subtle details are easily overseen. I would also like to encourage the authors to provide the mathematical interpretation of the quantity $\text{PDF}_{\text{t}}(s, c)$. What is the meaning of this function or probability measure?

From Eq.(7) the reader can finally understand, that the author follows a maximum likelihood approach for the reconstruction of past climate implicitly building upon Bayes

Theorem

$$p(c|t_1, t_2, ..., t_n) = \frac{p(t_1, t_2, ..., t_n|c)\, p(c)}{p(t_1, ..., t_n)},$$ (5)

where $\text{PDF}_{recnstrctn}(s, z)$ serves as the likelihood-function. I think that introducing Bayes theorem at the beginning and then deriving step by step a convenient expression for the likelihood-function would make it much easier for the reader to follow the line of thought.

There are two further obvious questions: First, why do the authors not make use of the full Bayesian theorem and deduce posterior probability distributions $p(c|t_1, ..., t_n, z)$ and with this a rigorous quantification of uncertainty? Introduction of convenient priors could actually account for the fact, that also for past climate there is a statistical bias in the sense that finding a species or a taxon under climatic conditions which occur very often is more likely than finding the species or taxon under climate conditions which occur only rarely even if the latter conditions are substantially preferred by the taxon or species. The author compensates this effect in the derivation of the climate response from the calibration data, however, he does comment on the fact that the same effect existed in the past. Second: the implicit normalization of the species climate responses undermines their interpretation as building blocks of the likelihood function. The individual normalization of the $\text{PDF}_{sp}(c, s)$ effectively introduces weights which do not have a mathematical meaning. As I already pointed out: $p(s|c)$ is not a probability density function with respect to $c$!

**Specific comments:**

l.9    I t

there is a space in the word.

l.22    Despite their conceptual simplicity and demonstrated capacity to reliably reconstruct climate from palaeoecological datasets, the limited availability of robust calibration datasets (i.e. regional collections of modern proxy samples) beyond the Northern Hemisphere extratropics has, however, hindered their application in these regions, despite the existence of suitable records from all environments worldwide (Chevalier et al., 2020b).

The reference of 'these regions' in unclear. Is 'these regions' regions beyond the Northern Hemisphere extratropics? Maybe 'outside this region'?

Should't extratropics be capitalized?

l.27    Built upon from the original work of Kühl et al. (2002) —— who first proposed to replace the commonly-used modern proxy samples with modern proxy geolocalised occurrence data to estimate probabilistic proxy-climate relationships — CREST estimates and combines probability density functions (PDFs) to reconstruct climate parameters.

I am not a native speaker, but are you sure that 'built upon FROM' is correct?

Also, at this stage 'CREST estimates and combines probability density functions (PDFs) to reconstruct climate parameters' could mean anything.

Maybe the sentence becomes more meaningful if you said: 'CREST estimates and combines climate response functions for numerous species to reconstruct climate parameters from fossil occurrences of these species.'

l.37    In addition to its broad applicability, CREST also bears some fundamental statistical properties that make it well-adapted to the analysis of palaeoecological datasets (Chevalier et al., 2020b).

I would rather say: CREST is equipped with some fundamental statistical features.

l.39    … CREST estimates, weights and propagates all the climate values that are compatible with the observed fossil data.

superfluous comma

l.45    analytical solution

This term could be misleading. Do you mean analytical in contrast to numerical?

l.52    In addition to its technical core, the package also contains an array of graphical diagnostic tools to represent the data at different pivotal steps of the reconstruction process and facilitate objective evaluations of the data and results.

It seems there is a 'to' missing in front of facilitate.

l.54    First, Section 2 summarises the mathematics and assumptions underpinning the approach and introduces the embedded calibration dataset.

I think underpinning should be replaced by underlying.

Fig.1   Conceptual illustration of the differences between a modelling approach based on the estimation of the most likely climate where all the probabilities are concentrated around the mode (light grey)

What exactly does the word 'mode' mean in this context? Also, in order to be precise, you should add, that already in panel (a) the best estimate contains statistical uncertainties. Otherwise the corresponding pdf would be a delta peak.

The two types of approaches are illustrated in (b) and (c) with two theoretical fossil samples (in blue and purple) representing two independent reconstructions of the same climatic parameter for the same time period.

It seems like what you are actually illustrating are the consequences of the two different reconstruction approaches?

The same samples are reconstructed using an approach that estimates their complete uncertainty distributions.

The 'samples' cannot be reconstructed.

In this case, the response of the blue sample is broader, and the response of the purple sample becomes bimodal.

Would you really say 'response'? It seems that you are talking about a posterior probability distribution. Maybe, it is worth introducing the term 'climate response' at a very early stage in the manuscript.

l.60    As is standard with statistical climate reconstruction techniques, the core process of CREST can be decomposed into two major stages: 1) estimating the modern climatic responses of the proxies observed in the fossil sequence (Fig. 2a-c) and 2) combining these responses to reconstruct past climates (Fig. 2d). In the following sections, the main elements of these two stages are presented along with all the parameters and/or modelling assumptions that can be modified in crestr. For an in-depth description of the method and its assumptions, the reader is, however, referred to .

please delete two doubled paragraph

Fig.2   Occurrences of four species part of the same pollen group exhibiting marked preferences for the lowest values of that climate (e.g. dark/cold values) cross the study area.

Shouldn't it be: the lowest value of that climate **variable across** the study area?

Maybe, it would be helpful if you pointed out, that the occurrence data is binary and does not account for the abundance of the species at a given location. For a better understanding, you could even indicate, that the species' PDFs are derived by binning the climate space and then compute the fraction of the climate variable's observation in a certain bin, that is accompanied by the presence of the species.

The histogram represents the proportion of the modern climate space (white) occupied by at least one of the four species (black), highlighting the higher chances of observing the taxon at the lower end of the climate gradient.

Please indicate that you are talking about the inset of panel (c).

Example of a posterior climate distribution resulting from the multiplication of PDFs and the type of synthetic statistics (e.g. optimum, mean, uncertainty range) one can derive from it.

This panel is very hard to understand within this Figure. I would suggest to divide the 4 species into two taxa such that panel c would show two different taxon's pdfs. Next, I would recommend supplementing the caption with the information that in a hypothetical proxy sample the two taxa were observed. This would allow the reader to understand much faster what is actually shown in panel (d). Please consider my objections presented in the comments on Section 2. with respect to the term 'posterior distribution' in this context.

l.72    In CREST, PDFs are used to transform the information contained in the modern observations of biological climate proxies into probabilistic climate responses. A PDF thus represents a weighted ensemble of all the conditions where the proxy is observed today. PDFs can be fitted in one or two steps depending on the nature and taxonomic resolution of the studied proxy. Climate responses are first fitted at the species level (hereafter PDF sp (c, s) with c representing the studied climate variable and s a species), and when necessary, these PDF sp (c, s) are then combined together to meet the taxonomic resolution of the fossil taxon (hereafter PDF tx (t, c) with t representing the observed taxon).

This sounds as if *a pdf* is something new…. The term simply denotes a specific class of functions.

l.72    In CREST, PDFs are used to transform the information contained in the modern observations of biological climate proxies into probabilistic climate responses.

I am not aware of the term 'climate response'. However, it might be that this is a typical term in the community that works with ecological proxy data, in that case, please ignore my comment. I would recommend to identify the climate response with the likelihood function in a Bayesian setting.

l.73    A PDF thus represents a weighted ensemble of all the conditions where the proxy is observed today.

hmm… There a two things which I find confusing about this sentence.

1) coming more from an ice perspective I am used to  proxies  whose  value  and  not  whose presence or absence can be used to for paleo climate reconstruction (most prominently levels of d18o can be used to study past temperature) – it seems that this is different in this context. Maybe this could be clarified at an early stage.

2) A pdf usually characterized the probability for a random variable to assume a certain value in a random experiment. I assume, the random variable in this case is the climate variable x under study, conditioned on the presence of a given proxy y.
P(a<x<b| y) = int_a^b pdf(x|y) dx

If this is what you aim to express, I don't think the above statement is very precise.

l.87 In CREST, this weighting can be accounted for by first sorting the N climate values that compose the climate space into bins of equal sizes (e.g. 2 ◦ C or 50 mm).

To me, it is unclear what 'the climate space' is. Is it a spatially extended region under study which is subdivided into N grid cell, each of which can be assigned a value in terms of a specific climate variable? Or is the 'the climate space' simply the value range which is covered by the climate variable globally (regionally)?

It seems, that N is the total number of observations and that each observations is associated with two variables, namely the climate variable – which is continuous – and the presence or absence of a certain species, which is a binary observation.

l.90 Please define the variables that you use in formulas. From the context, I could guess that $m_{s,c}$ is the mean value for some climate variable c, from different observations $c_i$ of this variable. Probably, conditioned on the presence of a certain proxy species s. Further I assume, that $k(c_i)$ are the weights?

l.110 Finally, the PDF sp (s, c) of the S(t) species composing taxon t are linearly combined to create the climate response of taxon t to climate variable c (Eq. 6).

What exactly is meant by linearly combined? Does that mean $PDF_{t,c} = \sum \alpha_i PDF_{s_i,c}$, where $\alpha$ are weights? how do you choose the weights?

The authors use the point estimates for mean and variance from the observed histograms to define a gaussian $pdf_{s,c}$. Is this equivalent to fitting a gaussian to the relative histogram in a least squared sense?

I must say, that I do not find the way how the species pdfs are combined to the taxon's pdf very convincing. To give a counter example: Let's say over a region as displayed in Fig.2 (a), there are k grid cells where the climate variable has the value c*. Let half of the grid cells be populated by species s1 and let also half these grid cells be populated by species s2, with an overlap such that in total ¾ of these k grid cells are populated by either s1 or s2.

Equation (6) now suggests, that the PDF_tax will yield a probability of 50% that under climate condition c* the taxon is present (given that the gaussian fit to for the species was fairly accurate). In fact, from the observations we know, that there is a 75% chance that under the climate conditions c* the taxon can be found.

Please see my considerations in my detailed comment to Section 2.

l.110 With the PDF tx (t, c) calibrated, posterior climate reconstructions can be estimated from their multiplication (Eq. 7, where z represents the age or depth of the sample to reconstruct, and Fig. 2d).

This is a very generic sentence, which is true only under very specific circumstanced, namely, for a given time in the past, the presence of different taxons at the same location must be evident from proxy records. Then, the different PDF_{tx} can be used to refine the past climate's reconstruction.

However, I believe, it would be worth explaining in one or two sentences, how past climate is reconstructed from a single taxon proxy record.

l.114 As such, it is possible to select a subset of climatically sensitive taxa to reconstruct each climate variable and maximise the reconstruction signal (Chevalier and Chase, 2015), even if it is not always mandatory (Chevalier et al., 2021).

It is unclear what is meant by the 'maximisation of the reconstruction signal'. I assume the authors mean, that the reconstruction's uncertainty is minimized or even more precise, that the width of the posterior distribution of the reconstructed climate variable is minimized.

Another sentence shortly before this one also starts with 'as such'.

even if it is not always mandatory (Chevalier et al., 2021)

I do not really understand this comment. In my view, it is always desired to reconstruct the past climate as precisely as possible. Of course, it is not mandatory to reconstruct past temperatures in northern Europe to the precision of two digits behind the comma to deduce that there have been ice ages. So why do the authors add this comment here and even provide a reference for this statement?

Eq.(7) Typically, equations are part of sentences. The normalization seems a bit odd: $1^{1/\#observed\ taxons}$?

I have so far not encountered an exponential weighing scheme like the one used here, but that does not mean anything. It only seems a bit odd, that previously objects which are not pdf's have been normalized to one and now, the pdf_recstrctn is obviously not integrate to one anymore after introduction of the weighing. - Well, it's a likelihood function and not a pdf, so there is no need that the expression integrates to one.

l.125 Maybe, before explaining the normalization of the data, you could say a few words about the structure of the data. How is the presence of a taxon measured? How can you compare the

abundance of different taxa with each other? Do you compare weights? Or maybe, this is clear to everybody from the community?

Eq.(8) Again, the formula should be embedded in a sentence. Also, it would be of help, if the variables used in the formula are defined close to the formula. Finally, z was defined as depth, which is a continuous variable – obviously, measurements are taken at discrete depths $z\_i$. The sum should then correctly run over the index i and not over the continuous variable z.

l.138 The is a spacing missing above the section title

l.150 The coordinates of all the presence records of these six common palaeoecological fossil proxies were upscaled at a spatial 150 resolution of $0.25 \times 0.25 \circ$

I assume the '150' is a misprint? What means 'upscaled at' ? Do you mean binned into the grid? Or maybe 'sorted into the grid'?

Fig.3 Data density of the six climate proxies available in the gbif4crest calibration database.

I assume that the density is defined by the number of presence observation of different species within a certain grid cell, divided by the specific surface of that cell? Please add a unit to the colorbar.

Table1 List of terrestrial variables available in the gbif4crest database. Each one can be selected in crestr using its associated code. List of abbreviations: (Temp.) Temperature, (Precip.) Precipitation.

Is this the *'climatological data'* mentioned in line 165? And is this data available for each grid continental cell that? Why are the *'environmental'* or *'geopolitical'* variables not listed here?

Table2 See table 1.

Fig. 4 Maybe, you could in the DISTRIB table add one line – I believe that would help to understand the structure of these tables.

What means 'type of observation: literature' in the DISTRIB_QDGC table?

The figure caption could be a lot longer and explain the purpose of the different tables.

l.204 inputs: contains the raw data (e.g. the counts/percentages, the ages of the samples or the names of the fossil taxa).

What are the *raw data* used as input for the crestObj? I may only guess, that this is proxy data the user has to provide?

l.211 – reconstructions: contains all the results (e.g. best estimates, synthetic error measurements as well as the full posterior distribution of the uncertainties).

From Section 2. it is unclear to me, how a full posterior distribution of the climate is obtained within the presented modeling approach. Please see my detailed comment on Section 2.

l.214    Five different input data files are compatible with crestr. However, most applications will only require two file (the df and PSE files, see below) to be created. More specific applications may require up to four of these files. All the files can be prepared outside the R environment and imported using standard R functions.

will only require two files

See l.204 – it becomes clear now, that this input data is actually the proxy data users aim to build their climate reconstruction upon. I would propose to state this explicitly – it might be obvious to the author, though for the reader it is not.

l.223    The proxy-species equivalency (PSE) table

I am little confused by this section. In Section 2. it is explained how the climate responses of different species of a given taxon are combined to the taxon's climate response. Hence, there are two levels involved.

Now, it seems that the category 'taxon' was replaced by the category 'family' and on top of that a third level 'Genus' was added. So I wonder how, climate responses on the species level are first combined to a 'genus' level and then to a 'family' level?

Maybe this confusion of mine is simply due to my lack of knowledge in this field. If you think, Sec 3.3.2 will be understood correctly by the relevant audience, then please ignore this comment.

Also, I would find it helpful to understand, what type of proxy data for past climate can actually be observed? Species? Or only taxa? Why and how does this differ between situations?

l.316    – To estimate reliable PDFs, it is recommended to use at least 20 distinct occurrences for each species, but different values can be specified with the minGridCell parameter.

Here, I do not understand whether 'distinct' refers to the level of individual observations as stored in the DISTRIB table, or to the level of QDGC grouped observations?

l.362    4.3 Estimating the climate responses (the PDFs)

I recommend to call *'the PDF's'* consistently climate response functions throughout the entire manuscript. PDF is a specific type of function that fulfills certain requirements. As mentioned previously, the climate response functions are likelihood functions and if they weren't normalized they would not be pdfs. Also, the term 'climate response functions' describes more accurate the purpose of the functions.

l.373    set geoWeighting to TRUE if the species PDFs of the different composing species should be weighted according to the square-root of the extent of their modern distribution.

This refers to combining the species' pdfs to the taxa pdfs?

l.387 2) the climate values to reconstruct are likely to be in the study area (the reconstructions are bounded by the lowest and highest values observed in the modern climate space)

Can you think of a term other than 'study area'? This sounds more a like a geographical location.

Or maybe you actually mean the geographical region. In that case, the above sentence should be specified, e.g.: the climate values to reconstruct are likely to be covered by present day climate values in the study area.

l.393 In our case study, the spatial variability represent true patterns in regional species diversity with the presence of several biodiversity hotspots across eastern and southern Africa (Myers et al., 2000).

represents

l.397 This diagnostic figure is also very important to identify potential local or global correlations between different climate variables and assess the risks of confounding variables (Juggins (2013), Chevalier et al. (2020b)).

What is the 'risk of confounding varaibles'? Please elaborate.

Fig.6 'Number of unique species occurrences' in other words means 'Number of different species observed at least once in a grid cell' is that correct?

Please give the different panel labels (a), (b),… and make sure that all axis have labels.

Fig.7 In the top row, the map represents the density of unique species occurrences per grid cell and the time series represents the variability of the taxon against time or depth.

If I am not completely mistaken, the map shows data from the QDGC_DISTRIB table, that is calibration data from present day observations, while the time series shows data from the proxy data from the sediment core. Please make sure to specify this unambiguously in the caption.

Also, make sure that all axis are labeled and provide labels for different panels of the plot.

The color bar is again labeled 'Number of unique species occurrences'. I assume the difference to Fig.6 is that here, the number of observed species belonging to the taxon 'Ericaceae' within a grid cell in shown, while Fig.6 shows the total number of observed species under consideration. Please make sure this difference is specified either in the caption or in the label of the colorbar itself.

The left panel on the bottom row is not very useful, since the information on the presence of observations hides the information on the climate. Maybe, you could simply sketch the outline of those connected patches (with some minimum size) which are covered by the taxon.

l.429 (all the probabilities sum to 1)

Maybe better: the PDFs integrate to 1.

l.435 *This type of representation can be particularly helpful to have objective interpretations of ecological changes from pollen diagrams (Chevalier et al., in press, Quick et al. (2021)).*

I do not understand this sentence. Interpretations can never be objective.

l.452 *Here for instance, both Aizoaceae and Chenopodiaceae/Amaranthaceae were excluded because they are not primarily sensitive to temperature in southern Africa.*

To understand this, it would be nice to have them included in the violin plot (Fig.8).

Fig.10 I can only guess, that the 64 in the upper right corner indicates the individual sample from the marine sediment core that is associated with a certain depth and that these kinds of plots can be obtained for all individual samples? If not, and the plot refers to the entire reconstruction, then I do not understand how the weights can be specified, since early in the manuscript the authors explained that the weights of the different taxa in the climate reconstruction can vary over the depth of the core.

*The black curve represents the posterior MAT reconstruction, from which a 'best' climate estimate can be estimated from the maximum of the curve and uncertainties derived by calculating the area under the curve.*

From what I understood from Section 2., the black curve does not show a posterior probability distribution for the past climate but instead the likelihood function.

The line thickness can hardly be discriminated by eye. Did you try plotting the different climate responses according to their weights as expressed in Eq.(7) of the manuscript?

l.501 *The presence of multimodality can be the underlying cause of apparent noise in the reconstructions because minor changes in the taxa composition or percentages can force the system to oscillate between two maxima and thus 'appear' noisy, even if the background rate of change is minor.*

We are not really confronted with an oscillating system here. Maybe rephrase:

The presence of multimodality in the climate reconstruction of subsequent samples can be the underlying cause of apparent noise in the reconstructions because minor changes in the taxa composition or percentages can easily switch the order of the two local maxima in terms of height. This results in a jumpy time series of the 'optimal climate reconstruction', even if the background rate of change is minor.

Fig. 11 Why are all anomalies positive? Is this by coincidence and due to the choice of taxa presented in the plot?

*Here, the results are only showed for a subset of the taxa observed in marine core MD96-2048 (only 20 out of the 171 available taxa are represented).*

*'Showed'* should be shown.

The title of the plot 'mean annual temperature' is not very suited. As far as I understood, the plot shows the anomalies for reconstructions that are based on all but one taxa with respect to the reconstruction based on the full set of taxa.

---

## Author Comment (AC1)

The comments of Reviewer 1 are in black, Reviewer 1's quotes of the original manuscript are in purple, and my responses in blue. Italicised texts are quotes from the revised manuscript.

**General Comments**

I very much appreciate the effort the author has made to share his code or should I say software with the community. Without having tested the R package by myself, I am confident to say that it constitutes a very valuable contribution for the community and that many researchers can benefit from it. The package is well described in the manuscript and it seems that the author achieved his goal of keeping the hurdles for application as low as possible while at the same time providing a reasonable degree of flexibility. From a user's perspective package's automated query of the databases is the optimal solution. Also, the diagnostic tools seem to be very helpful for assessing the quality of past climates reconstructions.

Thank you very much for this praise of the tool I developed. I understand that Reviewer 1 is not an expert on palaeoecological datasets, but the fact that Reviewer 1 did not try the package, considering it is the novelty presented in this paper, feels a bit like a missed opportunity. I am, however, very grateful for all the other comments and suggestions. Despite what I just said, I think Reviewer 1's "outside" perspective has also helped me clarify many details in the manuscript, especially things that are usually taken for granted but still need to be explained to reach a broad audience. I believe addressing Reviewer 1's comments has really helped improve the manuscript's clarity and accessibility, and I am very thankful for that.

Overall, the manuscript conveys the functioning of the R package in a decent way. However, there are several inaccuracies in the language and sometimes it seems the paper has been written with a lack of thoroughness. For example, there are plots without axis labels. The figure and table captions are generally very short. I would suggest to give all of them a careful review. In principle, the figures as well as the tables should be understandable from their captions and the reader should not have to search in the main text for the relevant pieces of information. Additionally, formulas appear to be detached from the text and the definitions of the entities that appear in formulas are sometimes given way before the formula is presented. In general, any formula should be part of a sentence. The structure of the manuscript is well chosen. The introductory part could, however, do a better job in setting the scene. It took me a long time to understand, that we are dealing with proxy data on the one hand, with the abundance of a certain proxy being given in continuous units with respect to a depth (or time) axis and on the other hand with binary presence data from modern observations spread over the entire globe, which if supplement by the corresponding climate data serves for the calibration of the 'climate response function'. Maybe, this is clear to everybody working with palaeoecological proxy data, to me it was not. Another point that confused me while reading, is that in Section 2. there is a clear two-level structure comprised of taxa and the species of each taxon. Correspondingly, the way how PDF's of the different species are combined to the one of a taxon is introduced. Later in the manuscript, more levels added to this structure like families and genus. I do neither understand how PDFs are combined on these intermediate levels nor what is actually measured in the proxy data? I thought, one could either measure the abundance a taxon or of specific species in a sediment core? Again, maybe this is obvious to people who are used to work with palaeoecological proxy data.

I considered the points raised by Reviewer 1 and tried to add as much clarity as possible across the manuscript. I agree that some elements might have been a bit too disconnected, indeed. Having taken some distance from the manuscript, I also realised that many comments came from an improper introduction of the studied objects and unclear areas of the manuscripts, also echoing some of the comments made by Reviewer 2. As such, I have largely modified the introduction and used it to better 'set the scene' and more clearly highlight the content of the paper. I also provided more details at the beginning of sections 2 and 3 to better introduce the type of data/proxies that I am discussing throughout the text. I think the level of details

was insufficient in places to reach a broad audience. I hope these changes will allow colleagues from other fields to understand the paper and the tool.

My greatest concern relates to the mathematical presentation of how the package actually computes the past climate's reconstruction and with this to Section 2. of the manuscript. This part of the manuscript is definitely lacking the required accuracy and is in some sense misleading. The author calls the function $PDF_{reconstrctn}(c, z)$ a posterior climate reconstruction (compare l.110). This suggests, that a full Bayesian approach has been pursued. Instead, after having careful studied Section 2. I understand that the climate reconstruction presented by the author is in fact a maximum likelihood estimate of the past climate and that the uncertainties presented in Fig.9 correspond to the percentiles of the corresponding likelihood function. I would really like to encourage the author to implement a full Bayesian approach in the R package, if not in this version, then in an updated version of the software.

As explained in length below, the use of 'posterior' was an inappropriate use of the language. The model is not Bayesian … yet! Going full Bayesian is definitely an avenue of development that is being explored now that all the foundations (calibration data for different proxies, climate variables, automatic plots, etc.) for the global applicability have been created with this package. Still, I think the model works well in its current form, and it is hoped that this first non-Bayesian version of the package will be of interest. Subsequent versions of the package will definitely include a complexification of the model.

Despite this major issue, I would like to emphasize that this manuscript is only the description of what has actually been the main part of the work, and that is the R package. I consider this package as a very valuable contribution and this paper is a decent description of the package and thus definitely merits publication.

Thank you for the general positive feedback. I hope that my responses below and my manuscript modifications will address your concerns adequately.

**Detailed Comments on Section 2.**

[I could not copy / paste this bit of text from the pdf because it was full of mathematical terms.]

I greatly appreciate all the efforts Reviewer 1 made to suggest a full Bayesian version of the CREST algorithm. This is actually a work that is currently underway. However, progress at this stage is insufficient to be released, but we will look in detail at the suggestions that were made by Reviewer 1. However, the model of Reviewer 1 is based on an erroneous assumption about the available data type. The model proposed by Reviewer 1 is based on the existence of both presence and absence data of plant taxa. However, absence data are not available. Only presences are. While some absence might be inferred from regions with dense data networks (*e.g.* in Europe), sampling is incomplete in many other regions of the world (see Fig. 3 for instance), and many locations where a taxon is not observed could be an actual absence or a proof of insufficient sampling. This information is thus unreliable and should not be used. As a consequence, the CREST model is based on presence-only data, which explains some of the modelling choices that were made to accommodate data.

This 'mistake' of Reviewer 1 is likely caused by the lack of clarity of my introduction and the model's description. I hope the new version of the introduction and the new section at the beginning of section 2 better introduce these problems and better support the choices made to develop the CREST model almost a decade ago.

**Specific comments:**

l.9 It
there is a space in the word.

Corrected.

l.22 Despite their conceptual simplicity and demonstrated capacity to reliably reconstruct climate from palaeoecological datasets, the limited availability of robust calibration datasets (i.e. regional collections of modern proxy samples) beyond the Northern Hemisphere extratropics has, however, hindered their application in these regions, despite the existence of suitable records from all environments worldwide (Chevalier et al., 2020b).
The reference of 'these regions' in unclear. Is 'these regions' regions beyond the Northern Hemisphere extratropics? Maybe 'outside this region'?

The introduction has been rewritten and the specific sentence highlighted here is rephrased as: "*However, the limited availability of the necessary calibration datasets beyond the Northern Hemisphere extratropics has often hindered their application in many environments and regions where quantified climate records are needed, despite the existence of suitable fossil records*".

Should't extratropics be capitalized?

I am not a native speaker either and have not seen it capitalised anywhere else.

l.27 Built upon from the original work of Kühl et al. (2002) —— who first proposed to replace the commonly-used modern proxy samples with modern proxy geolocalised occurrence data to estimate probabilistic proxy-climate relationships — CREST estimates and combines probability density functions (PDFs) to reconstruct climate parameters.
I am not a native speaker, but are you sure that 'built upon FROM' is correct?

'From' was deleted.

Also, at this stage 'CREST estimates and combines probability density functions (PDFs) to reconstruct climate parameters' could mean anything.
Maybe the sentence becomes more meaningful if you said: 'CREST estimates and combines climate response functions for numerous species to reconstruct climate parameters from fossil occurrences of these species.'

Rephrased as follow: "CREST estimates and combines probabilistic proxy-climate relationships to reconstruct past climate parameters from fossil proxy observations".

l.37 In addition to its broad applicability, CREST also bears some fundamental statistical properties that make it well-adapted to the analysis of palaeoecological datasets (Chevalier et al., 2020b).
I would rather say: CREST is equipped with some fundamental statistical features.

Corrected.

l.39 ... CREST estimates, weights and propagates all the climate values that are compatible with the observed fossil data.
superfluous comma l.45

The comma is necessary, but weighs (the verb) was misspelt as weights (the noun). Hence the confusion. The sentence has been simplified as follow: "*While techniques such as MAT or WA-PLS are primarily designed to associate modern proxy observations with their 'most likely' or 'mean' climate values only***, CREST estimates and weighs all the climate values that are compatible with the observed fossil data.** *As such, the climate reconstructions obtained from CREST can be understood as an ensemble of all data-compatible climate values and not a simpler, less informative 'most likely' or 'best' climate estimate with statistical errors.*"

 analytical solution
This term could be misleading. Do you mean analytical in contrast to numerical?

This term was confusing indeed. It is now removed.

l.52 In addition to its technical core, the package also contains an array of graphical diagnostic tools to represent the data at different pivotal steps of the reconstruction process and facilitate objective evaluations of the data and results.
It seems there is a 'to' missing in front of facilitate.

Grammatically, there is no need to repeat 'to' here. At least, according to my (British) English grammar corrector.

[Figure]

In addition to its technical core, the package also contains an array of graphical diagnostic tools to represent the data at different pivotal steps of the reconstruction process and  facilitate objective evaluations of the data and results.

l.54 First, Section 2 summarises the mathematics and assumptions underpinning the approach and introduces the embedded calibration dataset.
I think underpinning should be replaced by underlying.

These two words seem largely synonymous in this context, but 'underpinning' was replaced by 'underlying'.

Fig.1 Conceptual illustration of the differences between a modelling approach based on the estimation of the most likely climate where all the probabilities are concentrated around the mode (light grey)
What exactly does the word 'mode' mean in this context? Also, in order to be precise, you should add, that already in panel (a) the best estimate contains statistical uncertainties. Otherwise the corresponding pdf would be a delta peak.

I am not sure I understand why Reviewer 1 is confused about using this term. Here I used mode according to its definition, i.e. the value with the highest probability. It is also widely recognised that any parameter estimate has some associated errors. As such, I am not sure much emphasis is essential here. Maybe this is a difference between fields, but I am pretty sure nobody would expect reconstructions to be a single (delta?) peak. I have modified the caption as follow: *"(a) Conceptual illustration of the differences between a modelling approach based on the estimation of the most likely climate value with small statistical errors surrounding it (light grey), and a modelling approach focused on the full spread of the data with the probabilities more spread along the climate gradient (dark grey)."*

The two types of approaches are illustrated in (b) and (c) with two theoretical fossil samples (in blue and purple) representing two independent reconstructions of the same climatic parameter for the same time period.
It seems like what you are actually illustrating are the consequences of the two different reconstruction approaches?

Corrected as "The results of the two approaches are illustrated [...]".

The same samples are reconstructed using an approach that estimates their complete uncertainty distributions.
The 'samples' cannot be reconstructed.

Replaced by 'the same fossil assemblages are analysed'

In this case, the response of the blue sample is broader, and the response of the purple sample becomes bimodal.
Would you really say 'response'? It seems that you are talking about a posterior probability distribution. Maybe, it is worth introducing the term 'climate response' at a very early stage in the manuscript.

I am not sure the term 'climate response' is more explicit at this point of the manuscript. I am now referring to those using a more generic 'reconstruction'.

Full caption with the changes suggested in the past 3 comments in bold: *"(a) Conceptual illustration of the differences between a modelling approach based on the estimation of the full spread of the data with the probabilities spread along the climate gradient (dark grey), and **a modelling approach focused on the estimation of the most likely climate value with small statistical errors surrounding it** (light grey). In both cases, the area under the curve sums to one. The results of the two types of approaches are illustrated in (b) and (c) with **two theoretical fossil assemblages (in blue and purple) used to produce two independent reconstructions of the same climatic parameter for the same time period**. (b) The two reconstructions are derived from a method focused on only estimating the most likely climate value, resulting in 'apparently' incompatible reconstructions. (c) **The same fossil assemblages are analysed using an approach that estimates their complete uncertainty distributions**. In this case, **the blue reconstruction** is broader, and **the purple reconstruction** becomes bimodal. When the full spread of these uncertainties is considered, the two reconstructions are not incompatible anymore, and a joint climate estimate (gold) can be derived from their overlapping sections (hashed polygons)."*

l.60 As is standard with statistical climate reconstruction techniques, the core process of CREST can be decomposed into two major stages: 1) estimating the modern climatic responses of the proxies observed in the fossil sequence (Fig. 2a-c) and 2) combining these responses to reconstruct past climates (Fig. 2d). In the following sections, the main elements of these two stages are presented along with all the parameters and/or modelling assumptions that can be modified in crestr. For an in-depth description of the method and its assumptions, the reader is, however, referred to .
please delete two doubled paragraph

The duplicated paragraph was removed.

Fig.2 Occurrences of four species part of the same pollen group exhibiting marked preferences for the lowest values of that climate (e.g. dark/cold values) cross the study area.
Shouldn't it be: the lowest value of that climate **variable across** the study area?

It is, indeed. Corrected.

Maybe, it would be helpful if you pointed out, that the occurrence data is binary and does not account for the abundance of the species at a given location. For a better understanding, you could even indicate, that the species' PDFs are derived by binning the climate space and then compute the fraction of the climate variable's observation in a certain bin, that is accompanied by the presence of the species.

I have added a new section (2.1) that introduces the type of data required by the model. I think this will make the 'modelling' part that follows easier to follow and understand.

The histogram represents the proportion of the modern climate space (white) occupied by at least one of the four species (black), highlighting the higher chances of observing the taxon at the lower end of the climate gradient.
Please indicate that you are talking about the inset of panel (c).

Corrected as follow: "The **histogram (inset panel) represents** the proportion of the modern climate space (white) occupied by at least one of the four species (black), highlighting the higher chances of observing the taxon at the lower end of the climate gradient."

Example of a posterior climate distribution resulting from the multiplication of PDFs and the type of synthetic statistics (e.g. optimum, mean, uncertainty range) one can derive from it.
This panel is very hard to understand within this Figure. I would suggest to divide the 4 species into two taxa such that panel c would show two different taxon's pdfs. Next, I would recommend supplementing the caption with the information that in a hypothetical proxy sample the two taxa were observed. This would allow the reader to understand much faster what is actually shown in panel (d). Please consider my objections presented in the comments on Section 2. with respect to the term 'posterior distribution' in this context.

I have repackaged the figure to simplify its meaning. The right-most curve does not represent a reconstruction anymore but the response of the pollen taxon (a zoom on the black curve presented in panel c). As explained above, all notions to a posterior reconstruction have been removed. The figure shows how to estimate the response of a pollen taxon using a given climatology and the distributions of four composing species.

l.72 In CREST, PDFs are used to transform the information contained in the modern observations of biological climate proxies into probabilistic climate responses. A PDF thus represents a weighted ensemble of all the conditions where the proxy is observed today. PDFs can be fitted in one or two

steps depending on the nature and taxonomic resolution of the studied proxy. Climate responses are first fitted at the species level (hereafter PDF sp (c, s) with c representing the studied climate variable and s a species), and when necessary, these PDF sp (c, s) are then combined together to meet the taxonomic resolution of the fossil taxon (hereafter PDF tx (t, c) with t representing the observed taxon).
This sounds as if *a pdf* is something new…. The term simply denotes a specific class of functions.

A pdf is not anything new, but it is probably not something commonly used by the target users of crestr? As such, I believe a definition of what it is and how it could be grasped in the context of CREST is not superfluous.

l.72  In CREST, PDFs are used to transform the information contained in the modern observations of biological climate proxies into probabilistic climate responses.
I am not aware of the term 'climate response'. However, it might be that this is a typical term in the community that works with ecological proxy data, in that case, please ignore my comment. I would recommend to identify the climate response with the likelihood function in a Bayesian setting.

The sentence was rephrased as follow: "*The transformation of the information contained in the modern observations of the biological climate proxies into probabilistic climate responses can be done in one or two steps depending on the nature and taxonomic resolution of the studied proxy.*". Climate response is a term that I have heard and been using for many years without any apparent issues.

l.73  A PDF thus represents a weighted ensemble of all the conditions where the proxy is observed today.
hmm… There a two things which I find confusing about this sentence.

1) coming more from an ice perspective I am used to proxies whose value and not whose presence or absence can be used to for paleo climate reconstruction (most prominently levels of d18o can be used to study past temperature) – it seems that this is different in this context. Maybe this could be clarified at an early stage.

2) A pdf usually characterized the probability for a random variable to assume a certain value in a random experiment. I assume, the random variable in this case is the climate variable x under study, conditioned on the presence of a given proxy y.
$P(a<x<b \mid y) = \int_a^b pdf(x \mid y) \, dx$

If this is what you aim to express, I don't think the above statement is very precise.

The aim of this sentence was to clarify things, but it seems to be doing the exact opposite. Since it was unclear to both reviewers, it was removed from the manuscript.

l.87 In CREST, this weighting can be accounted for by first sorting the N climate values that compose the climate space into bins of equal sizes (e.g. 2 ◦ C or 50 mm).
To me, it is unclear what 'the climate space' is. Is it a spatially extended region under study which is subdivided into N grid cell, each of which can be assigned a value in terms of a specific climate variable? Or is the 'the climate space' simply the value range which is covered by the climate variable globally (regionally)?

*What climate space means is not defined in the new section 2.1. "Climatology(ies) of the variable(s) to reconstruct gridded at the same resolution as the modern occurrences. All the climate values observed in the study area define what is later referred to as the climate space."*

It seems, that N is the total number of observations and that each observations is associated with two variables, namely the climate variable – which is continuous – and the presence or absence of a certain species, which is a binary observation.

*Absence data are not available; this is now part of the new section 2.1 of the revised manuscript.*

l.90 Please define the variables that you use in formulas. From the context, I could guess that $m_{s,c}$ is the mean value for some climate variable c, from different observations $c_i$ of this variable. Probably, conditioned on the presence of a certain proxy species s. Further I assume, that $k(c_i)$ are the weights?

*All these variables are now clearly defined in the text (see section 2.2 of the revised manuscript).*

l.110 Finally, the PDF sp (s, c) of the S(t) species composing taxon t are linearly combined to create the climate response of taxon t to climate variable c (Eq. 6).
What exactly is meant by linearly combined? Does that mean $PDF_{\{t,c\}} = \sum \alpha_i PDF_{\{s_i,c\}}$, where $\alpha$ are weights? how do you choose the weights?

*The model offers to weigh the pdfs differently based on how much we know about the taxa distributions. This option is not mandatory in the model. In many regions a trade-off between quality and quantity has to be made and to include more species, we need to lower the quality bar. This weighting option is just a little trick to rebalance things a bit.*

*This linear combination ensures that all the climate values that support the presence of at least one species have a non-null probability in the PDFtx(t,c). Contrary to the previous step, no additional constraints are added here. The distribution of the PDFtx(t,c) can thus be asymmetrical and even multimodal if different (groups of) composing species exhibit distinct climate requirements. **An additional option is to weigh the different PDFsp(s,c) by the square root of the number of individual occur- rences composing their distribution (Ns). Considering that it is more difficult to estimate robust parameters with few points, this weighting gives more importance to the species with more extensive geographical distributions today. Said differently, it gives more weight to the species whose climate responses can be the most reliably defined.***

The authors use the point estimates for mean and variance from the observed histograms to define a gaussian pdf_{s,c}. Is this equivalent to fitting a gaussian to the relative histogram in a least squared sense?

Is it *strictly* equivalent? I cannot tell. But it would certainly lead to very similar results considering the nature of the estimated parameters.

I must say, that I do not find the way how the species pdfs are combined to the taxon's pdf very convincing. To give a counter example: Let's say over a region as displayed in Fig.2 (a), there are k grid cells where the climate variable has the value c*. Let half of the grid cells be populated by species s1 and let also half these grid cells be populated by species s2, with an overlap such that in total 3⁄4 of these k grid cells are populated by either s1 or s2.

Equation (6) now suggests, that the PDF_tax will yield a probability of 50% that under climate condition c* the taxon is present (given that the gaussian fit to for the species was fairly accurate). In fact, from the observations we know, that there is a 75% chance that under the climate conditions c* the taxon can be found.

Please see my considerations in my detailed comment to Section 2.

I think it is a very interesting way of looking at the data that deserve further thinking. This is very important and I will try to take it into account to improve the algorithm and make better reconstructions, indeed. However, this paper is about the implementation of the algorithm presented in my 2014 paper, and I want to keep it in that way. The model has proven to be able to produce reliable reconstructions as it is. I have, however, added a paragraph of discussion on how the CREST algorithm could be improved in the future, and it definitely includes some of the considerations Reviewer 1 made in its detailed comments of section 2 (including a transition to a fully Bayesian model).

As I mentioned earlier, developments towards a more Bayesian model are currently underway. But progress and validation are unfortunately not advanced enough to be published yet.

l.110 With the PDF tx (t, c) calibrated, posterior climate reconstructions can be estimated from their multiplication (Eq. 7, where z represents the age or depth of the sample to reconstruct, and Fig. 2d).
This is a very generic sentence, which is true only under very specific circumstanced, namely, for a given time in the past, the presence of different taxons at the same location must be evident from proxy records. Then, the different PDF_{tx} can be used to refine the past climate's reconstruction.

However, I believe, it would be worth explaining in one or two sentences, how past climate is reconstructed from a single taxon proxy record.

I am afraid I am not understanding what Reviewer 1 is asking here. Single taxon reconstructions are not a thing with palaeoecological datasets. The value of these datasets is their multitude! Reconstruction techniques tap into the co-variations of taxa to infer climate.

l.114 As such, it is possible to select a subset of climatically sensitive taxa to reconstruct each climate variable and maximise the reconstruction signal (Chevalier and Chase, 2015), even if it is not always mandatory (Chevalier et al., 2021).
It is unclear what is meant by the 'maximisation of the reconstruction signal'. I assume the authors mean, that the reconstruction's uncertainty is minimized or even more precise, that the width of the posterior distribution of the reconstructed climate variable is minimized.

This phrasing was unclear and has now been clarified, as follow: "In this approach, the presence of each taxon in a sample is considered independent from the others. It is thus possible to select a subset of climatically sensitive taxa. **In some cases, identifying a subgroup of climate-sensitive taxa can help disentangle the different climate signals represented by the palaeoecological data and improve the quality of the reconstructions [@CC15], even if it is not always necessary [@Chevalier2021Limpopo].** In practice, these choices should be dictated by the data themselves and the users' understanding of the studied proxy system. _crestr_ provides graphical tools to help identify the possible climate sensitivities of the studied taxa [@Chevalier2021Atlas].".

Another sentence shortly before this one also starts with 'as such'.

Corrected.

even if it is not always mandatory (Chevalier et al., 2021)
I do not really understand this comment. In my view, it is always desired to reconstruct the past climate as precisely as possible. Of course, it is not mandatory to reconstruct past temperatures in northern Europe to the precision of two digits behind the comma to deduce that there have been ice ages. So why do the authors add this comment here and even provide a reference for this statement?

This part of the sentence referred to the beginning 'selecting a subset of sensitive taxa', which is not always required depending on the data. This was rephrased as follow: "*In this approach, the presence of each taxon in a sample is considered independent from the others. **It is thus possible to select a subset of climatically sensitive taxa. In some cases, identifying a subgroup of climate-sensitive taxa can help disentangle the different climate signals represented by the palaeoecological data and improve the quality of the reconstructions [@CC15], even if it is not always necessary [@Chevalier2021Limpopo].** In practice, these choices should be dictated by the data themselves and the users' understanding of the studied proxy system. _crestr_ provides graphical tools to help identify the possible climate sensitivities of the studied taxa [@Chevalier2021Atlas].*"

Eq.(7) Typically, equations are part of sentences. The normalization seems a bit odd: $1^{1/\#observed\ taxons}$?

I have so far not encountered an exponential weighing scheme like the one used here, but that does not mean anything. It only seems a bit odd, that previously objects which are not pdf's have been normalized to one and now, the pdf_recstrctn is obviously not integrate to one anymore after introduction of the weighing. - Well, it's a likelihood function and not a pdf, so there is no need that the expression integrates to one.

I think Reviewer 1 misunderstood how omega(t,z) was calculated, which created some confusion. I have now better explained the different options and I hope this expanded text clarified things. I also agree that the pdf_rcnstrctn is not a pdf, indeed. The object has been renamed RECON(c, z) and all the equations have been included in a sentence.

l.125 Maybe, before explaining the normalization of the data, you could say a few words about the structure of the data. How is the presence of a taxon measured? How can you compare the abundance of different taxa with each other? Do you compare weights? Or maybe, this is clear to everybody from the community?

I have explained the structure of the data in section 2.1, and I think that the questions asked here by Reviewer 1 wouldn't be asked by colleagues from the field that are familiar with palaeoecological datasets. For instance, a presence is simply determined by the observation of the taxon in the sample (which can subsequently be truncated if it doesn't reach a sufficient percentage), the abundances are never compared directly, only the trends. It is actually to overcome this lack of comparability that I propose the normalisation of the percentages.

Eq.(8) Again, the formula should be embedded in a sentence. Also, it would be of help, if the variables used in the formula are defined close to the formula. Finally, z was defined as depth, which is a continuous variable – obviously, measurements are taken at discrete depths z_i. The sum should then correctly run over the index i and not over the continuous variable z.

All formulas and chunks of code are now embedded in the text. I believe that indexing z would only make things unnecessarily complicated.

l.138 The is a spacing missing above the section title

This is a problem generated by the Rmardown -> markdown -> latex scripts. Fixing this is beyond my skills and I have no doubt this will be fixed at the typesetting stage.

l.150 The coordinates of all the presence records of these six common palaeoecological fossil proxies were upscaled at a spatial 150 resolution of 0.25 × 0.25 ∘

I assume the '150' is a misprint? What means 'upscaled at' ? Do you mean binned into the grid? Or maybe 'sorted into the grid'?

150 was a typo and is now fixed. 'Upscaled' was replaced by 'mapped onto'.

Fig.3 Data density of the six climate proxies available in the gbif4crest calibration database. I assume that the density is defined by the number of presence observation of different species within a certain grid cell, divided by the specific surface of that cell? Please add a unit to the colorbar.

The title of the figure is 'Number of occurrence data per grid cell'. I believe this states what the plots are about. I have, however, replaced 'Data density' with 'Distribution and grid cell density of the six climate proxies available […]' to avoid further miscomprehension.

Table1  List of terrestrial variables available in the gbif4crest database. Each one can be selected in crestr using its associated code. List of abbreviations: (Temp.) Temperature, (Precip.) Precipitation. Is this the *'climatological data'* mentioned in line 165? And is this data available for each grid continental cell that? Why are the *'environmental'* or *'geopolitical'* variables not listed here?

Table2  See table 1.

The non-climatic environmental and geopolitical variables are not meant to be reconstructed. These tables are about the variables that can be reconstructed. This is now clarified in the text (see below).

*"In the _gbif4crest_ database, all the QDGC grid cells were associated with a collection of terrestrial and oceanic environmental variables that can be reconstructed (@Fick_Hijmans_2017, @Zomer2008, @WOA_2018_temp, @WOA_2018_salinity, @WOA_2018_oxy, @WOA_2018_nutrients, @Reynolds2007, see details in Tables \ref{table:variables-terr} and \ref{table:variables-mari}). Despite the diversity of variables available, it is recommended to avoid serial reconstructions and, on the contrary, to identify the few important variables for the studied palaeoecological datasets _a priori_**. The grid cells were also associated with 'non-reconstructible' environmental and geographical descriptors that serve to tailor the calibration dataset to the users' needs. These include the coordinates, the elevation and elevation variability within the grid cell (Amante and Eakins, 2009), the country (https://www.naturalearthdata.com) or ocean (https://www.marineregions.org) names, as well as different levels of ecological classification for the terrestrial (Olson et al., 2001) and marine (Costello et al., 2017) realm**s."*

Fig.4 Maybe, you could in the DISTRIB table add one line – I believe that would help to understand the structure of these tables.

I did not modify the figure, but I added some additional explanations in the text regarding the structure of the tables. However, I also need to point out that understanding the tables is not necessary to use the package as the interface with the database is done 'under the hood'.

What means 'type of observation: literature' in the DISTRIB_QDGC table?

I have expanded on that in the main text and added a link to a webpage that explains how GBIF classifies its data.

The figure caption could be a lot longer and explain the purpose of the different tables.

While I generally agree that figures should be understandable on their own, in this case, it would only lead to a repetition of the text, which is quite long.

l.204 inputs: contains the raw data (e.g. the counts/percentages, the ages of the samples or the names of the fossil taxa).
What are the *raw data* used as input for the crestObj? I may only guess, that this is proxy data the user has to provide?

Yes, indeed. This has been clarified.

l.211 – reconstructions: contains all the results (e.g. best estimates, synthetic error measurements as well as the full posterior distribution of the uncertainties).
From Section 2. it is unclear to me, how a full posterior distribution of the climate is obtained within the presented modeling approach. Please see my detailed comment on Section 2.

See my response to that point there as well.

l.214 Five different input data files are compatible with crestr. However, most applications will only require two file (the df and PSE files, see below) to be created. More specific applications may require up to four of these files. All the files can be prepared outside the R environment and imported using standard R functions.
will only require two files

Corrected.

See l.204 – it becomes clear now, that this input data is actually the proxy data users aim to build their climate reconstruction upon. I would propose to state this explicitly – it might be obvious to the author, though for the reader it is not.

Now clarified, see above.

l.223 The proxy-species equivalency (PSE) table

I am little confused by this section. In Section 2. it is explained how the climate responses of different species of a given taxon are combined to the taxon's climate response. Hence, there are two levels involved. Now, it seems that the category 'taxon' was replaced by the category 'family' and on top of that a third level 'Genus' was added. So I wonder how, climate responses on the species level are first combined to a 'genus' level and then to a 'family' level? Maybe this confusion of mine is simply due to my lack of knowledge in this field. If you think, Sec 3.3.2 will be understood correctly by the relevant audience, then please ignore this comment.

I believe that palynologists in general would understand the distinction here. There are two levels of identification that matter for CREST: at the species level (to fit species pdfs) or not at the species level (to fit taxa pdfs). However, Reviewer 2 has also requested that I clarify some of these details at the beginning of section 2 where I present the species and taxa PDFs. I have explained this in greater detail there, and I hope the PSE table can now be better understood with these additional details.

Also, I would find it helpful to understand, what type of proxy data for past climate can actually be observed? Species? Or only taxa? Why and how does this differ between situations?

The level of identification of fossil proxies depends on many factors, including the type of proxies. For example, vegetation macro remains can most often be identified at the species level, while pollen grain cannot. Tree pollen grains can often be identified at the genus level, while grass pollen grains are most often identified at the family level. Other factors also influence this, including the palynologist (experience increases resolution), fossil preservation (the more damaged the fossil the harder it is to specifically identify it), and many more. Lots could be written on the topic and I believe this is standard knowledge for people using this type of data. As such, I will not expand on this in the manuscript.

l.316 – To estimate reliable PDFs, it is recommended to use at least 20 distinct occurrences for each species, but different values can be specified with the minGridCell parameter.

Here, I do not understand whether 'distinct' refers to the level of individual observations as stored in the DISTRIB table, or to the level of QDGC grouped observations?

To fit reliable *species* PDFs. This is now clarified.

l.362 4.3 Estimating the climate responses (the PDFs)

I recommend to call *'the PDF's'* consistently climate response functions throughout the entire manuscript. PDF is a specific type of function that fulfills certain requirements. As mentioned previously, the climate response functions are likelihood functions and if they weren't normalized they would not be pdfs. Also, the term 'climate response functions' describes more accurate the purpose of the functions.

What Reviewer 1 is referring to here is the reconstruction, which is not what this section is presenting. Here, we are presenting how the proxy-climate responses are fitted. The term PDF for this type of reconstruction was not pinned down by me, and I am simply using terms that are used in the community.

l.373 set geoWeighting to TRUE if the species PDFs of the different composing species should be weighted according to the square-root of the extent of their modern distribution.
This refers to combining the species' pdfs to the taxa pdfs?

Absolutely. I have now added a pointer to Eqt. 6 to make sure readers will be able to make the connection as well.

l.387 2) the climate values to reconstruct are likely to be in the study area (the reconstructions are bounded by the lowest and highest values observed in the modern climate space)
Can you think of a term other than 'study area'? This sounds more a like a geographical location. Or maybe you actually mean the geographical region. In that case, the above sentence should be specified, e.g.: the climate values to reconstruct are likely to be covered by present day climate values in the study area.

geographical location is exactly what it is. The term study area is now defined at the beginning of section 2 (along with presenting the types of data required), and I have fixed the sentence based on Reviewer 1's suggestion.

l.393 In our case study, the spatial variability represent true patterns in regional species diversity with the presence of several biodiversity hotspots across eastern and southern Africa (Myers et al., 2000).
represents

Corrected.

l.397 This diagnostic figure is also very important to identify potential local or global correlations between different climate variables and assess the risks of confounding variables (Juggins (2013), Chevalier et al. (2020b)).
What is the 'risk of confounding varaibles'? Please elaborate.

Now defined as follow: "*assessing the risks of confounding variables (_i.e._ variables correlated with important variables but do not directly impact the studied proxies; @Juggins2013, @Chevalier_etal_2020)*"

Fig.6 'Number of unique species occurrences' in other words means 'Number of different species observed at least once in a grid cell' is that correct?

Yes, it is. After consideration, the title was not changed.

Please give the different panel labels (a), (b),... and make sure that all axis have labels.

Thank you for noticing these missing labels. I have been so used to seeing these figures (and I have generated so many versions of them…) that I became a bit blind to some of these obvious issues. Labels are now added to the different axes of figures 6 and 7 (ant the package updated accordingly).

Fig.7 In the top row, the map represents the density of unique species occurrences per grid cell and the time series represents the variability of the taxon against time or depth.
If I am not completely mistaken, the map shows data from the QDGC_DISTRIB table, that is calibration data from present day observations, while the time series shows data from the proxy data from the sediment core. Please make sure to specify this unambiguously in the caption.

Corrected.

'taxaCharacteristics' graphical diagnostic tool to assess the sensitivity of taxa to climate. In the top row, the map represents the density of unique species occurrences per grid cell **derived from the modern calibration datase**t and the **time series represents the variability of the taxon against time or depth**. The bottom row is specific to each selected variable (only one row in this case), where the map represents the geographical distribution of the taxon (in white) against the climate background, the histogram represents the climate space (in colour) and how this climate is sampled by the taxon (in black) and finally the right plot represents the climate response of the taxon (in black) as well as the response of all the composing species (in grey). This diagram shows that Ericaceae is preferentially associated with colder temperatures in the study area. The red diamond indicates the location of the studied record.

Also, make sure that all axis are labeled and provide labels for different panels of the plot.

Corrected. See above.

The color bar is again labeled 'Number of unique species occurrences'. I assume the difference to Fig.6 is that here, the number of observed species belonging to the taxon 'Ericaceae' within a grid cell in shown, while Fig.6 shows the total number of observed species under consideration. Please make sure this difference is specified either in the caption or in the label of the colorbar itself.

I do not think users generating the plots with their own R code would be confused. Figure 7 is specifically generated for 'Ericaceae' (see code for plot_taxaCharacteristics), while figure 6 is designed for the calibration dataset as a whole.

The left panel on the bottom row is not very useful, since the information on the presence of observations hides the information on the climate. Maybe, you could simply sketch the outline of those connected patches (with some minimum size) which are covered by the taxon.

I disagree with this statement. This type of figure is probably the one that I use the most to better understand the sensitivity of my taxa. I also think that all my figures are complementary, and this one should be used in conjunction with the climate space plot (fig. 6). In addition, this plot becomes also quite interesting when more than one variable is selected (not shown here) because it allows seeing where the distribution is in relation to the underlying climate gradients. While trying to plot an outline of the distribution might seem like a good idea, in practice the data are often too disconnected to lead to anything visually convincing. It would only add more complexity, if not noise.

l.429 (all the probabilities sum to 1)
Maybe better: the PDFs integrate to 1.

While this is more correct, I prefer to keep the wording a bit more accessible to my target audience, which is not necessarily expected to be well-versed in the statistical language.

l.435 This type of representation can be particularly helpful to have objective interpretations of ecological changes from pollen diagrams (Chevalier et al., in press, Quick et al. (2021)).
I do not understand this sentence. Interpretations can never be objective.

'Objective interpretation' was a poor choice of words, indeed. Replaced with 'more informed interpretations'.

l.452 Here for instance, both Aizoaceae and Chenopodiaceae/Amaranthaceae were excluded because they are not primarily sensitive to temperature in southern Africa.
To understand this, it would be nice to have them included in the violin plot (Fig.8).

This exclusion is based on ecological considerations of the taxa, *i.e.* what is known about them. It is independent of the shape or look of the associated *pdf*. I think it might in fact be more misleading to plot them because they might not appear widely different from the others.

Fig.10 I can only guess, that the 64 in the upper right corner indicates the individual sample from the marine sediment core that is associated with a certain depth and that these kinds of plots can

be obtained for all individual samples? If not, and the plot refers to the entire reconstruction, then I do not understand how the weights can be specified, since early in the manuscript the authors explained that the weights of the different taxa in the climate reconstruction can vary over the depth of the core.

In this example, the '64' indicates the age of the sample. It is derived from the *df* dataset. I have now clarified in the caption what it means.

The black curve represents the posterior MAT reconstruction, from which a 'best' climate estimate can be estimated from the maximum of the curve and uncertainties derived by calculating the area under the curve.
From what I understood from Section 2., the black curve does not show a posterior probability distribution for the past climate but instead the likelihood function.

This has been corrected.

The line thickness can hardly be discriminated by eye. Did you try plotting the different climate responses according to their weights as expressed in Eq.(7) of the manuscript?

They were actually plotted according to the log of suggested weight, which I agree did not lead to the best results. I chose this option because using the absolute weights leads to unpleasant and very crowded graphics, as the lines can get very thick in some cases. I have tried different options and I am happy with 1.5*sqrt(weight). It gives a reasonable thickness range without overfilling the graphics. Considering that the taxon name information is also colour coded and included in the type of dashed line, I think the results are satisfying. See some examples below (age 64kyr and 9kyr)

[Figure]

[Figure]

l.501 The presence of multimodality can be the underlying cause of apparent noise in the reconstructions because minor changes in the taxa composition or percentages can force the system to oscillate between two maxima and thus 'appear' noisy, even if the background rate of change is minor.

We are not really confronted with an oscillating system here. Maybe rephrase: "The presence of multimodality in the climate reconstruction of subsequent samples can be the underlying cause of apparent noise in the reconstructions because minor changes in the taxa composition or percentages can easily switch the order of the two local maxima in terms of height. This results in a jumpy time series of the 'optimal climate reconstruction', even if the background rate of change is minor. "

Rephrased as follow: "*In this plot (Fig. \ref{fig:samplePDFs}), the PDFs of all the taxa present in the sample and selected to reconstruct the climate variable are represented along with the reconstruction. This type of plot can help identify if a particular PDF is at odds with the general assemblage, which usually indicate the possible presence of a confounding factor. It is also helpful to visualise the full spread of the uncertainties and, by extension, highlight that reconstructions can be multimodal*. **While it is not the case in the example here, multimodality can be the underlying cause of apparent noise in the reconstructions with minor changes in the taxa composition or percentages forcing the system to oscillate between two maxima and thus 'appear' noisy.** *This effect could also be seen from the reconstruction plot where the full uncertainties would be represented (`simplify=FALSE`).*"

Fig. 11 Why are all anomalies positive? Is this by coincidence and due to the choice of taxa presented in the plot?

This is due to a lack of clarity of the caption. The plot represents the absolute anomalies with the bars, and the anomalies' sign is colour-coded. This was designed to save vertical space. As a result, the plot represents both positive and negative LOO anomalies. I have now modified the figure with a legend that specifies what the colour means. I have also clarified it in the caption.

*Leave-one-out (LOO) graphical diagnostic tool to illustrate the influence of different taxa on the reconstructions. Here, the results are only shown for a subset of the taxa observed in marine core MD96-2048 (only 20 out of the 171 available taxa are represented). The height of each bar represents the absolute effect (in °C) of removing the taxon from the reconstruction, and the sign of this effect (increase or decrease of the reconstructed temperature) is colour-coded. Here, blue (cf. Ericaceae) and red (cf. Combretaceae-type) taxa are cold and warm indicators, respectively.*

Here, the results are only showed for a subset of the taxa observed in marine core MD96-2048 (only 20 out of the 171 available taxa are represented).
*'Showed'* should be shown.

Corrected.

The title of the plot 'mean annual temperature' is not very suited. As far as I understood, the plot shows the anomalies for reconstructions that are based on all but one taxa with respect to the reconstruction based on the full set of taxa.

I have adapted the package so that the name of the plots will now be 'Leave-one-out anomalies for [studied variable (variable unit)]'.

---

## Author Comment (AC2)

Dear Prof. Bartlein,

Thank you very much for your input on my manuscript. I think the suggestions were really 'on-point' and I do feel that my manuscript has become clearer, better-organised and more engaging thanks to your inputs.

The comments of Reviewer 2 are in black, and my responses in blue. Italicised texts are quotes from the revised manuscript.

General comments:

This paper introduces an R package, crestr, for implementing a probability-density- function (PDF) variant of the Mutual Climatic Range (MCR) approach for making climate reconstructions from, for example, fossil-pollen data. The paper is not simply a user manual or vignette, but also discusses some of the conceptual underpinnings of the whole approach, and the philosophy behind some of the methodological choices that necessarily have to be made.

I generally prefer to see the pdf methods and MCR as both variants of a broader family of 'Indicator species' approaches. The pdf methods and MCR rely on distinct data modelling hypotheses, and the pdf methods are not a more complex version of MCR. However, I agree that the two types of methods are closer to each other than to other existing techniques, such as WA-PLS, MAT or Inverse modelling. I appreciate that Reviewer 2 appreciated the extra efforts put into re-explaining the different aspects of the method in the light of the package – even if some of these parts were a bit imprecise.

There are a few reorganization or further-explanation issues that need to be sorted out. For example, the motivation for producing the R package in the first place doesn't appear until Section 5 of the paper, and the particular reconstruction approach that is implemented here should be mentioned first among the list of various approaches mentioned in the introductory paragraph. Otherwise, the basic approach, and its implementation using the package is laid out nicely.

This comment is similar to a suggestion made by Reviewer 1. The introduction has been reworked to make it more package oriented. And all the types of data used by the package are also described at the beginning of section 2 (i.e. before describing the mathematics of the method) to avoid any misunderstanding. Overall, many more details have been added everywhere across the manuscript.

One important contribution of the paper is the release of the global data sets, both taxonomic and climatic, that can be used to apply the approach generally. However, there is a tension between attempting to reconstruct as many environmental variables as one can get into a database and reconstructing only those that can be mechanistically related to, say, terrestrial vegetation, which typically are simply growing season warmth, winter cold, and moisture stress. Those that argue for the former approach argue that the environmental variables are all related one way or another, so if you can reconstruct one, you can reconstruct all, while those that argue for the latter approach point out that assumption is nonsense. Likewise, there are

probably taxa included in the data base that are completely insensitive to the macroclimatic variables provided, but may wind up contributing to the reconstructions when they provide little real information. From a purely statistical perspective, overfitting is the issue here. I know users don't necessarily have to attempt to reconstruct all of the environmental variables in Tables 1 or 2, and that they can use their own data, and manage the particular variables or taxa that are used, but I think that providing so many variables creates an "attractive nuisance". So, I think it would be good to caution the users about these issues. No good deed goes unpunished.

This is something that I have also thought a lot about. I agree that plants respond to growing season warmth, winter cold, and moisture stress, especially in the mid to high latitudes. However, important climate drivers are different in tropical and subtropical regions, where the seasonality of precipitation and associated seasonal temperatures are key in defining vegetation. I can easily envision a climate zone and vegetation type where most of the variables included in the package could be important to look at. In my opinion, the only ones that are probably not decisive factors relate to diurnal or annual temperature variability (e.g. Bio2, bio3, bio4 or bio7). However, I think it is suitable for users to have access to more variables than could/should be reconstructed to understand their study area's climate regime better. In addition, the package is not designed for vegetation only, and these variables could be of importance for the other proxies that are part of the calibration dataset (rodents, beetles, insects). The list of marine variables in coordination with an oceanographer with experience in proxy-based reconstructions and the decision was made to use similar variables to those released with modern sample databases (de Vernal et al., 2020). In summary; I do not think there is a need to restrict the list of available terrestrial or marine variables. Still, I agree that it is important to add an explicit warning that variable selection should be made carefully and that running serial reconstruction is a generally bad idea.

In section 3 when introducing the calibration dataset as a whole: "*In the _gbif4crest_ database, all the QDGC grid cells were associated with a collection of terrestrial and oceanic environmental variables that can be reconstructed (@Fick_Hijmans_2017, @Zomer2008, @WOA_2018_temp, @WOA_2018_salinity, @WOA_2018_oxy, @WOA_2018_nutrients, @Reynolds2007, see details in Tables \ref{table:variables-terr} and \ref{table:variables-mari}). **Despite the diversity of variables available, it is recommended to avoid serial reconstructions and, on the contrary, to identify the few important variables for the studied palaeoecological datasets a priori.** *"

In section 5 when introducing the variable selection in the package: "***However, serial reconstructions should be avoided, even if many variables are provided with this package. Careful interpretations of the fossil data should be done before selecting variables**.*"

Similarly, I cannot curate the database to exclude all the species that might not be responsive to the selected variables. This is unrealistic and subject to personal interpretations anyway. The package contains explicit tools to determine which taxa should be used to reconstruct which variables (fully explained in section 5), and I believe this to be the optimal way of dealing with the problem.

Another issue that might be discussed a little is the "no analogue" one. Although usually raised in the context of modern analogue technique (MAT) approaches, it applies here too, as illustrated by Fig. 10, where the reconstruction lies in sort of trough of individual PDFs, and it's also probably the case that some PDFs don't overlap at all. If I understand this correctly, all of taxa with PDFs that appear in the figure co-occurred in the sample, but they don't today (otherwise their PDFs would overlap). This deserves a sentence or two of discussion, perhaps by handing it off to other papers.

I both agree and disagree with this comment. One reason the no-analogue situation is mainly associated with MAT is that MAT works with entire pollen assemblages and includes the relative percentages in the estimation of analogues. Techniques that rely on the individual responses of the pollen taxa are, on the contrary, a lot more resilient to the problem because they do not require the fossil assemblage to exist in the modern landscape. They only require the individual climate responses to do so. In the example represented in Fig. 10, 25 of the observed taxa were used in the reconstruction. 24 of the pdfs overlap between roughly 17 and 20 degrees (see the modified version of Fig. 10 below). Only one taxon (*Melochia,* in purple) stands on the side. This is a very mild no analogue case, especially considering the size of the catchment of the studied record (the catchment of the Limpopo River). And technically, all the pdfs are defined for any possible climate value, as opposed to MCR for instance, where the minimum and maximum climate values must be estimated. As such, the pdfs always overlap, although with a very low probability in this case for *Melochia*.

In addition, one might argue that all the samples from this core do not have modern vegetation analogues since they are derived from a marine core – there is nowhere across the Limpopo Basin where such vegetation composition can be observed, either today or in the past The strength of the approach is to consider each taxon independently of the others and to return an estimate of what climate values are the most likely considering the presence of a range of taxa. It does not require the taxa to be found in the same location in modern environments. However, I think it might be a nice idea to include something along the line of 'limited overlapping between pdfs' in future versions of the package and the generation of the figure itself. But this would be threshold dependent (*e.g.* do the 99% ranges of the pdfs overlap?), and the next question would immediately be what is a good threshold to define a no analogue situation? And if two pdfs do not overlap for temperature, can we still use them for precipitation or moisture? All these questions highlight that the nature of the problem is very different in the context of indicator species methods [no analogue defined from modelled climate responses] than it is from MAT [no analogue directly defined from the vegetation data themselves].

[Figure]

One stylistic thing about the paper is the sometimes jarring transitions between text and code blocks. Starting out, there are transitions like "...similar results would be obtained using the following command: (followed by the code block)", but that format gets abandoned later in the manuscript. I know from experience that Copernicus journals' choice of a type face for code makes it appear pretty clunky, and sometimes unreadable, which makes setting it off more important for readability in the two-column paper format.

*I have homogenised these transitions from text to code and from text to equations, and I have made sure that the width of each text block is not larger than one column in a two-column formatted paper format.*

Specific and technical comments:

line 2: "the methods ... are powerful at producing robust results..." I'm not sure that robustness in the usual statistical sense is either evaluated or demonstrated in this paper.

*The use of the word 'robust' seems to have bothered Reviewer 2 (this comment and about three more later). I am struggling a bit to understand where this resistance to the term comes from. But being pragmatic with this issue, I am assuming that if it bothered Reviewer 2, it is likely that it will also bother other readers. I have therefore removed all the mentions of robust and robustness from the manuscript.*

line 3: Not parallel: accessing/curating/the complexity" (action/action/characteristic). The sentence could be made parallel by rewording: "The problem of accessing and curating the necessary calibration data and the complexity of interpretation..."

*I have reworked the abstract and it is now phrased as follows: "However, the difficulty of accessing and curating these calibration data and the complexity of interpreting probabilistic results often limit their use in palaeoclimatological studies."*

line 17: "climate drivers" Meaning the climatic controls of the variations in the fossil data, or the controls of the climatic variations themselves?

Rephrased as 'The drivers of climate change'.

Line 19: "climate reconstructions"?

Text amended.

line 23: "robust" Again, I don't think this is the right word. I think that "robustness" is a property of a statistic (e.g. the median) that signals that it will perform well at estimating, in this case, location, no matter what the underlying distribution of the data looks like. I don't think this notion really applies to a dataset. Maybe "extensive datasets"?

I think it does apply, but see my generic response above. Now rephrased as: "However, the limited availability **of the necessary calibration datasets beyond** the Northern Hemisphere extratropics has often hindered their application in many environments and regions where quantified climate records are needed, despite the existence of suitable fossil records."

line 25: Before going further, it would be good to alert the reader as to which particular approach this paper implements (i.e. as in Section 5.1.2 of Chevalier et al. 2020), and a little about how it works. (It seems it's NOT WA, WA-PLS, MAT, etc., but what is it?). I don't think it would inappropriate at all (in terms of self-citation padding) to use Chevalier et al. 2020 a little more for background.

I agree that the introduction needed to be boosted by a bit more context and clarity in terms of what this paper presents. I have reworked large parts of it, and I hope things are now clearer. The second paragraph introduces CREST and broadly speaking how it works. And then I introduce why a new R package was needed.

line 26: By "un-quantified fossil pollen records" do you mean that the data are "qualitative" as opposed to "quantitative" or simply that quantitative reconstructions have yet be made? Same issue on line 35.

This sentence has been removed from the reworked introduction.

line 27: "built upon the"

Text amended.

line 28: "modern proxy geolocalised occurrence data". I don't know what that means.

Now defined as follow: *"the 'Indicator species' family of reconstruction techniques uses modern proxy occurrences (_i.e._ **collections of locations where the studied proxy species can be observed in modern environments**)."*

line 39: "estimates, weights and propagates…" I'm confused by the "propagates" idea.

The sentence has been simplified to: *"While techniques such as MAT or WA-PLS are primarily designed to associate modern proxy observations with their 'most likely' or 'mean' climate values only, **CREST estimates and weighs all the climate values that are compatible with the observed fossil data**. As such, the climate reconstructions obtained from CREST can be understood as an ensemble of all data-compatible climate values and not a simpler, less informative 'most likely' or 'best' climate estimate with statistical errors"*

line 40: "posterior climate reconstructions" I'm not exactly sure what's Bayesian about this.

Nothing, indeed. The word posterior was improperly used in this manuscript and has been removed.

line 41: "weighted ensemble" An ensemble in this context is a collection or a group of something. Do you mean a "weighted average of an ensemble of estimates" or "an ensemble average"?

The sentence has been simplified to: *"While techniques such as MAT or WA-PLS are primarily designed to associate modern proxy observations with their 'most likely' or 'mean' climate values only, **CREST estimates and weighs all the climate values that are compatible with the observed fossil data. As such, the climate reconstructions obtained from CREST can be understood as an ensemble of all data-compatible climate values** and not a simpler, less informative 'most likely' or 'best' climate estimate with statistical errors"*

line 46: "can model" Does this mean "can represent"?

I used 'can estimate' instead.

line 55: "embedded" Replace with "accompanying"?

Accompanying suggests two distinct elements, e.g. 'the dataset accompanying the package'. In the present case, the package and the dataset are only one thing, and I think 'embedded' is more adapted based on definition 2 from https://www.dictionary.com/browse/embedded).

Figure 1 caption: Describe the Full (or "true") distribution first.

Done.

line 61-70. Repeated material.

Corrected.

line 72: "PDFs are used to transform the information". Transform how? Doesn't the PDF (or histogram) of a particular taxon just represent the frequency of occurrence of a taxon along a climatic gradient?

The PDFs do not transform the information; they are the result of the transformation. Now clarified: "The transformation of the information contained in the modern observations of the biological climate proxies into probabilistic climate responses can be done in one or two steps depending on the nature and taxonomic resolution of the studied proxy.".

line 73: "weighted ensemble" again.

Removed.

line 77: It might be good to remind the reader that, for example, some pollen taxa represent individual species while other only genera (or indistinguishable types, e.g. Larix and Pseudotsuga). You could refer to section 3.3.2 for a discussion of how the species-to- taxon translation is made.

I have added a bit more explanation on that at the beginning of section 2.

> CREST takes into account that some fossil taxa can be identified at the species level (_e.g._ plant macrofossils), while others can only be identified at a lower taxonomic resolution (_e.g._ fossil pollen are commonly identified at the genus, sub-family, or family level; @Chevalier_etal_2020). The transformation of the information contained in the modern observations of the biological climate proxies into probabilistic climate responses is thus done in one or two steps depending on the taxonomic resolution of the studied proxy. When the observed fossil taxa are identified at the species level, determining a list of species that could have produced that fossil is, however, necessary (see Section 3.3.2 to know how to format this information for _crestr_).

Line 78: "empirical mean and associated variance" Isn't this step in fact adopting the approach you reject in panel (a) in Fig. 1, especially when a taxon might have a nonsymmetric distribution?

In Fig. 1, I present climate reconstructions, which can (and should) be multimodal and irregular if dictated by the data. On the contrary, species responses to climate are much more regular. If a species response along a climate gradient is irregular, it is most likely because they do not respond to the specified variable (and shouldn't be considered) or because another confounding variable is excluding its presence in places. It is unreasonable to assume that a plant species (not a pollen taxon!) might like living under 15degrees, not like it at 20, and like it again at 25. If a species is excluded from regions with 20degree it is either because another

factor is excluding it (e.g. different rainfall patterns or humidity) or because 20degrees is largely absent from the study area due to, for instance, incomplete data sampling. In both cases, it is not reasonable to assume it cannot grow under 20 degrees. In addition, CREST allows for asymmetry (right-skew) with the lognormal shape. It simply does not allow multimodality for monospecific responses, which is based on ecological considerations. When the monospecific responses are combined, multimodality and all sorts of irregularities become possible and be propagated to the climate reconstruction. I have made this more evident in Section 2.1.1.

*"Contrary to the previous step, no additional constraints are added here. The distribution of the PDFtx(t,c) can thus be asymmetrical and even multimodal if different (groups of) composing species exhibit distinct climate requirements."*

line 85: Right. And see also Liu et al., 2020, Proceedings of the Royal Society A https://doi.org/10.1098/rspa.2020.0346 and the fxTWAPLS package.

This is an interesting study that I had missed. However, I do not see what it would add here, since it is a study fully focused on WA-PLS.

page 5: Some things missing here. What exactly is k()? What does i index? N vs N-sub- s?

All these terms have now been properly defined in the text in a more logical order (see page 5 of the revised manuscript).

line 100: I think this combination-of-PDFs needs to be better explained. Would one want to ever lump all of the species of, say, Pinus, into one taxon? Some paleoecological data sources (e.g. pollen) are pretty "blunt" taxonomically, whereas others (e.g. plant macrofossils) are usually identified at the species level.

Users determine the type of combination based on their fossil identification. Only the first step is used if the fossil is identified at the species level. If it is identified at any higher taxonomic level, then species pdfs are fitted for each composing species, and these species pdfs are subsequently combined into a taxon pdf. If the fossil pollen is identified as Pinus, then yes, all the Pinus species found in the user-designed study area should be lumped together because they are all considered are likely to have produced the observed pollen grain. I think the miscomprehension here is linked to the remark made about line 77 above. I have clarified the distinction at the beginning of the section and hopefully, this clarifies the two steps.

line 105: "grid cells" This is the first mention of grids/gridding. Does that have something to do with "geolocalized occurrence data"?

Replaced with 'occurrences'.

lines 105-107: Again, I'm worried about "robustness" which I think is being used more to denote some notion of reliability than in its usual sense. If robustness of the location and scale parameters (eqn. 1 and 2) is really a concern, why not use robust estimators of them?

See above.

line 117: "These definitions of sensitive taxa are always specific to a specific region...". Too many specifics. Maybe "the definitions are specific to particular regions"?

The sentence was removed.

line 119: So the estimate is basically a weighted geometric mean of the individual taxon PDFs?

Absolutely.

line 135: "own design"?

Corrected. *"[…] users can also design their own weights to better account for the specificity of their data."*

line 139: It might be good to cite to some of the discussions on fitting SDMs with presence-only data. (See Valavi et al., 2022, Ecol. Mono. https://doi.org/10.1002/ecm.1486)

I have now clarified why I do not use these more complex models. CREST is designed to be usable in data-sparse regions, and usually, the most complex models require a lot more data than is commonly available. As such, the model that was designed is simpler in its assumptions and might lead to suboptimal results in regions with high data density, but its strength is its capability to produce reliable results everywhere.

*"Because CREST aims to be applicable even in data-sparse environments, the estimation of these responses is based on simple assumptions that exclude using complex algorithms, such as those described in, for instance, the recent review of @Valavi2021."*

line 148: "upscaled at a spatial 150 resolution"?

This was a typo. '150' has been removed.

Figure 4: Increase the spacing between the bottom two panels to match the others?

The last two panels are actually the same table, hence the different spacing. I have clarified it in the caption.

Figure 5: Nice figure. The resolution of the image in the discussion .pdf obscures the double-framed boxes, however.

I have increased the spacing a bit to avoid this type of issue in the final version.

Section 3.2: In a "real-world" example, how large might a crestObj become?

For example, the crestObj created by the application presented in the paper is 28.5Mb once exported as an RData file. This object includes all the calibration data, modelling steps and results. The size will vary with the diversity of the calibration data and I would expect the object to grow up to 100-200Mb for pollen studies in Europe or North America.

line 236: Hyphenate "species-to-proxy".

Corrected.

line 256: Are the weights completely arbitrary, or should the lie within some particular ranges (e.g. 0-1 or 0-100)?

Thank you for noticing this important omission. The weights must be between 0 and 100, and this is now clarified in the manuscript.

Section 3.3.4: Describe how the points in the climate-space data frame are associated with the taxon distribution data? Should there be a one-to-one correspondence between the rows?

The table is a standard XYZ matrix. Each row corresponds to a location, and a location is characterised with a longitude, a latitude and one or more climate values. This table is never connected with the distribution table presented in section 3.3.3. These data are used 1/ to make plots such as figures 6 and 7, and 2/ to weigh the climate space through equation 3. This is now clarified.

line 285: "the original CREST software" This raises the question "Why not just use the original CREST software?" which is answered on lines 578-579. I think that the motivation for the development of the R package should be moved up to the introduction.

This element of 'discussion' has now been moved to the introduction.

line 303: "the three input files" Are these files available anywhere? I don't see them in the GitHub repository.

These files were provided as supplementary material to the paper. This is now clarified in the manuscript.

line 320: Should there be some kind of transition between the text and code block? (As on line 349).

As mentioned at the beginning, the text-to-equation and text-to-code transitions have been smoothed and homogenised across the manuscript.

line 364: "different parameters" -- "different parameters that control the reconstruction"?

Corrected.

line 374: Again, some kind of transition is needed.

Corrected.

line 387: "the climate values to reconstruct are likely to be in the study area" Does this mean that the range(s) of climate values in the calibration data set should have the same (or larger) amplitude?

Only values present in the calibration dataset can be reconstructed. This is true for CREST and for any statistical reconstruction technique. Since all methods tend to shrink the reconstructed ranges, a modern range larger than the (unknown) range of past climate change will lead to better results. Rephased as follow: *"In every study involving estimating relationships between biological entities and environmental parameters, the first step is always to ensure that the defined study area and associated calibration dataset are as coherent as possible.* **This includes ensuring that** *1) all the essential taxa are present in the study area, and their distribution is not truncated, 2)* **the climate values to reconstruct are likely to be covered by present-day climate values (the reconstructions are bounded by the lowest and highest values observed in the modern climate space)** *and 3) there is no large sampling or representativity bias (_e.g._ along country borders due to different sampling efforts). The 'climateSpace' graphical diagnostic tool (Fig. \ref{fig:climatespace}) was designed for a rapid assessment of all these characteristics."*

line 390: "homogeneous" In geographical space? Climate space? Both would be good I guess.

Rephrased as: *"Ideally, the climate sampling should be as homogeneous as possible to ensure proper sampling of all the possible climate values, even if the extreme climate values will always be under-represented compared to the median ones. "*.

line 428: "violin plots" Violin plots are cool looking, but they are affected by our tendency to misinterpret/misjudge areas (i.e. Cleveland, W.S., 1993, Visualizing Data). I think viewers tend to notice the blobs as opposed to the profile, which is the important information. I think simply plotting the PDFs as in Fig. 6 of Quick et al. (2021) is more effective.

I generally agree that violin plots can induce interpretation biases; However, I would imagine that is only a problem if the figure cannot be looked at thoroughly for an extended period of time (e.g. during a talk). On the contrary, I would imagine that users who make the plot are actually interested in spending a bit of time on it to better understand their data. And the areas of all the violins are the same in this plot. Therefore, what remains is the shape of the violins, which I what people should be looking at.

I must also confess certain laziness to dive back into the code of the figures to make the necessary changes. Writing scripts that are robust (;-)) to a wide range of data is really difficult. I think the figure already does a good job and I am not convinced that the suggested benefits are worth the additional coding work. I am nevertheless taking note of this suggestion and will re-evaluate it down the road if I decide to update the diagnostic tools.

line 435: "to have objective" -- "to make objective"?

Corrected.

line 460: "to reduce the noise" One pollen type's noise may be another's signal. Pollen types have long-tailed distributions, and so an alternative approach might be to transform the data, with the square-root transformation in particular having some desirable properties.

I generally agree with this statement. I am personally not a user of these hard presence/absence thresholds. I prefer using my weighting function that will automatically consider 2% as important if the average percentage is relatively low and not important for taxa like Poaceae or Pinus that usually have much larger percentages. I have this option because I know it has been used in the past. The square-root transformation is very useful to compare assemblages to one another (such as in MAT). I think its usefulness would be more limited for indicator species. The option is not included in the package, but as now mentioned in the manuscript, users can square root transform their data and feed these data to the algorithm.

*"The data can be directly weighted by the values provided in `df`, which implies that users can define their own specific weighting strategy (**e.g. using the square-root transformation of the pollen percentages).**"*

Figure 10: "the thickness of the lines" I don't see much differentiation in line thickness.

I have tried different options and I am now assigning a thickness based on 1.5*sqrt(weight). It gives a reasonable thickness range without overfilling the graphics. Considering that the taxon name information is also colour coded and included in the type of dashed line, I think the results are satisfying. See some examples below:

[Figure]

line 505: "full uncertainties" Could a second panel be added to Fig. 9 to show that?

This is what is represented. The alternative is to plot, for instance, only the 90% range instead of the coloured background. I am not sure adding this to the paper would add much. In addition, this type of plot is illustrated in the get-started vignette that Reviewer2 extracted and played with (see below).

Figure 11: Why are all the values positive? What is the "net effect"?

The absolute value is plotted (what I inappropriately called the 'net' effect), and the sign of the difference is colour coded. This was designed to save vertical space on the figure. I have now modified the figure with a legend that specifies what the colour means. I have also rephrased it as follows to increase clarity:

*Leave-one-out (LOO) graphical diagnostic tool to illustrate the influence of different taxa on the reconstructions. Here, the results are only shown for a subset of the taxa observed in marine core MD96-2048 (only 20 out of the 171 available taxa are represented). The height of each bar represents the absolute effect (in °C) of removing the taxon from the reconstruction, and the sign of this effect (increase or decrease of the reconstructed temperature) is colour-coded. Here, blue (cf. Ericaceae) and red (cf. Combretaceae-type) taxa are cold and warm indicators, respectively.*

Code:

I was able to install the package and run the example with only a few issues. The code in the get-started.html vignette and the GitHub README.md is a little different. I was able to "purl" the get-started.Rmd R Markdown file without an issue to reproduce the example. It wouldn't hurt to also provide a pure get-started.R file.

This is an excellent suggestion. The file is now available on the webpage.

The results of the example wound up in an obscure temporary file. Adding "file.path(tempdir())" to the example would help the user to find that folder.

This is a requirement of CRAN – all files created in the vignettes must be saved in tempdir(). Users can however specify where they want to save their files. I added a note.

Pat Bartlein

References cited:

de Vernal, A., Radi, T., Zaragosi, S., Van Nieuwenhove, N., Rochon, A., Allan, E., De Schepper, S., Eynaud, F., Head, M. J., Limoges, A., Londeix, L., Marret, F., Matthiessen, J., Penaud, A., Pospelova, V., Price, A. and Richerol, T.: Distribution of common modern dinoflagellate cyst taxa in surface sediments of the Northern Hemisphere in relation to environmental parameters: The new n=1968 database, Marine Micropaleontology, 159(November 2019), 101796, doi:10.1016/j.marmicro.2019.101796, 2020.

---

## Referee Report (RR1)

**General comments:**

The author has presented a revised version of the manuscript with substantially improved readability and many inaccuracies being cleared out with respect to the original version. It would like to point out again, that probably the lion's share of the work has gone into the code of the R-package and the curation of the database and that this work is of highest value for the community.

On the other hand, I must admit that I am still not convinced by the presentation of the mathematics forming the bases of the climate reconstruction method. This criticism comprises twofold: First, the introduction of Equations (1-5) and (7) suffers from several mathematical inaccuracies – however, I noticed that these in parts trace back to the article presented by Kühl et al. (2002). Since the derivation of these equations is by no means the central aim of this paper, these inaccuracies might be acceptable, given that these equations appear to be correct. Also, it seems that some terms have established as domain specific language and thus may be clearer to the target audience than they are to me. Second, I believe that Equation (6) is in fact not correct, even though its use might generate reasonable results. I have expressed my concerns about Eq. (6) already in my previous review.

If not for this manuscript, I am convinced that putting the probabilistic climate reconstruction method on solid ground mathematically would be a beneficial task for the future. I have attached a pdf that outlines the derivation of Eq.(5) starting from Bayes theorem.

The remainder of the mansucript gives the reader a good overview of the *crest* R package, in terms of its capabilities, requirements and usage. It is well structured and the final example of application really takes the reader / user by the hand.

**Specific comments:**

l.2    In particular, the methods based on probability density functions (or PDFs) can be used in various environments and with different climate proxies because they rely on elementary calibration data (i.e. modern geolocalised presence data).

I would replace 'the methods' by 'methods'. Maybe 'methods based on probability density functions' are just 'probabilistic methods'.

l.14    It is hoped that crestr will be used to produce the much-needed quantified records from the many regions where climate reconstructions are currently lacking, despite the availability of suitable fossil records.

What is meant 'quantified records'? In my understanding a record is a directly measured time series – so a data processing software could not be used to 'produce a record'? Do you mean 'reconstruction'?

l.15    no paragraph in the abstract

l.19    Over the years, numerous techniques of increasing complexity have been proposed, each one based on a unique set of assumptions regarding the modelling of ecological datasets and their translation into climate reconstructions (e.g. Birks et al. (2010), Chevalier et al. (2020b)).

I assume that with 'ecological datasets' your refer to observations. In that case I would say that observations are not being modeled. Of course, sometimes one uses models to draw inference from datasets and probably that is what you mean?

l.26    ...and their accessibility with multiple software solutions.

Saying that an analysis technique is 'accessible with multiple software solutions' sounds strange to me. Do you mean, there exist relatively simple software implementations of the techniques?

l.26    However, the limited availability of the necessary calibration datasets beyond the Northern Hemisphere extratropics has often hindered their application in many environments and regions where quantified climate records are needed, despite the existence of suitable fossil records (Chevalier et al., 2020b).

Again, what are quantified climate records? It seems you mean reconstruction – in my understanding records are not reconstructions.

l.32    Because modern occurrence data are generally easier to obtain than modern proxy assemblages, this fundamental difference implies that Indicator species methods can contribute to filling in the reconstruction gaps that exist at the global scale.

Grammar: Modern occurrence data are generally easier to obtain than modern proxy assemblages. This fundamental difference implies that Indicator species methods can contribute to filling in the reconstruction gaps that exist at the global scale.
For the non-paleoecologists: what's the difference between proxy assemblages and proxy occurrence data?

l.36    Derived from the original work of Kühl et al. (2002) —— who

It seems there is an extra hyphen in the pdf.

l.37    CREST estimates and combines probabilistic proxy-climate relationships to reconstruct past climate parameters from fossil proxy observations.

Maybe you could add: CREST estimates and combines probabilistic proxy-climate relationships **from modern occurrence data** to reconstruct past climate **variables** from fossil proxy observations.

l.45    However, the complexity of collating and formatting the thousands of distinct occurrences required to estimate reliable PDFs limited its practical use.

I understand, that in your context the term PDF carries a very specific meaning. However, in general, this is not the case and a reader that is not used to the specific use of the term PDF as a synonym to your 'climate response functions' will probably struggle to understand the above statement. Also, you introduced the abbreviation PDF only in the abstract, but up to this point not in the main text.

l.48    'climate records'

l.50    This paper thus introduces  the new multi-platform R package crestr designed to replace the original interface.

l.51    crestr includes the global calibration dataset

It is linked to the dataset but does not include it – strictly spoken.

l.59    As such, the climate reconstructions obtained from CREST can be understood as an ensemble of all data-compatible climate values

Maybe 'As such, the application of CREST yields a probabilistic quantification of the past climate in view of the data under study as opposed to simpler, less informative 'most likely' or 'best' climate estimates. While the latter may only capture statistical uncertainties, the former rigorously takes into account the large quantitative uncertainties inherent to analysis of this kind.'

Just to highlight a little bit the fact, that in terms of uncertainty propagation *crest* performs a lot better than the mentioned 'best-estimate' methods.

fig.1    Conceptual illustration of the differences between a modelling approach based on the estimation of the full spread of the data with the probabilities spread along the climate gradient (e.g. CREST; dark grey), and a modelling approach focused on the estimation of the 'most likely' or 'best' climate value with small statistical errors surrounding it (e.g. MAT or WA-PLS; light grey).

I would say the probabilistic approach is based on the 'full spread of the data' but not on the 'estimation of the full spread of the data'.

l.96    The **individual?** climate responses of all the species identified are estimated as univariate probability density functions (PDFs) for every climate variable.

l.99    The individual species'

l.99    estimation of the empirical mean ($m_{s,c}$)

An empirical mean is not estimated, but computed from the data. Just delete 'the estimation of'.

l.107    Here, the weights are calculated by first sorting the N climate values (all the $c_i$) that compose the modern climate space into bins of equal width (e.g. $2 \circ$ C or 50 mm). Then, each climate value $c_i$ is given a weight $k(c_i)$ defined as the inverse of the relative size of the bin $c_i$ it belongs to:

1. As far as I understand N is the number of gridded observations of the modern climate variable c, while $N_s$ is the number occurrences of the species S. According to line 100, the $c_i$ are only those climate values, that coincide with the occurrence of the species. However, for the weights $k(c_i)$, the entire climate space should be taken into account. So the addition (all the

c_i) is not correct. Maybe, the difference between the c_i and **all** climate values form the study area can be clarified somehow.

2. The first sentence defines all bins such that they have equal width. The second sentence refers to different 'sizes' of the bins. I assume that the 'size' of a bin here means the number of observations falling into one bin. However, typically one would use 'size' and 'width' interchangeably, so 'size' might be misleading, here.

l.117 Once estimated, m s,c and s 2 s,c are used to define a regular, unimodal distribution for the PDF sp (s, c) of species s for climate variable c. Here, we assume that the shape of these species responses should be unimodal and can be either normal:

I know it is a detail, but the term 'distribution' is slightly misused in this context. The distribution of a random variable in statistics defines the the probability for the random variable to assume a certain value. Hence, here the PDF **is** a distribution.

Maybe: 'Assuming unimodality and either normality or log-normality, the estimated $m_{(s,c)}$ and $s^2_{(s,c)}$ are used to define the species climate response PDF's as follows: …'

l.120 delete paragraph

l.125 I have expressed my doubts about this equation already in my previous review. After going through the math once again, I still belief this equation is not correct – even though it might lead to reasonable results. Also in the cited literature, I could not find a convenient derivation of Eq.(6).

l.126 delete paragraph

l.138 Climate c is reconstructed from fossil sample z (z can be an age, depth or any identifyer) by multiplying the PDF tx (t, c) of the T (z) selected taxa:

- Maybe 'Past climate c that corresponds to a specific age (or depth) z from which a fossil sample is available can be reconstructed…'
- you first assign z := the fossil sample and then state that either z:= age, or z:= depth.
  - what exactly is 'the fossil sample'? A dataset comprised of abundance data of different taxa on a depth or age axis? This is a little inaccurate.

l.141 delete paragraph

l.151 The summation index should be 'i' or sth else and then the depths / ages should be discretized as $z_i$, but the summation index cannot be a continuous variable. The inner parenthesis in the denominator are not required.

l.152 delete paragraph

l.181 In the gbif4crest database, all the QDGC grid cells were associated with a collection of terrestrial and oceanic environmental variables that can be reconstructed (Fick and Hijmans (2017), Zomer et al. (2008), Locarnini et al. (2019), Zweng et al. (2018), Garcia et al. (2019a), Garcia et al. (2019b), Reynolds et al. (2007), see details in Tables 1 and 2).

Does this mean, that all variables listed in Tab.1 and Tab.2 are reconstructable? If so, I would suggest to express this a bit more clearly and also emphasize this in the table captions.

l.193 For example, the first version of the gbif4crest dataset released in 2018 contained about 17.5 million QDGC entries, while the new version contains approximately 25.3 million entries (~44% increase).

Why QDGC entries and not only entries? The second is a duplication of the first sentence of the paragraph and can be deleted, except the (~44% increase).

l.230 Maybe, it would be helpful to explain, which function call initializes the crestObj in first place already at this stage?

l.257 Does the df have as many columns (+1 for the depth/ age) as fossil taxa are considered for the climate reconstruction?

l.280 In a second time

In a second step

l.281 Mabe here, a sentence like

*The pdf for the fossil taxon Stoebe-type is thus comprised of the linear combination of species pdfs according to equation (6) associated with all species that fall into the geni Stoebe and Elytropappus.*

would be helpful at this stage, provided that my interpretation is correct.

l.284 Additional taxa can also be added to the PSE file to exclude species known not to be part of a group. For instance, this 'trick' could have been used to simplify the climate response of the 'Asteraceae undiff.' group by excluding more species from it, even if the pollen grains corresponding to these species have not been observed.

I would suggest to move the 'even if… ' to the first sentence.

l.290 If I understand correctly, if I provide a 'distributions' table as an input to the get_modern_data() function, then I do not need the PSE is that correct? Why is the 'distributions' table not listed in Fig. 5 in the 'input' category? If users decide to use the 'distributions' table as input, then they have to provide climate_space data frame as well, is that correct?

l.392 here called rcnstrctn; Fig. 5

here called rcnstrctn and whose structure is displayed in Fig. 5

l.393 Alternatively, the function crest.set_modern_data() could be called instead of crest.get_modern_data() to use personal calibration data instead of the gbif4crest database.

This could already be mentioned in 4.3.3 and 4.3.4.

l.449 Ideally, the climate sampling should be as homogeneous as possible to ensure proper sampling of all the possible climate values, even if the extreme climate values will always be under-represented compared to the median ones. However, deviations from a theoretical one-to-one (or at least proportional) equivalence between climate and occurrence data abundance are not necessarily a bad characteristic.

I know this has not changed much with respect to the previous version of the manuscript, yet I must admit I do not fully understand these sentences.

What exactly is 'the climate sampling'? If c is the climate variable to be reconstructed, I assume, the term refers to the entirety of values for this variable present in the study area, irrespective of the presence or absence of species used for the reconstruction. This relates to my comment with respect to line 107 – a clear distinction between 'the climate space' that comprises all N climate values and the 'the climate sampling' which is comprised only of the $N_s$ climate values accompanied by species occurences (?) would be helpful.

If that interpretation holds true, does the climate sampling contain multiple instances of a climate value $c_i$ if the corresponding grid cell contains multiple independent occurrences of the species?

What is meant by a 'one-to-one equivalence between climate and occurrence data abundance'? Would that be something like, the warmer the clime, the more occurrence data there is in the study region?

l.451 In our case study, the spatial variability represents actual patterns in regional species diversity with the presence of several biodiversity hotspots across the mountainous regions of eastern and southern Africa.

Spatial variability of what?

l.457 (i.e. variables correlated with important variables but do not directly impact the studied proxies; Juggins (2013), Chevalier et al. (2020b))

i.e. variables which are correlated …

Fig.6 number of unique species occurrences

I understand that the author does not want to change the title of the right subplot, however, I would appreciated a lot a clarifying not in the caption, that unambiguously defines the term 'number of unique species occurrences'.

l.508 Here for instance, both Aizoaceae and Chenopodiaceae/Amaranthaceae were excluded because they are not primarily sensitive to temperature in southern Africa.

(from my previous review) To understand this, it would be nice to have them included in the violin plot (Fig.8).

Answer by the author: This exclusion is based on ecological considerations of the taxa, i.e. what is known about them. It is independent of the shape or look of the associated pdf. I think it might in fact be more misleading to plot them because they might not appear widely different from the others.

This does not make sense to me. If these two taxa are not sensitive to temperature, this should be reflected in the data and this be visible in their climate response (or PDF), which in the violin plot should appear more stretched along the y-axis. In fact, the violin plot is advertised a tool to assess the climate sensitivity of the different taxa – the authors answer is not consistent with this role of the violin plot.

Fig.11 Couldn't the strong Loo values for the Ericaceae already be interpreted as some sort of bias in the sense of what you say in line 583?

Large LOO values can arise when the PDFs are biased by unaccounted factors and are, as a result, at odds with the rest of the PDFs.

**Outline for a derivation of Equations (1-5) in a Baysian framework**

In my previous review, I raised concerns about the mathematical accuracy in the derivation of the equations which constitute the basis for the presented method. I understand, that these equations have already been introduced in a similar way by Kühl et al. (2002) and repeatedly presented by the author himself. Yet, I would still argue, that this article and therewith the credibility of the R package would largely benefit from putting equations (1-7) on more solid ground from a mathematical perspective. While equations (1-5) at least seem to be correct, I have expressed my concerns about equation (6) in my previous review and did not find it to be adressed explicitly in the authors replies. The author's key argument to not carry out a derivation of these equations along the lines I present in my previous review, was that these considerations would require 'absence data' of the species $S$. The author is correct, that the considerations I presented in my last review relied on the assumption that for any grid cell 'no presence' would mean 'absence' and hence, this is not the way to go. Yet, it is still possible to thoroughly derive the equations (1-5) starting from Bayes theorem without using 'absence data'. I have made another attempt to provide the author with some thoughts on this below.

The author aim to derive a probabilistic climate reconstruction based on palaeoecological data, that is the presence of a given species (here for simplicity I ignore anything beyond species) at a given time. After careful reevaluation of the manuscript and the paper by Kühl et al. (2002), I came to the conclusion, that the probabilstic climate reconstruction as presented here, is in fact still based on Bayes theorem:

$$\underbrace{\rho_{C|s=1}(c|s=1)}_{\text{posterior probability distribution}} = \frac{\overbrace{\rho_{S|c}(S=1|c)}^{\text{likelihood function}} \rho_C(c)}{\rho_S(S=1)} \quad (1)$$

where the posterior distribution $\rho_{C|s=1}(c|s=1)$ expresses the probability (or one should say plausibility) for the climate variable $C$ to assume the value $c$ given the presence ($s=1$) of the species $S$ at some point in the past (one could add dependency on $t$ in all probability densities, this is omitted here for sake of readibility). It is reasonable to assume, that the likelihood function in the above equation did not change substantially over time, while the prior distribution of the climate $\rho_C(c)$ definitely. Assuming that we have little (or no) prior knowledge about the past distribution of the climate variable it seems a reasonable conservative approach to base the assessment of past posterior distributions of the climate variable only on the likelihood function.

The authors term the likelihood function $\rho_{S|c}(s=1|c)$ — interpreted as a function of $c$ and evaluated at the value $s=1$ — *climate response of a given species* ($\text{PDF}_{\text{sp}}(c,s)$). This function is indepent from the distribution of the climate variable $C$ and after convenient normalization it can also be interpreted as a probability density function with respect to $c$.

The authors estimate the likelihood function as follows: First, they consider the set of tuples $\{(s_i, c_i) : s_i = 1\}_{N_s}$ as a sample with $N_s$ members from the modern distribution

of the climate variable $c$ conditioned on the species presence $\rho^*_{C|s}(c|s=1)$. Assuming that this sample is representative one can approximate the distribution as

$$\rho^*_{C|s}(c|s=1) \simeq \frac{1}{N_s} \sum_{i=1}^{N_s} \delta(c - c_i). \tag{2}$$

Next, application of Bayes theorem relates the *climate response function* or PDF$_s$ with the modern conditioned climate distribution that generated the observed sample:

$$\frac{1}{A} \rho_{S|c}(s=1|c) = \frac{\rho^*_{C|s}(c|s=1)\rho_S(s=1)}{\rho^*_C(c)}, \tag{3}$$

where $\rho^*_C(c)$ is the modern distribution of the climate variable $C$. Computing the mean and the standard deviation from the right hand side allows to approximate the left hand as a normal or log-normal distribution. Eq.(5) in the manuscript can be derived as follows:

$$
\begin{aligned}
\mathrm{E}(C)|_{S=1} &= \int \frac{c}{A} \rho^*_{S|c}(s=1|c)\mathrm{d}c \\
&= \int c \frac{\rho^*_{C|s}(c|s=1)\rho_S(s=1)}{\rho^*_C(c)}\mathrm{d}c. \\
&= \int c \frac{\sum_{i=1}^{N_s} \delta(c-c_i)\rho_S(s=1)}{\rho^*_C(c)}\mathrm{d}c \\
&= \sum_{i=1}^{N_s} \frac{c_i \rho_S(s=1)}{\rho^*_C(c_i)} \\
&\simeq \sum_{i=1}^{N_s} \frac{c_i \rho_S(s=1)}{\rho^*_C(c_i)} \\
&\simeq \frac{1}{\sum_i k(c_i)} \sum_{i=1}^{N_s} k(c_i)c_i.
\end{aligned}
\tag{4}
$$

In the last step, the distribution $\rho^*_C(c)$ was approximated by a coarse grained or local density $k(c)^{-1} = \frac{1}{N}\sum_{j=1}^N \mathbf{1}_{c_j \in \mathrm{bin}(c)}$. Here, $N$ is the number of **all** observations of the climate variable $C$ and not only of the those $c_i$ that coincide with the presence of the species $S$ as indicated by the authors in line 107.

**Several concluding remarks**

- *It seems that Kühl et al. (2002) make a mistake when they claim they would derive pdf(tmp—taxon). I am fairly certain, that what they do derive is pdf(taxon—tmp), as a function of tmp though.*

- *Furthermore, please note that for considerations presented here, no 'absence data' for the speciees is required. This is because the values $c_i$ can be considered as a representative sample from the distribution of $C$, conditioned on the presence of the species.*

- *Currently, the crest package reconstructs the past climate only based on the likelihood function - this assessment could easily be supplemented by prior information on past climate, for example from climate modelling studies carried out with EMICS.*

---

## Author Response (AR2)

I am really grateful that Reviewer 1 looked at my manuscript carefully for the second time and agreed with the proposed rewritten sections. The new suggestions were also quite valuable. However, he/she is expressing some concerns regarding the 'understandability' of the manuscript in places (e.g. the data used). I believe these concerns will not directly apply to colleagues from the palaeoecological community. I understand that Reviewer 1 is from another field, which probably explains some misunderstanding. Lots of field-related 'known unknowns' are associated with palaeoecological datasets, and these can be hard to grasp for colleagues from different fields. This paper cannot, however, start explaining these elements. Otherwise, the size and focus of this manuscript would change too much. But I regularly refer to a recent and exhaustive review I wrote on the topic (Chevalier et al. 2020b), which I believe to be sufficient. Any element I may add to this manuscript to 'explain' palaeoecological datasets would only be a repetition and/or simplification of what is in the review.

I hear his/her concerns regarding the possible mathematical inaccuracies or rather shortcuts. But as he/she mentioned, he/she was able to derive the same results from a complex mathematical suite of Bayesian equations. This is thus a strong validation of the simple equations presented. The goal of this paper is not to be technical but rather to make the use of the technical method appealing to a broad range of potential users, as illustrated by the relative length of Sections 3 to 5 compared to Section 2. Section 2 was only meant to summarise an existing technique and put the package's different parameters in context. By no means did I intend to make a detailed theoretical description. I also want to emphasise that this paper is about the R package and that all the other elements are already exhaustively described in other articles: the method itself (Chevalier et al. 2014), the calibration dataset (Chevalier 2019), the characterisation of pollen taxa climate sensitivities (Chevalier et al. 2021b), and different papers with direct applications (Chevalier and Chase 2015, 2016, and a few more by myself and independent research groups). I hear Reviewer 1's concerns, and I acknowledge that the model presented here could be improved (in many ways). All models can. I am also grateful for all the efforts to reframe my model on more grounded foundations. These results and, more broadly, this discussion through the review process is something that I will consider for future versions of the model or package (as suggested by Reviewer 1). But at this stage, I prefer finalising the package part of the 'crest project', which consists in creating a tool for the broader community to more easily apply an accepted and already used method. Changing the methodology, even by a little, would imply starting a complete phase of development and testing phase to be validated. I believe this to be the work of an entirely new paper.

The comments made by Reviewer 1 are in black and purple, and my responses are in blue.

**General comments:**

The author has presented a revised version of the manuscript with substantially improved readability and many inaccuracies being cleared out with respect to the original version. It would like to point out again, that probably the lion's share of the work has gone into the code of the R-package and the curation of the database and that this work is of highest value for the community.

On the other hand, I must admit that I am still not convinced by the presentation of the mathematics forming the bases of the climate reconstruction method. This criticism comprises twofold: First, the introduction of Equations (1-5) and (7) suffers from several mathematical inaccuracies – however, I noticed that these in parts trace back to the article presented by Kühl et al. (2002). Since the derivation of these equations is by no means the central aim of this paper, these inaccuracies might be acceptable, given that these equations appear to be correct. Also, it seems that some terms have established as domain specific language and thus may be clearer to the target audience than they are to me. Second, I believe that Equation (6) is in fact not correct, even though its use might generate reasonable results. I have expressed my concerns about Eq. (6) already in my previous review.

If not for this manuscript, I am convinced that putting the probabilistic climate reconstruction method on solid ground mathematically would be a beneficial task for the future. I have attached a pdf that outlines the derivation of Eq.(5) starting from Bayes theorem.

The remainder of the mansucript gives the reader a good overview of the *crest* R package, in terms of its capabilities, requirements and usage. It is well structured and the final example of application really takes the reader / user by the hand.

**Specific comments:**

l.2 In particular, the methods based on probability density functions (or PDFs) can be used invarious environments and with different climate proxies because they rely on elementary calibration data (i.e. modern geolocalised presence data).

I would replace 'the methods' by 'methods'. Maybe 'methods based on probability density functions' are just 'probabilistic methods'.

I believe 'the' to be correct because I want to specify a subset of methods from the broad range of existing techniques. Both are grammatically correct, but 'the' includes a nuance that I want to have. And I will keep on referring to 'methods based on probability density functions' because this is how the community knows them.

l.14 It is hoped that crestr will be used to produce the much-needed quantified records from the many regions where climate reconstructions are currently lacking, despite the availability of suitable fossil records.

What is meant 'quantified records'? In my understanding a record is a directly measured time series – so a data processing software could not be used to 'produce a record'? Do you mean 'reconstruction'?

Corrected as follow: "*It is hoped that crestr will be used to produce the much-needed quantified climate reconstructions from the many regions where they are currently lacking, despite the availability of suitable fossil records.*"

l.15 no paragraph in the abstract

Paragraph removed, and sentence linked as follow: "*It is hoped that crestr will be used to produce the much-needed quantified climate reconstructions from the many regions where they are currently lacking, despite the availability of suitable fossil records. To support this development, the use of the package is illustrated with a step-by-step replication of a 790,000 year long mean annual temperature reconstruction based on a pollen record from southeastern Africa.*"

l.19 Over the years, numerous techniques of increasing complexity have been proposed, each one based on a unique set of assumptions regarding the modelling of ecological datasets and their translation into climate reconstructions (e.g. Birks et al. (2010), Chevalier et al. (2020b)).

I assume that with 'ecological datasets' your refer to observations. In that case I would say that observations are not being modeled. Of course, sometimes one uses models to draw inference from datasets and probably that is what you mean?

I do not understand the nuance Reviewer 1 is making here. Any statistical model that exists is derived from observations based on certain assumptions about these observations and is then followed by the modelling of said observations. I did not modify that sentence because its meaning will be clear to the community.

l.26 ...and their accessibility with multiple software solutions.

Saying that an analysis technique is 'accessible with multiple software solutions' sounds strange to me. Do you mean, there exist relatively simple software implementations of the techniques?

Replaced 'with multiple software solutions' with 'via multiple software tools'.

l.26 However, the limited availability of the necessary calibration datasets beyond the Northern Hemisphere extratropics has often hindered their application in many environments and regions where quantified climate records are needed, despite the existence of suitable fossil records (Chevalier et al., 2020b).

Again, what are quantified climate records? It seems you mean reconstruction – in my understanding records are not reconstructions.

'Records' replaced by 'reconstructions'. Here and elsewhere.

l.32 Because modern occurrence data are generally easier to obtain than modern proxy assemblages, this fundamental difference implies that Indicator species methods can contribute to filling in the reconstruction gaps that exist at the global scale.

Grammar: Modern occurrence data are generally easier to obtain than modern proxy assemblages. This fundamental difference implies that Indicator species methods can contribute to filling in the reconstruction gaps that exist at the global scale.
For the non-paleoecologists: what's the difference between proxy assemblages and proxy occurrence data?

The community will understand proxy assemblages, and the meaning of occurrence data is explained in the previous sentence. I did not modify the sentence.

l.36 Derived from the original work of Kühl et al. (2002) —— whoIt

seems there is an extra hyphen in the pdf.

Corrected.

l.37 CREST estimates and combines probabilistic proxy-climate relationships to reconstruct past climate parameters from fossil proxy observations.

Maybe you could add: CREST estimates and combines probabilistic proxy-climate relationships **from modern occurrence data** to reconstruct past climate **variables** from fossil proxy observations.

This sentence is self-explanatory within its paragraph. The definition of the relationships from modern occurrence data is defined in the first part of the sentence (not reported by Reviewer 1, but see original or revised text).

l.45 However, the complexity of collating and formatting the thousands of distinct occurrencesrequired to estimate reliable PDFs limited its practical use.

I understand, that in your context the term PDF carries a very specific meaning. However, in general, this is not the case and a reader that is not used to the specific use of the term PDF as a synonym to your 'climate response functions' will probably struggle to understand the above statement. Also, you introduced the abbreviation PDF only in the abstract, but up to this pointnot in the main text.

I actually wanted to remove the use of the acronym 'PDF' before the methods section – thank you for noticing this last occurrence. I have replaced it with 'proxy-climate relationships' to keep the naming coherent across the text.

l.48    'climate records'

Corrected.

l.50    This paper thus introduces  the new multi-platform R package crestr designed to replace the original interface.

Keeping 'a'.

l.51    crestr includes the global calibration dataset

It is linked to the dataset but does not include it – strictly spoken.

'includes' replaced with 'integrates'. The nuance Reviewer 1 wants to make is correct, but I think it is also important to highlight in a simple way that the package comes with calibration data. Practically, many tools that exist are difficult to use because of this absence of calibration data. I do not want to lose this important point to indicate early on that the calibration is in a cloud-based database and not with the package itself. The information is available in the relevant section.

l.59    As such, the climate reconstructions obtained from CREST can be understood as an ensemble of all data-compatible climate values

Maybe 'As such, the application of CREST yields a probabilistic quantification of the past climate in view of the data under study as opposed to simpler, less informative 'most likely' or 'best' climate estimates. While the latter may only capture statistical uncertainties, the former rigorously takes into account the large quantitative uncertainties inherent to analysis of this kind.'

Just to highlight a little bit the fact, that in terms of uncertainty propagation *crest* performs a lot better than the mentioned 'best-estimate' methods.

I like the first sentence suggested by Reviewer 1. The second one is also very interesting from a technical perspective but I fear that it would raise more questions. I have thus rephrased the sentence as follow: "*As such, the application of CREST yields a probabilistic quantification of all the climate values that are compatible with the studied data instead of simpler, less informative 'most likely' or 'best' climate estimates. While the 'best estimate' approach might be optimal when a fossil assemblage is analysed in complete isolation, the presence of independent - local or regional - information (_e.g._ other reconstructions from the same core or independent records) usually provides additional information that may not always be consistent with best estimate reconstructions.*"

fig.1    Conceptual illustration of the differences between a modelling approach based on the estimation of the full spread of the data with the probabilities spread along the climate gradient (e.g. CREST; dark grey), and a modelling approach focused on the estimation of the 'most likely' or 'best' climate value with small statistical errors surrounding it (e.g. MAT or WA-PLS; lightgrey).

I would say the probabilistic approach is based on the 'full spread of the data' but not on the 'estimation of the full spread of the data'.

Corrected.

l.96 The **individual?** climate responses of all the species identified are estimated as univariate probability density functions (PDFs) for every climate variable.

Yes, corrected.

l.99    The individual species'

Corrected as 'The species' individual climate responses are' for clarity.

l.99    estimation of the empirical mean (m s,c )

An empirical mean is  not estimated,  but computed  from  the data.  Just delete 'the  estimation of' .

Corrected, indeeed.

l.107 Here, the weights are calculated by first sorting the N climate values (all the c i ) that composethe modern climate space into bins of equal width (e.g. 2 ◦ C or 50 mm). Then, each climatevalue c i is given a weight k(c i ) defined as the inverse of the relative size of the bin c i itbelongs to:

1. As far as I understand N is the number of gridded observations of the  modern  climatevariable c, while N_s is the number occurrences of the species S. According to line 100, the c_i are only those climate values, that coincide with the occurrence of the species. However, for the weights  k(c_i), the entire climate space should be taken into account. So the addition (all the c_i) is not correct. Maybe, the difference between the c_i and **all** climate values form the study area can be clarified somehow.

I have tried to improve the labelling of all the quantities used in this modelling. For example, I defined Cs,i as the climate values where species s is observed (i varies between 1 and Ns).

2. The first sentence defines all bins  such  that they have  equal width. The second sentence refers to different 'sizes' of the bins. I assume that the 'size' of a bin here means the number of observations  falling  into  one  bin.  However,  typically  one  would  use  'size'  and  'width' interchangeably, so 'size' might be misleading, here.

Rephrased as: "*defined as the inverse of the number of values that belong to same bin $\text{bin}_{c_{j}}$:*".

l.117   Once estimated, m s,c and s 2 s,c are used to define a regular, unimodal distribution for the PDF sp (s, c) of species s for climate variable c. Here, we assume that the shape of these species responses should be unimodal and can be either normal:

I know it is a detail, but the term 'distribution' is slightly misused in this  context.  The distribution of a random variable in statistics defines the the probability for the random variableto assume a certain value. Hence, here the PDF **is** a distribution.

Maybe: 'Assuming unimodality and either normality or log-normality, the estimated m_(s,c) and s²_(s,c) are used to define the species climate response PDF's as follows: …'

Rephrased as: "*Once estimated ${m}_{s,c}$ and $s_{s,c}^2$ are used to define a regular, unimodal $\text{PDF}_{\text{sp}}(s,c)$ for species $s$ and climate variable $c$.*"

l.120   delete paragraph
No.

l.125  I have expressed my doubts about this equation already in my previous review. After going through the math once again, I still belief this equation is not correct – even though it might lead to reasonable results. Also in the cited literature, I could not find a convenient derivation of Eq.(6).

I have acknowledged Reviewer 1's concerns in my previous response, and this is currently under investigation. This model can probably be improved – every model can – but this is beyond the scope of the present paper, which consists in presenting an implementation of a model that has been successfully used for many years now.

l.126  delete paragraph

No. I really do not understand why Reviewer 1 would suggest this deletion. Defining all the parameters and transformations is important. In addition, I believe this paragraph (and the ones below that Reviewer 1 also suggests to delete) to be very important for non-statisticians to understand what things are, and also what they are not.

l.138  Climate c is reconstructed from fossil sample z (z can be an age, depth or any identifyer) by multiplying the PDF tx (t, c) of the T (z) selected taxa:

- Maybe 'Past climate c that corresponds to a specific age (or depth) z from which a fossil sample is available can be reconstructed…'

This suggested phrasing will appear convoluted to colleagues with palaeoecological datasets. The working unit is the sample, and it is more often than not irregularly sampled along with age and depth. I am not convinced that indexing on depth would make sense, especially since many depths will have no data associated. The idea here is only to indicate that it is applied for each sample. I believe that too much mathematical formalisation can sometimes add unnecessary complexity, especially for non-experts, which ultimately masks the intended message.

- you first assign z := the fossil sample and then state that either z:= age, or z:= depth.
  - what exactly is 'the fossil sample'? A dataset comprised of abundance data of different taxa on a depth or age axis? This is a little inaccurate.

Rephased as: "*Climate $c$ is reconstructed from fossil sample $z$ (itself associated with a unique age, depth or any other identifier) by multiplying the $\text{PDF}_{\text{tx}}(t,c)$ of the $T(z)$ selected taxa*:"

l.141  delete paragraph

No.

l.151  The summation index should be 'i' or sth else and then the depths / ages should be discretized as z_i, but the summation index cannot be a continuous variable. The inner parenthesis in the denominator are not required.

'z' indicates the samples across the paper. It is definitely not a continuous variable (see response aboce). I do not want to add another layer of variable indexing – this would be too complex for something that is fairly simple: sum across all samples.

I have actually removed the outer parentheses. I prefer keeping the inner ones for clarity even if they are not mandatory.

l.152  delete paragraph

No.

l.181 In the gbif4crest database, all the QDGC grid cells were associated with a collection of terrestrial and oceanic environmental variables that can be reconstructed (Fick and Hijmans (2017), Zomer et al. (2008), Locarnini et al. (2019), Zweng et al. (2018), Garcia et al. (2019a), Garcia et al. (2019b), Reynolds et al. (2007), see details in Tables 1 and 2).

Does this mean, that all variables listed in Tab.1 and Tab.2 are reconstructable? If so, I would suggest to express this a bit more clearly and also emphasize this in the table captions.

Clarified in the captions as: "*List of terrestrial/marine variables available for reconstruction in the \textit{gbif4crest} database. Each one can be selected in \textit{crestr} using its associated code. List of abbreviations: (Temp.) Temperature, (Precip.) Precipitation.*".

l.193 For example, the first version of the gbif4crest dataset released in 2018 contained about 17.5 million QDGC entries, while the new version contains approximately 25.3 million entries (~44% increase).

Why QDGC entries and not only entries? The second is a duplication of the first sentence of the paragraph and can be deleted, except the (~44% increase).

Only the QDGC data are used for the constructions. The sentence was corrected as suggested.

l.230 Maybe, it would be helpful to explain, which function call initializes the crestObj in first place already at this stage?

Sentence corrected as follow: "*In crestr, all the CREST-related data are stored within a single S3 object of the class crestObj that is first initialised by either crest.get_modern_data or crest.set_modern_data (see section 5.2 for details).*".

l.257 Does the df have as many columns (+1 for the depth/ age) as fossil taxa are considered for the climate reconstruction?
Absolutely. See text with "with either the age, depth, or sample ID as the first column and the fossil data in the subsequent columns." The package also contains an example dataset, and there is the Limpopo dataset also shared as supplementary material. There is also online documentation to help people build their datasets https://mchevalier2.github.io/crestr/articles/get-started.html.

l.280   In a second time

In a second step

Replaced with 'in subsequent steps'.

l.281   Mabe here, a sentence like

*The pdf for the fossil taxon Stoebe-type is thus comprised of the linear combination of species pdfs according to equation (6) associated with all species that fall into the geni Stoebe and Elytropappus.*

would be helpful at this stage, provided that my interpretation is correct.

This is a good suggestion to make this step more relatable to the data modelling. I have clarified this at the

beginning of the PSE section.

*"The PSE data frame is required to use the gbif4crest calibration dataset. It is used to associate individual species available in the TAXA table with their corresponding fossil taxon. This step is important to estimate the species responses (PDFsp(s,c)) and taxon responses (PDF_{tx}(t,c)) described in Section 2.2."*

l.284 Additional taxa can also be added to the PSE file to exclude species known not to be part of a group. For instance, this 'trick' could have been used to simplify the climate response of the 'Asteraceae undiff.' group by excluding more species from it, even if the pollen grains corresponding to these species have not been observed.

I would suggest to move the 'even if… ' to the first sentence.

Corrected.

l.290 If I understand correctly, if I provide a 'distributions' table as an input to the get_modern_data() function, then I do not need the PSE is that correct? Why is the 'distributions' table not listed in Fig. 5 in the 'input' category? If users decide to use the 'distributions' table as input, then they have to provide climate_space data frame as well, is that correct?

It would be an input to 'set_modern_data()' but that is correct. The distribution data go into the 'modelling' part of the object – same as the other ones – because they are used to infer the climate responses. This is imperfect, but the possibility of using other datasets was a late addition to the package. Users will have to add a climate_space dataset if they want to use the weighting option and plot maps. I would certainly recommend using it, but it is not mandatory. I have made this distinction more obvious.
*"Including a `climate_space` dataset is recommended, even if it is not mandatory."*

l.392 here called rcnstrctn; Fig. 5

here called rcnstrctn and whose structure is displayed in Fig. 5

Corrected.

l.393 Alternatively, the function crest.set_modern_data() could be called instead of crest.get_modern_data() to use personal calibration data instead of the gbif4crest database.

This could already be mentioned in 4.3.3 and 4.3.4.

I tried to avoid having too many cross-references across sections. In my opinion, it would only complexify the first reading of the paper. And in practice, knowing which functions will be used is not necessary to build the correct datasets. In addition, there is online documentation with further details. An R package is a complex tool and accounting for all possibilities is difficult and probably not recommended. I present a way of running such analyses, and if people follow each step one by one they will obtain results.

l.449 Ideally, the climate sampling should be as homogeneous as possible to ensure proper sampling of all the possible climate values, even if the extreme climate values will always be under- represented compared to the median ones. However, deviations from a theoretical one-to-one (or at least proportional) equivalence between climate and occurrence data abundance are not necessarily a bad characteristic.

I know this has not changed much with respect to the previous version of the manuscript, yet I must admit I do not fully understand these sentences.

What exactly is 'the climate sampling'? If c is the climate variable to be reconstructed, I assume, the term refers to the entirety of values for this variable present in the study area, irrespective of the presence or absence of species used for the reconstruction. This relates to my comment with respect to line 107 – a clear distinction between 'the climate space' that comprises all N climate values and the 'the climate sampling' which is comprised only of the $N_s$ climate values accompanied by species occurences (?) would be helpful.

A definition of what is meant by climate space is already provided in section 2.1. Then N and Ns are defined in section 2.2 with N corresponding to the size of the climate space and Ns being the climate values where species s is observed.

If that interpretation holds true, does the climate sampling contain multiple instances of a climate value $c_i$ if the corresponding grid cell contains multiple independent occurrences of the species?

By construction, every species is only observed once in a grid cell. So the Ci are unique. However, if several species part of the same pollen taxon are observed in the grid cell, the Ci will be counted once for each.

What is meant by a 'one-to-one equivalence between climate and occurrence data abundance'? Would that be something like, the warmer the clime, the more occurrence data there is in the study region?

The paragraph was rephrased as follow to make it simpler. "*Ideally, the climate values sampled by the calibration data should be as homogeneous as possible to ensure proper representation of all the possible climate values, even if the extreme climate values will always be under-represented compared to the median ones. However, deviations from a theoretical equivalence between the observed climate distribution and the climate values sampled by the calibration data are not necessarily a bad characteristic. In our case study, the variability of the sampling density represents actual patterns in regional species diversity with the presence of several biodiversity hotspots across the mountainous regions of eastern and southern Africa [@Myers_etal_2000]. This higher diversity in the colder areas explains why the black histogram (_i.e._ the climate values associated with occurrence data) on Fig. \ref{fig:climatespace} is skewed towards the left compared to the grey histogram (_i.e._ the distribution of the climate space in the study area). All these elements should be checked and accounted for while designing the final calibration dataset.*"

l.451   In our case study, the spatial variability represents actual patterns in regional species diversity with the presence of several biodiversity hotspots across the mountainous regions of eastern and southern Africa.

Spatial variability of what?

Corrected as "the variability of the sampling density". See above.

l.457 (i.e. variables correlated with important variables but do not directly impact the studied proxies; Juggins (2013), Chevalier et al. (2020b))

i.e. variables which are correlated …

Corrected.

Fig.6 number of unique species occurrences

I understand that the author does not want to change the title of the right subplot, however, I would appreciated a lot a clarifying not in the caption, that unambiguously defines the term 'number of unique species occurrences'.

The term is first defined in the introduction, further explained in section 2.1, and these data represent the core of the entire approach. I am confused by the fact that Reviewer 1 is still unclear about what these data are about. He/She seems to have understood them correctly though, especially since he/she has generously offered a Bayesian translation of my model.

l.508   Here for instance, both Aizoaceae and Chenopodiaceae/Amaranthaceae were excluded because they are not primarily sensitive to temperature in southern Africa.

(from my previous review) To understand this, it would be nice to have them included in the violin plot (Fig.8).

Answer by the author: This exclusion is based on ecological considerations of the taxa, i.e. what is known about them. It is independent of the shape or look of the associated pdf. I think it might in fact be more misleading to plot them because they might not appear widely different from the others.

This does not make sense to me. If these two taxa are not sensitive to temperature, this should be reflected in the data and this be visible in their climate response (or PDF), which in the violin plot should appear more stretched along the y-axis. In fact, the violin plot is advertised a tool to assess the climate sensitivity of the different taxa – the authors answer is not consistent with this role of the violin plot.

This is exactly what confounding factors are. Due to modern correlations between unrelated parameters, they can both "look like" they are good predictors. But independent ecological knowledge tells us that their presence is related to other factors, e.g. evaporative conditions and relatively saline environments. The violin tool is advertised as one of the tools that can be used to make such an assessment. I also strongly refer to Fig. 7 to help identify these sensitivities (what does the complete distribution look like? When is the taxon present/absent/abundant? Are different variables correlated? etc). Finally, the reference Chevalier et al. 2021b, which is mentioned several times in the paper, is entirely focused on this specific aspect.

Pollen data are really complex data and this type of assessment is commonly made by palynologists. It is probably the limited inside knowledge that makes this surprising or confusing to Reviewer 1.

Fig.11   Couldn't the strong Loo values for the Ericaceae already be interpreted as some sort of bias in the sense of what you say in line 583?

Large LOO values can arise when the PDFs are biased by unaccounted factors and are, as a result, at odds with the rest of the PDFs.

It could in theory, but it doesn't in this case. The ecology of Ericaceae is well-known and it is a taxon that thrives in the tropics during cold periods. This type of analysis is really an interplay between statistics and ecology, and both should always be accounted for to define the assumptions and interpret the results.